# 📊 Spectrum Tuning: Post-Training for Distributional Coverage and In-Context Steerability

**Taylor Sorensen**[1], **Benjamin Newman**[1], **Jared Moore**[2], **Chan Young Park**[3], **Jillian Fisher**[1], **Niloofar Mireshghallah**[4], **Liwei Jiang**[1], **Yejin Choi**[2]

[1]University of Washington, [2]Stanford University, [3]Microsoft Research, [4]Carnegie Mellon University
Correspondence: `tsor13@cs.washington.edu`, `yejinc@stanford.edu`
⚙ Code and Dataset: `github.com/tsor13/spectrum`
🤗 Models: `huggingface.co/collections/tsor13/spectrum`

## Abstract

Language model post-training has enhanced instruction-following and performance on many downstream tasks, but also comes with an often-overlooked cost on tasks with many possible valid answers. On many tasks such as creative writing, synthetic data generation, or steering to diverse preferences, models must cover an entire distribution of outputs, rather than a single correct answer. We characterize three desiderata for conditional distributional modeling: in-context steerability, valid output space coverage, and distributional alignment, and document across three model families how current post-training can reduce these properties. In particular, we disambiguate between two kinds of in-context learning: ICL for eliciting existing underlying knowledge or capabilities, and *in-context steerability*, where a model must use in-context information to override its priors and steer to a novel data generating distribution. To better evaluate and improve these desiderata, we introduce SPECTRUM SUITE, a large-scale resource compiled from >40 data sources and spanning >90 tasks requiring models to steer to and match diverse distributions ranging from varied human preferences to numerical distributions and more. We find that while current post-training techniques elicit underlying capabilities and knowledge, they hurt models' ability to flexibly steer in-context. To mitigate these issues, we propose SPECTRUM TUNING, a post-training method using SPECTRUM SUITE to improve steerability and distributional coverage. We find that SPECTRUM TUNING often improves over pretrained and typical instruction-tuned models, enhancing steerability, spanning more of the output space, and improving distributional alignment on held-out datasets.

## 1 Introduction

Current post-training recipes (Rafailov et al., 2024; Tie et al., 2025; Wang et al., 2025) have made language models (LLMs) easier to use via instruction-following (Ouyang et al., 2022), improved safety, and led to performance increases across many tasks, especially those with a single correct answer (e.g., mathematical reasoning, programming, chat preferences, etc.). However, the effect of current post-training on tasks requiring steerability and distribution matching is less studied. We show that current post-training can also negatively impact three related desiderata for conditional distributional modeling: in-context steerability, output coverage, and distributional alignment.

In this paper, we contribute: 1) an outline of these related desiderata, including the novel concept of *in-context steerability*; 2) SPECTRUM SUITE, a dataset for evaluating and enhancing these desiderata; 3) a novel finding that while current post-training helps at many objective tasks, it can *hurt* LLMs' in-context steerability; and 4) empirical evidence from our and related work that current post-training hurts output coverage and distributional alignment. To alleviate these weaknesses, we contribute 5) SPECTRUM TUNING, a post-training technique utilizing SPECTRUM SUITE to improve these desiderata, and 6) show that our method enhances these properties compared to pretrained and

current instruction-tuned models. To our knowledge, our method is the first to improve distributional alignment over pretrained models.

## 2 DESIDERATA FOR CONDITIONAL DISTRIBUTIONAL MODELING

Before the age of post-training, in-context learning was necessary to reliably get pretrained language models to perform tasks such as sentiment classification, translation, entailment, summarization, etc. (Brown et al., 2020; Dong et al., 2024). Let us call this use of in-context learning *capability elicitation*, as its main purpose is to elicit some latent knowledge or capability of a language model (Min et al., 2022b). As post-training methods have increased LLMs' instruction-following capability, zero-shot instruct models have even surpassed their few-shot pretrained counterparts (Wei et al., 2022; Sanh et al., 2022; Ouyang et al., 2022), obviating the need for in-context capability elicitation.

**In-Context Steerability.** In contrast to knowledge elicitation, many tasks require steering, or modifying output probabilities, based on novel information at inference time. For example, if a user wants an LLM to write an email in their style, it needs to either see examples of their writing or have an in-depth description of their style, and be able to effectively leverage this information to change its output distribution. This is distinct from pure capability/knowledge elicitation on unambiguous tasks, where the model can place a sharp prior on the "correct" answer. Instead, the model must 1) maintain a prior over many possible generation functions and 2) maximally leverage in-context information in a well-calibrated way to form a posterior. Let us term this ability *in-context steerability*. For example, this steerability is necessary for predicting a particular user's preferences or estimating an unknown numerical distribution from draws. In-context steerability can also be seen as implicit Bayesian reasoning (Qiu et al., 2025) or as a subset of in-context learning/instruction-following tasks where the model must utilize novel information in-context.

**Valid Output Coverage.** Many prompts entail multiple valid responses. For example, in creative story-writing, hypothesis proposal, and synthetic data generation, the number of possible valid outputs can be thousands or more. While in some cases it may be sufficient to produce one reasonable output, more value may lie in producing *many* outputs so that a user can select the most interesting story, test all possible hypotheses, or otherwise span the entire task space. In the words of Wilson & Izmailov (2022), "we want the support of the model to be large so that we can represent any hypothesis we believe to be possible, even if it is unlikely."

**Distributional Alignment.** Sometimes, a user may not want a particular output, but rather a *distribution* over outputs (Meister et al., 2024). For example, Sorensen et al. (2024b) propose *distributional pluralism* for modeling or representing a population by matching their opinion distribution. In addition, distributional alignment can simulate stochastic processes and estimate uncertainty. Distinct from valid output coverage, distributional alignment includes a target probability mass function.

## 3 DATASET AND METHOD

### 3.1 SPECTRUM SUITE 🟢

To measure and elicit these properties, we compile datasets that either 1) exhibit natural person-to-person variation (e.g., opinion modeling, chat preferences, subjective NLP tasks); 2) involve a large collection of interchangeable texts drawn from a particular distribution (e.g., synthetic data, poems in a particular format); 3) are i.i.d. draws from a random distribution (e.g., draws from a normal distribution); or 4) involve rea-

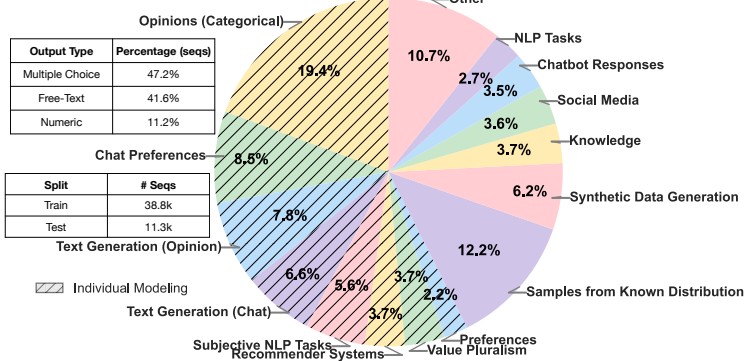

| Output Type | Percentage (seqs) |
|---|---|
| Multiple Choice | 47.2% |
| Free-Text | 41.6% |
| Numeric | 11.2% |

| Split | # Seqs |
|---|---|
| Train | 38.8k |
| Test | 11.3k |

Figure 1: Task composition from SPECTRUM SUITE. Individual modeling tasks (data from the same person) are shaded.

soning under uncertainty. We draw from $>40$ data sources in order to make $>90$ separate tasks. We unify each task data into a common format including: `description`: a natural language description of the task, `input`: any given information for a particular data instance, and `output`:

the output sequence which we would like the model to learn. Some tasks require an `input` associated with each `output` (e.g., the question asked in a survey is needed to contextualize the answer), while other tasks consist of only `outputs`. In particular, we focus on individual modeling data on tasks with human variation. We do so for a couple of reasons: many use cases involve steering to a particular individual at inference time; and these data sources are very rich as modeling each person involves a different data generation task. These data comprise 50.1k distinct sequences consisting of a `description` followed by multiple `inputs/outputs`. For summary statistics and task breakdown of SPECTRUM SUITE, see Figure 1. For information on all data sources, see App. D. We split SPECTRUM SUITE into non-overlapping train and test tasks, with held-out test tasks drawn from separate data sources to ensure generality.

## 3.2 SPECTRUM TUNING 📊

Let $T_i \in \mathcal{T}$ be some task (or, data generation process) that we want to model. Let $Y_i$ be the `output` space to approximate, $X_i$ be any known covariates (optional `input`), and $Z_i$ be a latent context for the task (optional `description`). $T_i : X_i, Z_i \rightarrow P(Y_i)$ maps to a probability distribution over potential outputs. This is the classic meta-learning formulation (Hospedales et al., 2020), except that the target is a distribution over $P(Y^i)$ instead of a single $y_i$. Because the task $T_i$ may be difficult to directly observe, we may instead wish to learn it from data (e.g., Monte Carlo samples).

The method (Algorithm 1) is simple: for a collection of tasks, tokenize the task context/description $z_i$ followed by (randomly ordered) in-context examples $x_{ij}, y_{ij}$, then perform supervised finetuning calculating cross-entropy loss *only* on the output tokens. Because cross-entropy loss on Monte Carlo samples from a distribution encourages a well-calibrated estimate of the underlying distribution in the underfit regime ($\leq 1$ epoch, Ji et al. 2021) the optimal model solution is to approximate the true underlying distribution $P(Y_i)$.

To build intuition on how SPECTRUM TUNING supports the desiderata, let us consider a few cases. When a model predicts the first output, it must rely only on the description, and shift its probabilities to outputs fitting the description. Because there can be many possible valid outputs and the model has no information about which output to expect, it is incentivized to *cover* the entire possible distribution of outputs. Additionally, if the distribution over valid outputs is skewed in some predictable way (e.g., an opinion distribution),

---

**Algorithm 1** SPECTRUM TUNING

**Inputs:** Pretrained LM $m_\theta$; train task distribution $\mathcal{T}^{\text{train}}$; tokenizer $t(\cdot)$ with template for description/input/output; terminal token $\langle\text{END}\rangle$; loss ignore index $i_{\text{drop}}$; description drop probability $p_{\text{drop}}$ (default 0.2).

**Output:** Finetuned parameters $\theta'$

1: **for** each task $T \sim \mathcal{T}^{\text{train}}$ **do** ▷ Sample a task
2:     Sample description $z$ and support set $S = \{(x_j, y_j)\}_{j=1}^n$.
3:     Randomly permute indices $\pi$ of $\{1, \ldots, n\}$.
4:     **if** $\text{Uniform}(0,1) > p_{\text{drop}}$ **then** ▷ Keep description
5:         $seq \leftarrow t(z)\|t(x_{\pi[0]})\|t(y_{\pi[0]})\|\langle\text{END}\rangle$
6:         $labels \leftarrow i_{\text{drop}}(t(z)\|t(x_{\pi[0]}))\|t(y_{\pi[0]})\|\langle\text{END}\rangle$
    ▷ Loss on first output, no loss on description/output
7:     **else** ▷ Description dropout w/ prob. $p_{\text{drop}}$
8:         $seq \leftarrow t(x_{\pi[0]})\|t(y_{\pi[0]})\|\langle\text{END}\rangle$
9:         $labels \leftarrow i_{\text{drop}}(t(x_{\pi[0]})\|t(y_{\pi[0]}))\|\langle\text{END}\rangle$ ▷ No loss on first output if description is missing
10:     **end if**
11:     **for** $j$ in $\pi[1:]$ **do** ▷ Add remaining
12:         $seq \leftarrow seq \| t(x_j) \| t(y_j) \| \langle\text{END}\rangle$
13:         $labels \leftarrow labels \| i_{\text{drop}}(t(x_j)) \| t(y_j) \| \langle\text{END}\rangle$ ▷ Loss on output, no loss on input
14:     **end for**
15:     $L \leftarrow \text{CrossEntropy}(m_\theta(seq), labels)$
16:     $\theta \leftarrow \theta - \eta \nabla_\theta L$
17: **end for** ▷ Train for one epoch
18: **return** $\theta' \leftarrow \theta$

---

the model is further incentivized to *match* said distribution. On subsequent outputs, the model must *steer* its output distribution, utilizing in-context examples to update its beliefs in a well-calibrated way. Additionally, SPECTRUM SUITE tasks allow the model to utilize assumptions which don't apply to the pretraining distribution: predictions are invariant to output ordering,[1] the underlying generative process remains constant, and the model can concentrate all probability mass on valid outputs instead of on other possible text continuations. In many ways, SPECTRUM TUNING is similar to supervised fine-tuning on instruction data (Zhang et al., 2025c), as loss is calculated only on an output. However, it differs in several important respects: 1) many identically-distributed outputs are

---

[1]i.e. "exchangeable" in Bayesian analysis (Kokolakis, 2010), as the posterior is invariant to sample order.

included in-context, encouraging meta-learning; 2) training on data that is distributional in nature; 3) sole focus on distribution fitting instead of chat-style data; and 4) inputs are optional, unlike chat user messages which are always required.

## 3.3 IMPLEMENTATION DETAILS

We train models from three families using SPECTRUM TUNING on the train tasks from SPECTRUM SUITE: `gemma-3-12b` (Team et al., 2025), `Llama-3.1-8B` (Grattafiori et al., 2024), and `Qwen3-14B` (Yang et al., 2025). We refer to pretrained or base models as PT models and instruction-tuned post-trained models as IT models, and utilize each family's provided PT/IT model as baselines. To match our meta-learning task setup (as opposed to chat), we adapt each model's chat template to use the `description/input/output` roles instead of `system/user/assistant` (cf. Fig. A2). For SPECTRUM TUNING, we initialize with the PT model weights, except for the uninitialized (un/)embedding weights for the two or three special format tokens which we initialize from the IT model. See App. G for more training details.

## 4 IN-CONTEXT STEERABILITY

We use SPECTRUM SUITE to evaluate models' ability to steer to varied generation tasks. We measure $k$-shot learning by 1) fitting the description and examples from a single task into context, 2) measuring the loss (negative log-likelihood) of each `output` conditioned on the prior examples under the model $m_\theta$: $\mathrm{NLL}_{m_\theta}(y_k) = -\log p_{m_\theta}(y_k|z, y_0, \cdots, y_{k-1})$. Additionally, for multiple-choice datasets, we calculate the accuracy of the output: whether the greedily-decoded model response results in the correct answer. For each task, we choose $K_{\max}$ such that it maximizes the total number of examples that we can evaluate when we restrict to only sequences with at least $K_{\max}$ examples that fit into a 1024-token context-window. In order to maximize sample efficiency and evaluate a model's ability to steer for varied $k$, we report the average loss and accuracy for $k$-shot learning for $k \in \{1, \cdots, K_{\max}\}$.

First, we ask: how does current instruction-tuning impact in-context steerability? For the PT models, we use the same prompt template for all models, with `Description:/Input:/Output:` delineated by newlines. To ensure we are leveraging maximum performance from the IT models, we test each IT model's

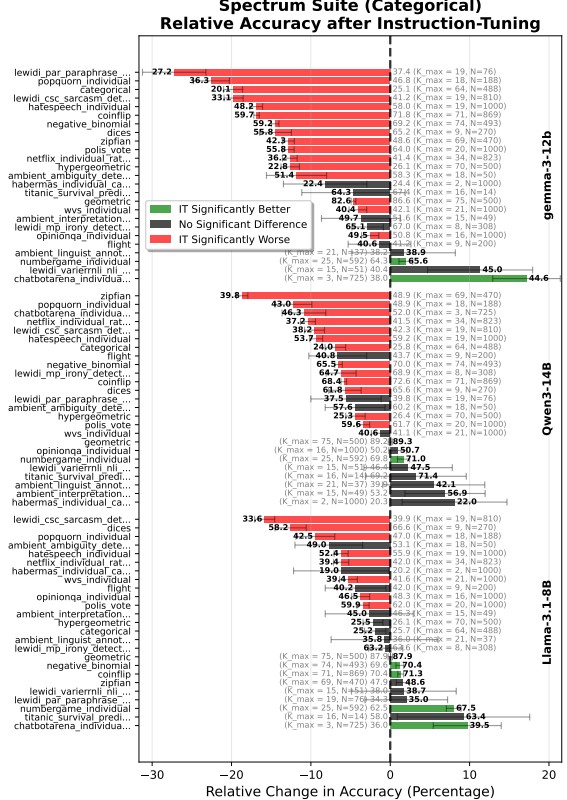

Figure 2: Change in accuracy on SPECTRUM SUITE from the pretrained to instruction-tuned model. Current instruction-tuning hurts in-context steerability.

performance on both the PT prompt and two chat-style ICL prompts, and report results for the best performing prompt template (see App. M). We evaluate in-context steerability on all of SPECTRUM SUITE for the PT/IT models. We include the entire suite of results in Appendix K, and highlight the principal results below.

**Current instruction-tuning hurts in-context steerability.** First, let's examine the change in accuracy for the IT models. We report accuracy for all categorical data (multiple-choice + small support numeric distributions) in Figure 2. Out of 76 model family/task comparisons, instruction-tuning *significantly decreases* accuracy in 35 cases, doesn't significantly affect accuracy in 33 cases, and significantly increases accuracy in only 7 cases. Additionally, two of the seven comparisons where instruction-tuning helped were on predicting an individual's chatbot preferences—which is adjacent to precisely what instruct models are optimized for (chat). The performance drop is even more stark

on loss: for Gemma and Qwen, loss is higher on 50/50 comparisons, while on Llama loss is worse in 11 cases, the same in 11 cases, and better in 3 cases. Loss results are similar on the free-text SPECTRUM SUITE datasets: out of 144 comparisons, IT loss is worse than PT loss in 117 cases, tied in 25 cases, and better only in 2 cases.

**ICL for general capability elicitation is not degraded by instruction-tuning.** To disambiguate in-context steerability from general capability elicitation, we also run the exact same experiment with eight general capability task datasets (Fig. 3). In contrast with the SPECTRUM SUITE datasets, accuracy *increases* in 8 of 24 cases, is the same in 13 cases, and decreases in 2 cases.

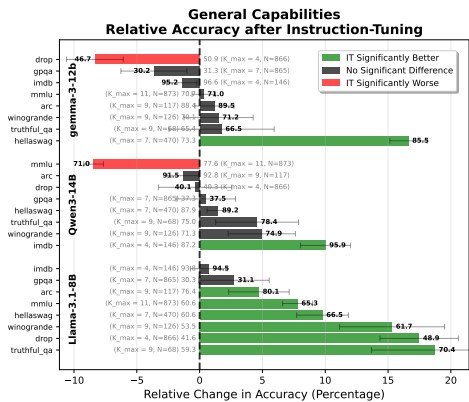

All in all, we believe that this characterizes a difference in behavior for IT models—while they maintain the ability to utilize in-context demonstrations for general capability elicitation, they seem to struggle to adapt at tasks that require heavy in-context steerability. Limited prior work has suggested that instruction-tuned models sometimes perform better without in-context examples (Asai et al., 2024; Lambert et al., 2025); however, to our knowledge, ours is

Figure 3: Current instruction-tuning generally helps on capability benchmarks.

the first work to empirically characterize this in-context learning performance degradation for in-context steerability tasks.

What explains this difference? While we leave an in-depth exploration of this phenomenon to future work, we hypothesize that it could be due to some combination of 1) instruction-tuning inducing very strong priors that are difficult to override even with in-context demonstrations, 2) over-optimization on tasks with a single ground truth, or 3) overfitting to particular benchmarks.

### 4.1 SPECTRUM TUNING AND IN-CONTEXT STEERABILITY ON HELD-OUT TASKS

We have characterized that current instruction-tuned models struggle at in-context steerability, but how does our method compare? We evaluate Spectrum-Tuned (ST) models on SPECTRUM SUITE test tasks and compare them to their PT and IT counterparts (Table 1). Note that the test task data sources have no overlap with the train split, requiring generalization.

**SPECTRUM TUNING usually matches, and sometimes improves upon, PT steerability.** Out of 15 multiple-choice (MC) loss comparisons, ST ties with PT models in one case and achieves lower loss compared to PT models in 14 cases. On MC accuracy, ST matches/improves/worsens on 10/3/2 comparisons. On the free-text datasets, ST matches PT in 28 cases, is worse in 1 case and is better in 4 cases. In most cases, SPECTRUM TUNING matches (but does not beat) the very strong baseline of a pretrained model at in-context steerability, but does improve performance more often than it hurts performance.

**Models trained with SPECTRUM TUNING most often have the best calibration.** We report calibration in Table 2. In 9/15 cases, the ST models have the best calibration. Additionally, the Gemma and Qwen IT models have worse calibration in 10/10 cases than their pretrained counterparts, showing another side effect of heavy instruction-tuning (cf. Tian et al. 2023; OpenAI et al. 2024).

### 5 SPANNING THE OUTPUT SPACE (OR; DIVERSITY VS. VALIDITY)

To measure how each model trades off validity and diversity, we create 22 generation tasks for which there can be many valid values and we can programmatically verify correctness ($\mathbb{1}_{\text{correct}}$). Given a prompt, we generate 100 completions $o_1, \cdots, o_{100}$ (temperature $= 1$ here and throughout) from each model, and report the following statistics: the percentage of outputs which are valid ($\sum_{i=1}^{100} \mathbb{1}_{\text{correct}}(o_i)$), the percentage of valid generations that are unique ($\frac{|\text{dedup}(\{o_i : \mathbb{1}_{\text{correct}}(o_i)=1\})|}{\sum_{i=1}^{100} \mathbb{1}_{\text{correct}}(o_i)}$), and the number of distinct valid generations (or, *yield*: $|\text{dedup}(\{o_i : \mathbb{1}_{\text{correct}}(o_i) = 1\})|$). We perform deduplication with exact string matching. Yield is a particularly important metric for settings such as synthetic data generation, ideation, or creative writing where you want to cover a space as much

| Multiple-Choice Datasets | Metric | gemma-3-12b | | | Qwen3-14B | | | Llama-3.1-8B | | |
|---|---|---|---|---|---|---|---|---|---|---|
| | | ST (ours) | PT | IT | ST | PT | IT | ST | PT | IT |
| habermas_individual_categorical ($K_{max}$=2, N=1000) | **Loss** | **2.47** | **2.50** | 10.5 | **1.97** | 2.62 | 9.10 | **1.99** | 2.58 | 2.74 |
| | **Acc** | 23.8 | **24.4** | 22.4 | **23.5** | 20.3 | 22.0 | **20.8** | 20.2 | 19.0 |
| wvs_individual ($K_{max}$=21, N=1000) | **Loss** | **1.36** | 1.50 | 4.10 | **1.48** | 1.74 | 4.35 | **1.42** | 1.57 | 1.76 |
| | **Acc** | **42.6** | 42.1 | 40.4 | **44.3** | 41.1 | 40.6 | **41.7** | 41.6 | 39.4 |
| numbergame_individual ($K_{max}$=25, N=592) | **Loss** | **.639** | .705 | 1.80 | **.621** | .697 | 1.28 | **.618** | .864 | .770 |
| | **Acc** | **70.2** | 64.3 | 65.6 | 70.6 | 69.8 | **71.0** | 69.1 | 62.5 | 67.5 |
| chatbotarena_individual_prefs ($K_{max}$=3, N=725) | **Loss** | **1.43** | 1.62 | 4.94 | **1.34** | 1.47 | 4.39 | **1.39** | 1.76 | 1.77 |
| | **Acc** | 38.6 | 38.0 | **44.6** | 51.4 | **52.0** | 46.3 | 38.9 | 36.0 | **39.5** |
| flight ($K_{max}$=9, N=200) | **Loss** | **1.09** | 1.32 | 4.06 | **1.08** | 1.29 | 2.92 | **1.12** | 1.45 | 1.41 |
| | **Acc** | 39.8 | 41.2 | 40.6 | **43.7** | **43.7** | 40.8 | 33.4 | **42.0** | 40.2 |

| Free-Text Datasets | Metric | ST (ours) | PT | IT | ST (ours) | PT | IT | ST (ours) | PT | IT |
|---|---|---|---|---|---|---|---|---|---|---|
| novacomet_hypothesis ($K_{max}$=11, N=155) | **Loss** | **104** | **104** | 135 | **106** | **106** | 129 | **107** | **106** | 112 |
| novacomet_premise ($K_{max}$=55, N=51) | **Loss** | **27.7** | **28.0** | 35.5 | **28.1** | **27.5** | 38.0 | **27.8** | **27.7** | 28.6 |
| habermas_question ($K_{max}$=29, N=30) | **Loss** | **23.8** | **23.1** | 41.4 | **23.8** | **24.0** | 31.8 | **23.8** | **23.8** | 24.8 |
| habermas_opinions ($K_{max}$=2, N=186) | **Loss** | **930** | **928** | 1070 | **948** | **949** | 1070 | **943** | **944** | 991 |
| habermas_individual ($K_{max}$=2, N=1000) | **Loss** | **164** | **164** | 203 | **168** | **168** | 210 | **166** | 167 | 176 |
| numbergame_perc ($K_{max}$=24, N=182) | **Loss** | **4.23** | **4.22** | 6.68 | **4.22** | **4.24** | 5.61 | **4.24** | 4.43 | 4.41 |
| globaloqa ($K_{max}$=8, N=231) | **Loss** | **14.0** | 14.4 | 21.5 | **14.0** | 14.4 | 20.9 | **14.2** | 14.7 | 15.6 |
| chatbotarena_prompts ($K_{max}$=3, N=988) | **Loss** | 70.2 | **69.4** | 117 | 69.1 | **68.2** | 97.8 | **72.0** | **72.0** | 77.6 |
| chatbotarena_assistant ($K_{max}$=5, N=716) | **Loss** | 127 | **125** | 259 | **124** | **124** | 169 | 134 | **133** | 149 |
| chemistry_esol ($K_{max}$=8, N=59) | **Loss** | 8.94 | **8.37** | 12.9 | **8.07** | 8.47 | 11.8 | **8.28** | 8.51 | 8.55 |
| chemistry_oxidative ($K_{max}$=9, N=101) | **Loss** | **7.57** | **7.58** | 11.6 | **7.64** | 7.84 | 10.2 | **7.64** | 7.72 | 7.84 |

Table 1: In-context steerability on held-out SPECTRUM SUITE-Test. SPECTRUM TUNING generally matches or improves upon the pretrained model performance. Best values (and ties, failing to find a significant difference at $\alpha = .05$) are bolded.

| Expected Calibration Error (ECE, ↓) | gemma-3-12b | | | Qwen3-14B | | | Llama-3.1-8B | | |
|---|---|---|---|---|---|---|---|---|---|
| Multiple-Choice Dataset | ST (ours) | PT | IT | ST (ours) | PT | IT | ST (ours) | PT | IT |
| habermas_individual_categorical | 0.116 | **0.069** | 0.239 | **0.032** | 0.05 | 0.198 | **0.037** | 0.084 | 0.055 |
| wvs_individual | **0.006** | 0.015 | 0.223 | **0.017** | 0.02 | 0.191 | **0.005** | 0.012 | 0.024 |
| numbergame_individual | **0.015** | 0.029 | 0.163 | 0.027 | **0.026** | 0.108 | 0.028 | 0.024 | **0.017** |
| chatbotarena_individual_prefs | **0.020** | 0.041 | 0.194 | 0.048 | **0.046** | 0.189 | **0.046** | 0.075 | 0.049 |
| flight | **0.011** | 0.040 | 0.271 | 0.038 | **0.035** | 0.228 | 0.046 | 0.070 | **0.038** |

Table 2: Calibration on SPECTRUM SUITE-Test, binning label token probabilities every decile for expected calibration error (ECE $= \sum_{b=1}^{B} \frac{n_b}{N} |\text{acc}(b) - \text{conf}(b)|$, where $B = 10$ bins, $n_b$ is the number of samples in bin $b$, $\text{acc}(b)$ is the accuracy in bin $b$, and $\text{conf}(b)$ is the average confidence in bin $b$). SPECTRUM TUNING (ST) usually results in the best calibration (9/15 cases).

as possible within some requirements. Additionally, we evaluate each model under three settings: zero-shot with a task description, three-shot with no task description, and three-shot with a task description (also see App. N). Results can be found in Fig 4. Tasks are the same across models.

**Instruction-tuned models have high validity but low diversity.** IT models produce valid outputs $> 70\%$ of the time, even in the zero-shot setting. However, this comes at the price of diversity, resulting in fewer than 30 unique valid generations in few-shot settings. Yield is even lower in the zero-shot setting—Qwen and Gemma average yields of only 5–6, while Llama averages only 24.

**Pretrained models are more diverse, but rely on few-shot examples for validity.** Pretrained models do not suffer from the same mode collapse, and consistently have higher diversity ($> 40\%$ of valid generations are unique). However, this comes at a trade-off with validity, where their generations are universally less valid than the IT models'. The pretrained models also rely heavily on the few-shot examples to elicit valid generations, achieving a validity of $< 20\%$ in the zero-shot case. However, in the few-shot cases, they have a significantly higher yield than the instruction-tuned models due to their higher coverage of the space.

**SPECTRUM TUNING offers a Pareto improvement on diversity and validity, matching or exceeding pretraining yield.** In eight of nine model/setting comparisons, SPECTRUM TUNING offers either a Pareto or strict improvement over the PT/IT models on validity/diversity. In all eight settings with a Pareto improvement, this also leads to a higher yield—i.e., **for a fixed generation budget, SPECTRUM TUNING generates the most unique valid generations.**

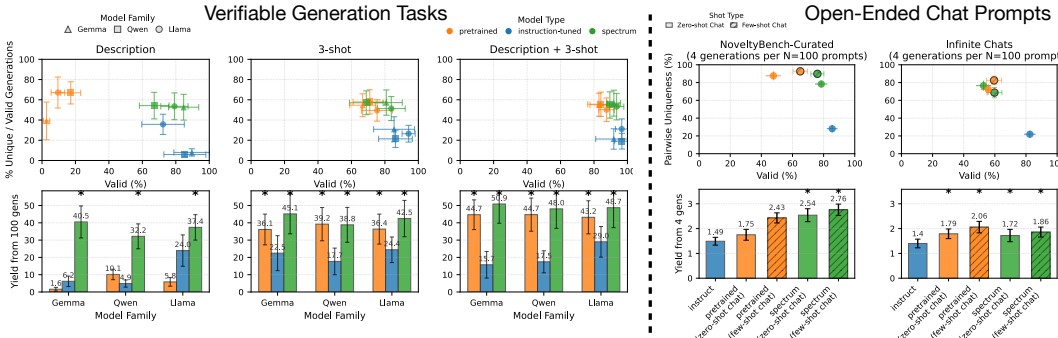

Figure 4: Diversity vs. Validity. Left: Results on 22 verifiable tasks across 100 generations. Right: Human-annotated validity results on two sets of 100 open-ended prompt sets (Gemma). SPECTRUM TUNING generally offers a Pareto improvement on diversity-validity over PT/IT models. In particular, SPECTRUM TUNING increases the yield (# of unique usable generations) in the zero-shot case and on NoveltyBench-Curated. Error bars are 95% confidence intervals over the SEM, and asterisks (∗) show the best in family performance (within 95% confidence).

**SPECTRUM TUNING achieves much higher yield in the zero-shot setting.** Focusing in on the zero-shot setting, SPECTRUM TUNING particularly shines. The IT models are able to follow the description and produce a valid output, but have very low diversity ($\sim 30\%$ for Llama, $\sim 5\%$ for Qwen and Gemma). Meanwhile, the pretrained models are unable to consistently generate valid outputs ($< 20\%$ validity). ST models, however, are able to follow the instructions and produce valid outputs $> 60\%$ of the time while maintaining $50\%$ diversity. This leads to much higher yields compared to PT and IT models (Gemma: 40.5 vs. 6.2; Qwen: 32.2 vs. 10.1, Llama: 37.4 vs. 24.0).

**SPECTRUM TUNING's gains hold across temperature values.** One way to trade-off validity for diversity for a given model is sweeping temperature. To ensure that our results hold across temperatures, we ran the same experiment with $T = [10, 5, 2, 1.5, 1, .9, .7, .5]$. We found that SPECTRUM TUNING A) still expanded the Pareto frontier and B) gave the highest possible yield when choosing an optimal temperature (see App. E for more details).

## 5.1 HUMAN EVAL

We extend the verifiable task experiments with a human evaluation on open-ended chat prompts: NoveltyBench-Curated (100 prompts, Zhang et al. 2025d) and Infinite-Chats-Eval (100 prompts, yet to be published, obtained from the authors). However, SPECTRUM TUNING does not optimize for chat capabilities, but rather for fitting to `description/input/output`. In order to elicit chat capabilities in-context, we try two approaches: zero-shot chat, where we prompt with `description: You are a helpful AI assistant, input: <prompt>`; and few-shot chat, where we utilize the same description and four examples of prompt inputs and chat responses as outputs. Additionally, we use a similar prompt for the pretrained model as a baseline, with the description, a prefix for the prompt of `User:`, and an output prefix of `Assistant:`, zero-shot and with the same four few-shot examples (similar to URIAL, Lin et al. 2023). More details in App. N.

For each prompt, we generate four completions from the model. We recruit annotators to judge whether a given generation is a valid response to the prompt. Each generation is annotated by four annotators, and we count the generation as valid if three of four annotators marked it as valid. Overall, annotators had a 73% pairwise agreement rate. Due to the cost of the evaluation, we only annotate generations for one model family, `gemma-3-12b`. For additional evaluation details, see App. I. For calculating diversity, we follow NoveltyBench's approach and utilize their `deberta-v3-large`-based model for assigning two generations as duplicates. We report the Pairwise Uniqueness %, or the probability that any two valid generations are not considered duplicates, along with yield. Results are in Tab. 4.

**Few-shot pretrained models improve yield over instruct models.** While lagging in validity, pretrained models produce much more diverse responses than their instruct counterparts, and are able to achieve >40% validity from few-shot chat examples, improving yield and offering a strong baseline.

**SPECTRUM TUNING offers a Pareto improvement on diversity/validity and improves yield over baselines on NoveltyBench-Curated.** On NoveltyBench-Curated, our method offers higher validity than the pretrained model, while offering substantially higher diversity than the instruct model. This improvement results in a statistically significant increase in yield over the baselines. On Infinite-Chats, the pretrained models and our models do not perform significantly differently, covering roughly the same space on the Pareto frontier and on yield. While disambiguating the reason for the differing performance may require further investigation, we do note that many of the Infinite-Chat eval prompts have specific requirements, such as "In five words", "In a couple of paragraphs," etc., which our models often fail to adhere to. In contrast, the NoveltyBench-Curated prompts are far more open-ended. It may be that our model performs best at generating shorter outputs, and future work may be needed to enhance precise instruction-following while maintaining diversity. However, on both datasets, the instruct model has significantly lower yield and diversity.

## 6 DISTRIBUTIONAL ALIGNMENT AND PLURALISM

Next, we evaluate our system's ability to steer to match a target distribution. We utilize seven held-out datasets [2] mainly focusing on human response distributions and a synthetic random draws task. We prompt models zero-shot with a description of the setting and a target question. We then calculate the probability of each possible valid output, normalize, and calculate Jensen-Shannon divergence from the target distribution. We also measure coverage, or the total probability mass on the set of valid answers. Results are in Table 3, and takeaways are as follows. (More details in App. O.)

| *Distributional Alignment: JS-Divergence* $\downarrow$ | gemma-3-12b | | | Qwen3-14B | | | Llama-3.1-8B | | |
|---|---|---|---|---|---|---|---|---|---|
| Dataset | ST (ours) | PT | IT | ST (ours) | PT | IT | ST (ours) | PT | IT |
| **Machine Personality Inventory** (N=120, $|Y|$=6) | **0.083** | 0.126 | 0.347 | **0.100** | **0.093** | 0.405 | **0.063** | 0.087 | 0.131 |
| **Rotten Tomatoes** (N=1000, $|Y|$=2) | **0.032** | **0.032** | 0.134 | **0.028** | **0.028** | 0.122 | **0.035** | **0.035** | 0.086 |
| **NYTimes Books** (N=940, $|Y|$=4) | **0.051** | 0.063 | 0.328 | **0.070** | 0.088 | 0.344 | **0.046** | 0.061 | 0.247 |
| **GlobalOQA** (N=1000, $|Y| \leq 6$) | **0.077** | 0.094 | 0.270 | **0.090** | 0.088 | 0.274 | **0.091** | 0.108 | 0.163 |
| **Urn** (N=1000, $|Y| \leq 6$) | **0.021** | 0.071 | 0.185 | **0.051** | 0.059 | 0.198 | **0.032** | 0.124 | 0.086 |
| **Habermas** (N=658, $|Y|$=7) | **0.149** | **0.147** | 0.436 | **0.123** | **0.127** | 0.434 | **0.151** | **0.155** | 0.242 |
| **Number Game** (N=1000, $|Y|$=2) | **0.051** | **0.049** | 0.138 | 0.052 | **0.043** | 0.131 | **0.055** | **0.060** | 0.094 |

Table 3: Distributional alignment results. Instruction-tuning drastically hurts distributional alignment. SPECTRUM TUNING generalizes to unseen tasks and improves or matches distributional alignment compared to the pretrained model. Best result (within 95% statistical significance) in bold. $N$ is the number of distinct instances, $|Y|$ is the number of possible outputs.

**Instruction-tuned models have higher distributional divergence than pretrained models.** In line with prior work (Sorensen et al., 2024b), we find that instruction-tuned models show higher distributional divergence than pretrained models on all tasks. We believe that this is in large part due to their low-entropy, spiky distributions. In other words, for distribution matching, current instruction-tuning categorically hurts performance compared to the pretrained model.

**SPECTRUM TUNING generally improves distributional alignment over pretrained models.** Out of 21 model/dataset comparisons, SPECTRUM TUNING improves distributional alignment in 10 cases, matches PT models in 10 cases, and degrades performance in 1 case. Pretrained models are a strong baseline—the pretraining objective entirely consists of trying to estimate a well-calibrated distribution over the next token. To our knowledge, ours is the *first method to improve distributional alignment on unseen datasets* over pretrained models.

**SPECTRUM TUNING improves coverage of valid answers over pretrained models and roughly matches instruction-tuned models.** For each of the datasets, there is a limited set of valid answers. Pretrained models often struggle to shift their probability mass based on instructions in a zero-shot manner to only cover the valid output distribution, achieving $\sim 50\%$ coverage in our evaluation.

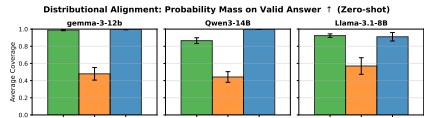

Figure 5: Valid answer coverage ($\uparrow$).

---

[2]Machine Personality Inventory (Jiang et al., 2023), Rotten Tomatoes (u/Business-Platform301, 2024), NYTimes Books (Meister et al., 2024), GlobalOQA (Durmus et al., 2023), Urn (ours, new contribution), Habermas (Tessler et al., 2024), Number Game (Bigelow & Piantadosi, 2016; Tenenbaum, 1999).

| Abl. # | Weight Init | Special Tokens Embedding Init | Train on SPECTRUM SUITE | # Train Seqs | Loss only Outputs | MC Loss (Norm.) | MC Acc (Norm.) | Free-text Loss (Norm.) | Dist. Align. JS-Div. | Yield - Description | Yield - 3-shot | Yield - 3-shot + Description |
|---|---|---|---|---|---|---|---|---|---|---|---|---|
| | | Ablation Components | | | | ICL Steerability | | | Dist. Align. | Valid Output Coverage | | |
| *A - Default: 1) Spectrum Tuning, 2) Pretrained, and 3) Instruction-Tuned* | | | | | | | | | | | | |
| 1 | PT | IT | ✓ | 38.8k | ✓ | **1.00** | **1.00** | **1.00** | .069 | 36.7 | 42.1 | **49.2** |
| 2 | PT | - | × (PT prompt) | - | - | 1.19 | 0.99 | 1.00 | .083 | 5.8 | 37.2 | 44.2 |
| 3 | IT | IT | × (IT prompt) | - | - | 2.62 | 0.98 | 1.30 | .228 | 11.7 | 21.5 | 20.7 |
| *B - Training method ablations: 1) Default; 4) Loss only first output (Instruct-SFT on S-Suite); 5) Loss only last output (Meta-ICL on S-Suite); 6) Loss on all tokens (S-Suite)* | | | | | | | | | | | | |
| 1 | PT | IT | ✓ | 38.8k | ✓ | **1.00** | **1.00** | **1.00** | .069 | 36.7 | 42.1 | **49.2** |
| 4 | PT | IT | ✓ | 38.8k | first only | 1.03 | **1.00** | 1.01 | **.067** | **37.9** | 33.0 | 44.0 |
| 5 | PT | IT | ✓ | 38.8k | last only | 1.02 | 0.99 | **1.00** | .103 | 17.1 | 35.4 | 39.6 |
| 6 | PT | IT | ✓ | 38.8k | × | 1.01 | 0.98 | **1.00** | .075 | 33.0 | 40.6 | 47.1 |
| *C - Data ablation: 7) Train only on capability / knowledge elicitation data, 8) Train on Spectrum Suite, data size matched to capability data* | | | | | | | | | | | | |
| 7 | PT | IT | × (capability data) | 3.9k | ✓ | **1.03** | 0.99 | 1.02 | .111 | 12.7 | 21.2 | 39.5 |
| 8 | PT | IT | ✓ | 3.9k | ✓ | **1.03** | **1.00** | **1.01** | .086 | **21.8** | **35.5** | **40.8** |
| *D - Weight Init Ablation: Spectrum Tuning with 1) Default weight init; 9) PT init, bracket as special token embed, 10) PT init, random special token embed, 11) IT init* | | | | | | | | | | | | |
| 1 | PT | IT | ✓ | 38.8k | ✓ | **1.00** | 1.00 | **1.00** | .069 | 36.7 | 42.1 | 49.2 |
| 9 | PT | <</>> (PT) | ✓ | 38.8k | ✓ | 1.43 | **1.03** | 1.02 | **.063** | 28.0 | 30.0 | 33.0 |
| 10 | PT | Random | ✓ | 38.8k | ✓ | 1.44 | 0.87 | 1.25 | .079 | 21.0 | 21.0 | 26.4 |
| 11 | IT | IT | ✓ | 38.8k | ✓ | 1.08 | 1.02 | 1.05 | .069 | 33.4 | 42.0 | 45.2 |

Table 4: Ablations, averaged across models and tasks. Shaded rows are default Spectrum-Tuned results. We show averaged results for A) the default setup, B) training on SPECTRUM SUITE with different methods, C) training on capability-focused data in place of SPECTRUM SUITE, and D) different model weight initializations. Best result within each ablation is bolded, and second best is underlined. ICL Steerability results are normalized to the default configuration.

In contrast, SPECTRUM TUNING achieves $> 90\%$ coverage, nearly matching the instruction-tuned model coverage (Fig 5).

## 7    ABLATIONS AND GENERAL CAPABILITIES

In Table 4, we ablate parts of SPECTRUM TUNING in order to further disentangle the effect of each component. We report averaged results for all three desiderata across all models and tasks. In A), we see the normalized data from the prior sections, illustrating Spectrum-Tuned models improvements over base and default instruct models.

**SPECTRUM SUITE's selective loss is important for performance on all desiderata.** In B), we hold the Spectrum Tuning data constant, and ablate the training method. We compare against training on the first output only (similar to Instruct-SFT),[3] training on the last output only (similar to MetaICL, Min et al. 2022a), and calculating loss on all tokens, including `description/inputs`. We find that training on the first output only causes a degradation in few-shot learning capabilities (ICL loss, few-shot yield), and training on the last output only causes across the board degradation, especially on zero-shot tasks (distributional alignment, description yield). Training on all tokens (including `description/input`) leads to slight degradations across the board.

**Training on capability-focused data only underperforms training on SPECTRUM SUITE.** We train on a subset of data in the same format as SPECTRUM SUITE, but focused on capability data instead of data requiring steerability (Table 4, C). We find that including the SPECTRUM SUITE data is important for eliciting the desiderata. Finally, we find that D) the default weight initialization (PT model weights, IT special token embeddings) overall elicits the best performance, although initializing the special tokens with bracket token embeddings seems to improve the multiple-choice accuracy and distributional alignment.

While the default recipe offers strong performance, future work could i) further optimize hyperparameters (as we have done limited optimization),[4] ii) reduce reliance on initializing the special tokens from IT models, and iii) probe which data is most important in eliciting gains.

**SPECTRUM TUNING does not harm general model capabilities.** Lastly, we evaluate whether our method affects general model capabilities. While we do not necessarily expect our method to

---

[3]However, we also consider this distinct from traditional instruction-tuning, as the focus is on fitting the data generation task of the description as opposed to generating a helpful chat assistant response.

[4]In fact, after running the main suite of experiments, we suspected that our models were somewhat underfit. We found that simply reducing the batch size resulted in significant gains in distributional alignment and yield (see App. H for more details). We believe that this illustrates exciting opportunities for further optimization and improvements to improve performance—the performance ceiling has not been hit.

improve upon standard evaluations where there is a single correct answer, we want to understand if it degrades performance compared to pretrained models. While we find that Spectrum-Tuned models generally perform worse than instruction-tuned variants (as expected), we find that Spectrum-Tuned models perform similarly to the pretrained models on which they are based. (c.f. App. D.5)

## 8    RELATED WORK

**Diversity, distributional alignment, and steerability.**  Several other works have documented diversity collapse in LLMs (Shumailov et al., 2023; Dohmatob et al., 2024; Yang et al., 2024; Zhang et al., 2024a; Li et al., 2024; West & Potts, 2025), often linking it to alignment (Murthy et al., 2024; Kirk et al., 2024a; 2023) or insufficient training data diversity (Chen et al., 2024). Potential consequences of diversity collapse include reduced creativity, loss of minority perspectives, spread of bias, and overall decline in model utility and trustworthiness (Anderson et al., 2024; Kapania et al., 2024). Distributional alignment has been explored by a few prior works (Meister et al., 2024; Durmus et al., 2023; Sorensen et al., 2024b), but literature here is far less developed. Additionally, other works have focused on measuring steerability to system messages (Lee et al., 2024), persona descriptions (Miehling et al., 2025; Castricato et al., 2024), and values or attributes (Sorensen et al., 2024b; 2025). Our work builds on these directions by generalizing steerability to include any in-context information, including examples, and evaluating on a broader swath of distributions.

**Pluralistic alignment and integrating disagreement into LLMs.**  Many have recently challenged the idea of a single ground truth (Aroyo et al., 2023; Basile et al., 2021; Gordon et al., 2022). Pluralistic alignment (Sorensen et al., 2024b; Kirk et al., 2024b) is concerned with integrating diverse values and perspectives directly into the alignment process. Steerability in particular is related to user fairness and ensuring that AI systems are usable by diverse stakeholders (Alamdari et al., 2024).

**Related Methods**  Zhang et al. (2024a) found that training on samples from diffuse distributions helps LLMs to avoid mode collapse, and served as inspiration for some experiments. SPECTRUM TUNING is similar in spirit, but also includes in-context samples and leverages orders of magnitude more data. Entropy maximization in finetuning can help increase diversity (Li et al., 2025). MetaICL (Min et al., 2022a) uses in-context examples as in our method, but focuses on NLP datasets with a single ground truth and only trains on the last example. Centaur (Binz et al., 2024) similarly modifies cross-entropy loss to only focus on tokens of interest, but focuses on a different data distribution (cognitive-science human experiments). Some very recent works have somewhat improved the diversity/validity Pareto frontier by adding some sort of diversity regularization to preference optimization or RL reward (Lanchantin et al., 2025; Chung et al., 2025; Li et al., 2025). Finally, several recent papers have found that prompting instruct models for multiple samples in-context can help to mitigate mode collapse (Zhang et al., 2025a;b;d).

## 9    DISCUSSION AND CONCLUSION

We have outlined three desiderata for conditional distributional modeling with LLMs: in-context steerability, output space coverage, and distributional alignment, and shown across three model families that current post-training can systematically hurt these properties. These results have implications for user steerability—e.g., when possible, pretrained models may be preferred over instruction-tuned models when steering to a particular user in a well-calibrated way is important.[5] In addition, we have introduced SPECTRUM SUITE and SPECTRUM TUNING, a resource and post-training method for enhancing these desiderata. Models trained with SPECTRUM TUNING usually match or exceed their pretrained counterparts at these properties—to our knowledge, ours is the first method to improve upon pretrained models at distributional alignment or in-context steerability. However, much work remains. Promising directions for future work include 1) exploring which data is most important for eliciting the desiderata; 2) further characterizing why and how instruction-tuning hurts in-context steerability; 3) more work to combine the strengths of instruction-tuned models and SPECTRUM TUNING models (e.g., Zhu et al. 2025);[6] and 4) scaling SPECTRUM TUNING to larger models and more data.

---

[5]However, access to the pretrained model is restricted in many proprietary cases. This illustrates a gap: Can companies offer very steerable and distributionally-aligned models, while maintaining safety constraints?

[6]On the other hand, it is possible that top-1 chat performance and our desiderata are so fundamentally in tension, that we may need to specialize models to either top-1 chat performance or our desiderata, and select the appropriate model for each use case or combine strengths at inference (e.g., Zhu et al. 2025)

## ETHICS STATEMENT

In this paper, we seek to enable AI systems that can work for a variety of perspectives and estimate human preferences and opinions in a well-calibrated manner. We believe that these are net positive developments, allowing AI systems to work properly for more people. Additionally, well-calibrated human preferences may be especially important as AI systems are used agentically - it will be important that an agent have a good model of what the user wants, as opposed to a modal preference. Calibration, where current instruction-tuned systems really struggle, can be especially important for agents to safely act autonomously when they are (properly) very confident about a users' preference, and ask for direction when they are less confident.

With SPECTRUM SUITE, we perform experiments on several datasets which may include personal information such as demographics. However, all included datasets are anonymized, we attempt to use the data only in line with their intended use, and we do not distribute the underlying datasets in SPECTRUM SUITE directly. Instead, we refer people interested in extending our work to the original data sources, and provide only the code to unify the data into the `description/input/output` format. Because of this, we believe that our compilation of SPECTRUM SUITE does not pose an additional privacy risk.

## REPRODUCIBILITY STATEMENT

We have attempted to ensure that every portion of the paper is reproducible, and release code[7] containing: SPECTRUM SUITE construction, including processing and pointers to hydrate each dataset; SPECTRUM TUNING training code; and code for running all evaluations. We also release the weights for all trained SPECTRUM TUNING models.[8] We include additional training details on hardware and hyperparameters used in App. G and additional experimental details in App. M, N, O. In App. P, we show demonstrative example prompts for each test task in SPECTRUM SUITE and include example prompts for remaining tasks in supplementary materials.[9]

## ACKNOWLEDGMENTS

We would like to thank Hannaneh Hajishirzi, Sewon Min, Luke Zettlemoyer, Peter West, and Kshitish Ghate for draft feedback and Christopher Sorensen for help designing the SPECTRUM TUNING logo. This research was supported in part by DARPA under the ITM program (FA8650-23-C-7316) and by the AI 2050 Schmidt Sciences Senior Fellowship.

---

[7]`https://github.com/tsor13/spectrum`

[8]`https://huggingface.co/collections/tsor13/spectrum-68dac670f618224845c0fb7d`

[9]`https://tsor13.github.io/files/spectrumprompts.pdf`

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

# A  LIMITATIONS

We hope that SPECTRUM SUITE can serve as a useful resource for others to evaluate and train models that support better in-context steerability, valid output coverage, and distributional alignment. We also believe that SPECTRUM TUNING serves as a useful step in improving these desiderata. However, our work has several limitations.

*Experiments performed only on ≤14B parameter models.* While we have ensured that results generalize across 3 model families, all models tested are in the 8B–14B parameter range. We have no reason to believe that our findings will not scale to larger model sizes, but this remains to be empirically verified.

*Not optimized for chat.* While most current post-training techniques optimize for (potentially multi-turn) chat, models trained with SPECTRUM TUNING instead focus on the description/input/output framework. While it can be possible to elicit chat-style messages via few-shot examples (see App. N) from ST models, we would expect that instruct models would be better at outputting a single chat response that is preferred by humans. It may be possible to combine the desiderata with a chat-style model, but they may also be fundamentally in tension, requiring distinct models for diversity/coverage and for chat.[5]

*Additional work needed on safety guardrails.* Currently, models trained with SPECTRUM TUNING always attempt to steer to the description and examples, regardless of content. This is, of course, also true of pretrained models, which is one justification for why a model developer may choose to keep certain pretrained models with advanced capabilities unavailable to the public. All of our experiments are with models with public pretrained variants, and we do not believe releasing our SPECTRUM TUNING models enable any fundamentally new capabilities over the pretrained variants, but rather increase alignment with the desiderata. However, if a pretrained model has potentially harmful or dangerous capabilities that a model developer wishes to restrict, SPECTRUM TUNING would need to be modified to adhere to these restrictions. While it is easy to imagine potential extensions to e.g. refuse to produce an output that violates a policy, we leave such exploration to future work.

# B  SUPPLEMENTARY FIGURES

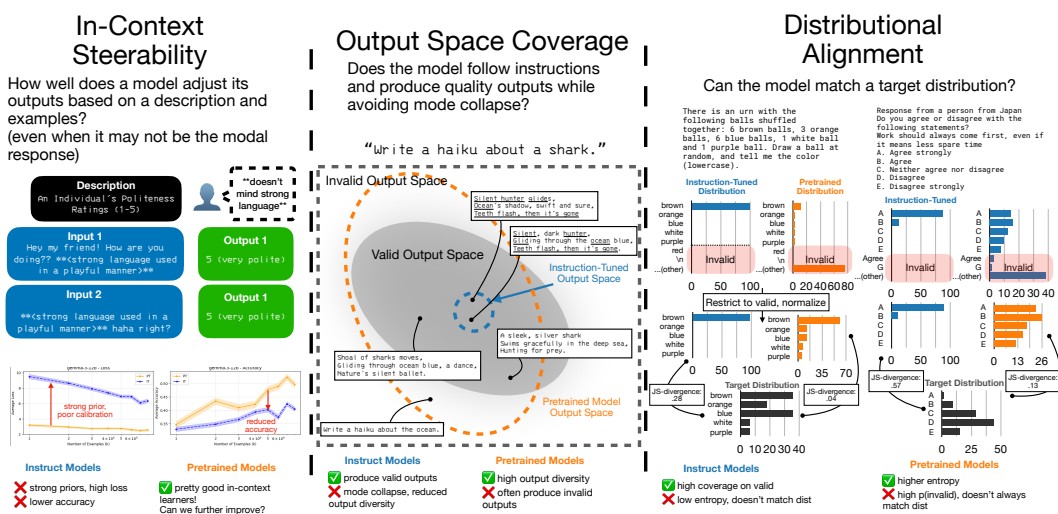

Figure A1: Three desiderata for conditional distributional modeling. Example outputs and data are drawn from google/gemma-3-12b.

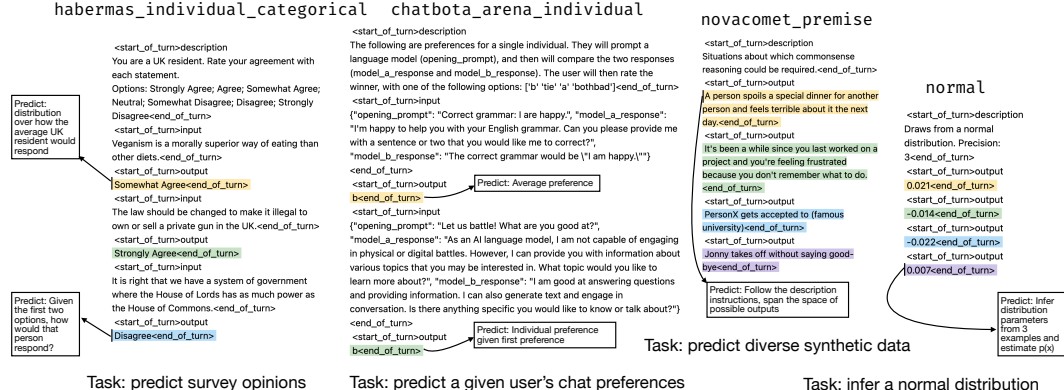

Figure A2: Example tasks from SPECTRUM SUITE in the format used for SPECTRUM TUNING. In our method, we shuffle the data, put it into the above format, and finetune with cross-entropy loss only on the (highlighted) output tokens, including the terminal token.

## C  FREQUENTLY ASKED QUESTIONS, INTUITIONS, AND HYPOTHESES

*Q1: What unifies the three desiderata?*
A1: At first glance the desiderata may not seem very related, but they actually all have something in common - they all have to do with tasks where there is not a canonical, single correct answer. Rather, all three desiderata involve either matching or steering to a broad spectrum of potentially valid answers. This is in contrast with the majority of tasks on which we currently train and evaluate instruction-tuned LLMs.

*Q2: Why does instruction-tuning post-training lead to spiky distributions and mode collapse?*
A2: We have two principal hypotheses for this: 1) the RL objective in RLHF/DPO/GRPO/etc. encourages the model to collapse its distribution to the highest reward output (c.f. West & Potts 2025) and 2) most instruction-tuning training and evaluations focus on tasks with a single verifiable answer. While outside the scope of this work, comparing the desiderata at different stages of instruction-tuning (e.g., during and after Instruct-SFT, during and after RL) would help to elucidate this.[10]

*Q3: It makes sense that SPECTRUM TUNING improves in-context steerability, as it maps easily onto the training data format. However, why does Spectrum Tuning improve diversity and distributional alignment/calibration?*
A3: While we hope to flesh out our understanding of this mechanism in future work, our best intuition is this - It largely has to do with the fact that 1) all training tasks involve interchangeable data and 2) we shuffle the data before training. As a simple example, let us consider the `diffuse_distribution` task: "Output a random country in Asia, chosen completely at random, without replacement." In training, we collect a list of all countries in Asia, shuffle them, and finetune on them as outputs: e.g., "Brunei", "Lebanon", "Singapore", "Laos", "Vietnam", ... An instruction-tuned model will often exhibit mode collapse - outputting the same country each time. Meanwhile, a base model will often output a valid country, but is heavily affected by training data frequency / n-gram statistics. In contrast, in the limit, Spectrum Tuning encourages the model to actually instantiate a uniform distribution over all countries in Asia - increasing the diversity of outputs across many samples. For distributional alignment and calibration, it is a similar story - base models are heavily affected by things like n-gram statistics, instruct models have uncalibrated, spiky distributions. In contrast, Spectrum Tuning in the limit encourages the model to fit the actual described distribution, (partially) overcoming n-gram frequency.

---

[10]For an example of the checkpoint setup one might use, please refer to Bhatia et al. 2025, where they explore the effect of post-training on value drift.

# D SPECTRUM SUITE DATA SOURCES

## D.1 DATA CONSTRUCTION

As SPECTRUM SUITE is the first-such large-scale resource of such subjective datasets requiring steering, it was necessarily constructed in a somewhat ad-hoc manner. However, here we provide some general principles for data that we attempted to source:

1. Any NLP datasets with corresponding annotator IDs, allowing us to link multiple annotations to the same person. We especially sourced from datasets where variation is to be expected, as opposed to be eliminated.
2. Datasets related to opinion modeling or computational democracy;
3. Synthetically-generated NLP datasets;
4. Lists of interchangeable things;
5. Draws from random distributions;
6. Tabular data.

## D.2 DATA SOURCES

Below, we cite all data sources used in SPECTRUM SUITE. Additionally, we include any subtask names along with the number of sequences included in SPECTRUM SUITE. We release the processing code to go from raw data to our `description/input/output` in our github repo (`https://github.com/tsor13/spectrum`).

Note that many data sources have much more additional data that we could utilize (e.g., OpinionQA (Santurkar et al., 2023), Polis (The Computational Democracy Project, 2025), synthetically generated random data). We generally restricted each data source to a maximum of 1-2k sequences to ensure training data diversity, and in all but a couple of cases with very few data instances (e.g. Diffuse Distributions; Zhang et al. 2024b) additionally ensured that the same piece of data was not used in more than one sequence.

## D.3 TRAIN SPLIT

**Ambient Ambiguity Detection** (Liu et al., 2023)

- `ambient_ambiguity_detection` (50 sequences)
- `ambient_annotation_distributions` (50 sequences)
- `ambient_disambiguation` (50 sequences)
- `ambient_interpretation_labels` (50 sequences)
- `ambient_linguist_annotations` (54 sequences)
- `ambient_premise_hypothesis` (50 sequences)

**Social Security Administration Baby Names** (Social Security Administration, 2025)

- `babynames` (500 sequences)

**Base-Refine Synthetic Data Generation** (Zhu et al., 2025)

- `bare_enron` (55 sequences)
- `bare_gsm8k` (108 sequences)
- `bare_hotpot` (50 sequences)
- `bare_lcb` (136 sequences)
- `bare_newsgroups` (60 sequences)
- `bare_pubmed` (46 sequences)

**Draws from a binomial distribution (generated)**

- `binomial` (500 sequences)

**Draws from a shuffled deck of cards (generated)**

- `cards` (100 sequences)

**Draws from a categorical distribution (generated)**

- `categorical` (500 sequences)

**ChangeMyView Reddit** (Kolyada et al., 2020)

- `changemyview_categories` (809 sequences)
- `changemyview_posts` (1159 sequences)

**Draws from a biased coin (generated)**

- `coinflip` (1000 sequences)

**Collective Alignment Dataset** (OpenAI, 2025)

- `collective_alignment_individual` (993 sequences)

**Community Alignment Dataset** (Zhang et al., 2025b)

- `community_alignment_individual_preferences` (770 sequences)
- `community_alignment_individual_reply` (1031 sequences)
- `community_alignment_initial_prompt` (139 sequences)
- `community_alignment_response` (941 sequences)

**DICES dataset** (Aroyo et al., 2023)

- `dices` (295 sequences)

**Diffuse Distributions** (Zhang et al., 2024b)

- `diffuse_distribution` (270 sequences)

**Generative Social choice** (Fish et al., 2025)

- `generativesocialchoice_freetext` (200 sequences)
- `generativesocialchoice_validation` (400 sequences)

**Draws from a geometric distribution (generated)**

- `geometric` (500 sequences)

**Draws from a geometric beta distribution (generated)**

- `geometric_beta` (500 sequences)

**Grade-school math problems (GSM8K)** (Cobbe et al., 2021)

- `gsm8k_answer_from_question` (50 sequences)
- `gsm8k_question` (50 sequences)
- `gsm8k_question_answer` (50 sequences)
- `gsm8k_question_from_answer` (50 sequences)

**Haikus** (Neiman, 2018)

- `haikus` (600 sequences)

**Hatespeech annotations from diverse annotators** (Kumar et al., 2021)

- `hatespeech_individual` (1000 sequences)

**Helpsteer2 Synthetic Chat Preferences** (Wang et al., 2024b)

- `helpsteer` (320 sequences)

**Draws from a hypergeometric distribution, generated** (Wang et al., 2024b)

- `hypergeometric` (500 sequences)

**IssueBench (measuring political leaning of LLMs)** (Röttger et al., 2025)

- `issuebench` (4 sequences)

**Jeopardy! questions and answers** (trexmatt, 2014)

- `jeopardy_answer_prediction` (1000 sequences)
- `jeopardy_question_generation` (1000 sequences)

**Sarcasm detection (multiple annotators)** (Jang & Frassinelli, 2024)

- `lewidi_csc_sarcasm_detection_individual` (872 sequences)

**Irony detection (multiple annotators)** (Casola et al., 2024)

- `lewidi_mp_irony_detection_individual` (475 sequences)

**Paraphrase detection with rationales (multiple annotators)** (Leonardelli et al., 2025)

- `lewidi_par_paraphrase_detection_individual` (80 sequences)
- `lewidi_par_paraphrase_detection_individual_categorical` (80 sequences)

**Entailment (multiple annotators)** (Weber-Genzel et al., 2024)

- `lewidi_varierrnli_nli_detection_individual` (52 sequences)
- `lewidi_varierrnli_nli_detection_individual_categorical` (52 sequences)

**Draws from a multinomial distribution (generated)**

- `multinomial` (500 sequences)

**Draws from a negative binomial distribution (generated)**

- `negative_binomial` (500 sequences)

**Netflix views and rating data** (Netflix, Inc., 2009)

- `netflix_individual_ratings` (1000 sequences)
- `netflix_individual_views` (2000 sequences)

**Draws from a normal distribution (generated)**

- `normal` (1000 sequences)

**OpinionQA: Large-scale opinion survey dataset** (Santurkar et al., 2023)

- `opinionqa_individual` (3000 sequences)
- `opinionqa_questions` (15 sequences)

**Draws from a poisson distribution (generated)**

- `poisson` (500 sequences)

**Polis OpenData: Votes from a digital town hall** (The Computational Democracy Project, 2025)

- `polis_comment` (336 sequences)
- `polis_vote` (7452 sequences)

**Popquorn: Annotator disagreement on 5 NLP tasks, with demographics** (Pei & Jurgens, 2023)

- `popquorn_individual` (400 sequences)
- `popquorn_og_categorical` (80 sequences)

**Prism: World-wide, pluralistic chat preferences** (Kirk et al., 2024b)

- `prism_individual_preferences` (1333 sequences)
- `prism_prompts` (54 sequences)
- `prism_prompts_individual` (1393 sequences)

**Titanic survival prediction: classic machine learning tabular dataset** (mstz, 2023)

- `titanic_all_variables` (14 sequences)
- `titanic_survival_prediction` (14 sequences)

**Value Consistency: Multi-lingual value laden questions** (Moore et al., 2024)

- `valueconsistency` (21 sequences)

**ValuePrism: datasets with moral judgments and relevant values, rights, and duties** (Sorensen et al., 2024a)

- `valueprism_misc` (400 sequences)
- `valueprism_situation` (105 sequences)
- `valueprism_vrd` (500 sequences)
- `valueprism_vrds_noncontextual` (74 sequences)

**Draws from a zipfian distribution (generated)**

- `zipfian` (500 sequences)

### D.4 TEST SPLIT

**ChatbotArena Individual Preferences** (Zheng et al., 2023)

- `chatbotarena_assistant` (928 sequences)
- `chatbotarena_individual_prefs` (1183 sequences)
- `chatbotarena_prompts` (1000 sequences)

**Tabular Chemistry Dataset** (Ramos et al., 2023)

- `chemistry_esol` (310 sequences)

- `chemistry_oxidative` (102 sequences)

**Synthetic Flight Preferences** (Qiu et al., 2025)

- `flight` (200 sequences)

**GlobalOQA: Country-specific Value Surevy Distributions** (Durmus et al., 2023)

- `globaloqa` (274 sequences)

**Habermas Dataset: AI Deliberation with UK residents** (Tessler et al., 2024)

- `habermas_individual` (1996 sequences)
- `habermas_individual_categorical` (2000 sequences)
- `habermas_opinions` (199 sequences)
- `habermas_question` (43 sequences)

**NovaCOMET: Synthetic Commonsense Dataset** (West et al., 2023)

- `novacomet_hypothesis` (170 sequences)
- `novacomet_premise` (68 sequences)

**NumberGame dataset: cognitive science dataset used to study human reasoning under uncertainty** (Bigelow & Piantadosi, 2016)

- `numbergame_individual` (606 sequences)
- `numbergame_perc` (182 sequences)

**World Values Survey, Wave 7: Global survey on human values** (EVS/WVS, 2024)

- `wvs_individual` (2000 sequences)

## D.5  CAPABILITY SPLIT

**AI2 Reasoning Challenge** (Clark et al., 2018)

- `arc` (118 sequences)

**DROP: Reading Comprehension** (Dua et al., 2019)

- `drop` (943 sequences)

**GPQA: Google-Proof QA Benchmark** (Rein et al., 2023)

- `gpqa` (995 sequences)

**Hellaswag: commonsense benchmark** (Zellers et al., 2019)

- `hellaswag` (503 sequences)

**IMDB sentiment classification** (Maas et al., 2011)

- `imdb` (192 sequences)

**MMLU: Massive Multitask Language Understanding Benchmark** (Hendrycks et al., 2021)

- `mmlu` (1000 sequences)

**TruthfulQA: factual questions** (Lin et al., 2022b)

- `truthful_qa` (69 sequences)

**Winogrande: Commonsense sentence completion** (Sakaguchi et al., 2021)

- `winogrande` (127 sequences)

# E    EFFECT OF TEMPERATURE ON DIVERSITY VS. VALIDITY

Temperature can have a major effect on the diversity vs. validity tradeoff when sampling from a model. In §5, we observed that, when sampling across three levels of prompting information and three model families, Spectrum tuning offered a pareto improvement on diversity vs. validity and overall improved yield. However, the question still remains - does Spectrum tuning still offer an improvement, even after sweeping temperature values?

To answer this question, we evaluated the same models under the same setup, but sampled at various temperatures: `[10, 5, 2, 1.5, 1, 0.9, 0.7, 0.5]`. In Figure A3, we plot diversity vs. validity for all three model families, prompting methods, and model types. We find that, in eight of nine settings, Spectrum Tuning expands the diversity / validity Pareto frontier, as compared to using instruction-tuned or pretrained models alone. In addition, Spectrum Tuning models typically expand the Pareto frontier in the high validity region, increasing diversity for a given validity. In line with the temperature=1 results, Spectrum Tuning's gains offer the largest improvement in the lowest information setting, when only a description of the task is provided.

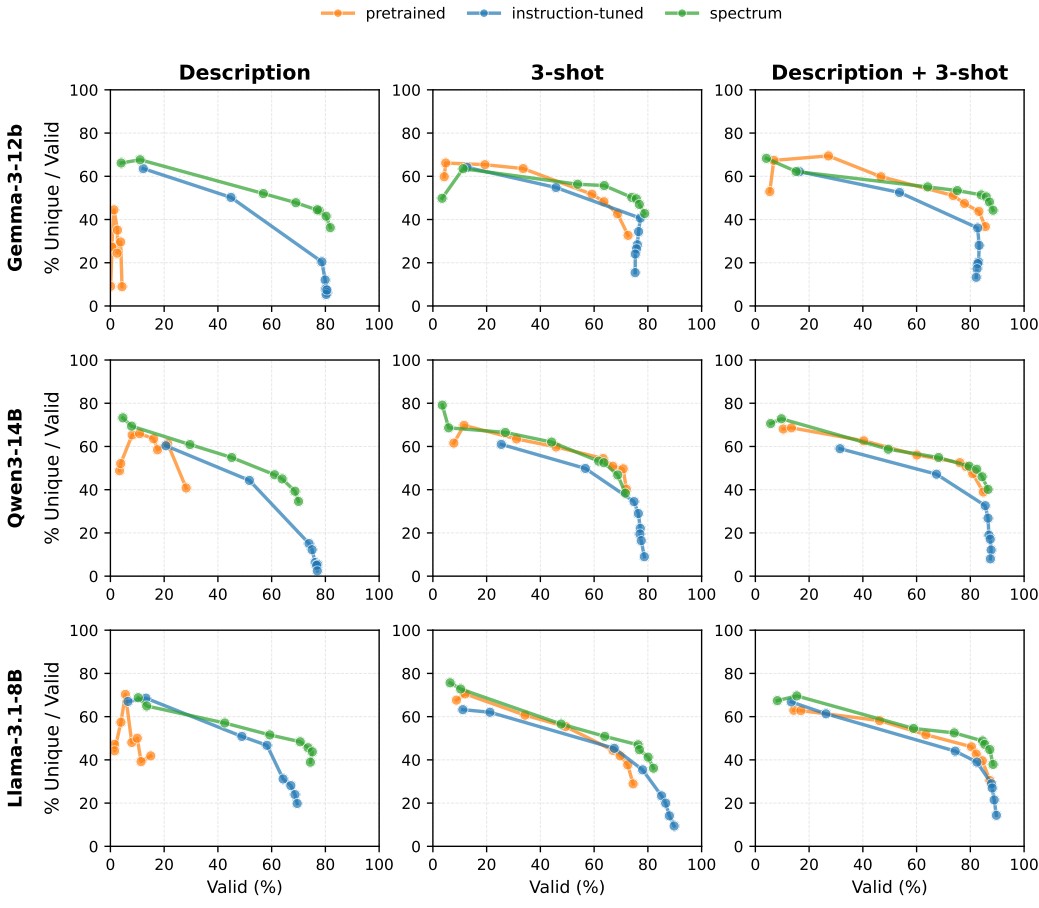

Figure A3: Effect of temperature on diversity and validity. Tested temperatures: `[10, 5, 2, 1.5, 1, 0.9, 0.7, 0.5]`. Lines are connected for temperature in ascending order, with the right-most endpoint being lowest temperature and the left-most endpoint being highest temperature. Spectrum Tuning generally offers a Pareto improvement, especially in the high validity region.

In Figure A4, we also plot the yield for each setting against the temperature. We find that in eight of nine cases, Spectrum Tuning offers the highest possible yield across all models and temperatures - implying that, even if when selecting the optimal temperature for each generation task, we would expect the highest number of distinct valid generations from the Spectrum-Tuned models.

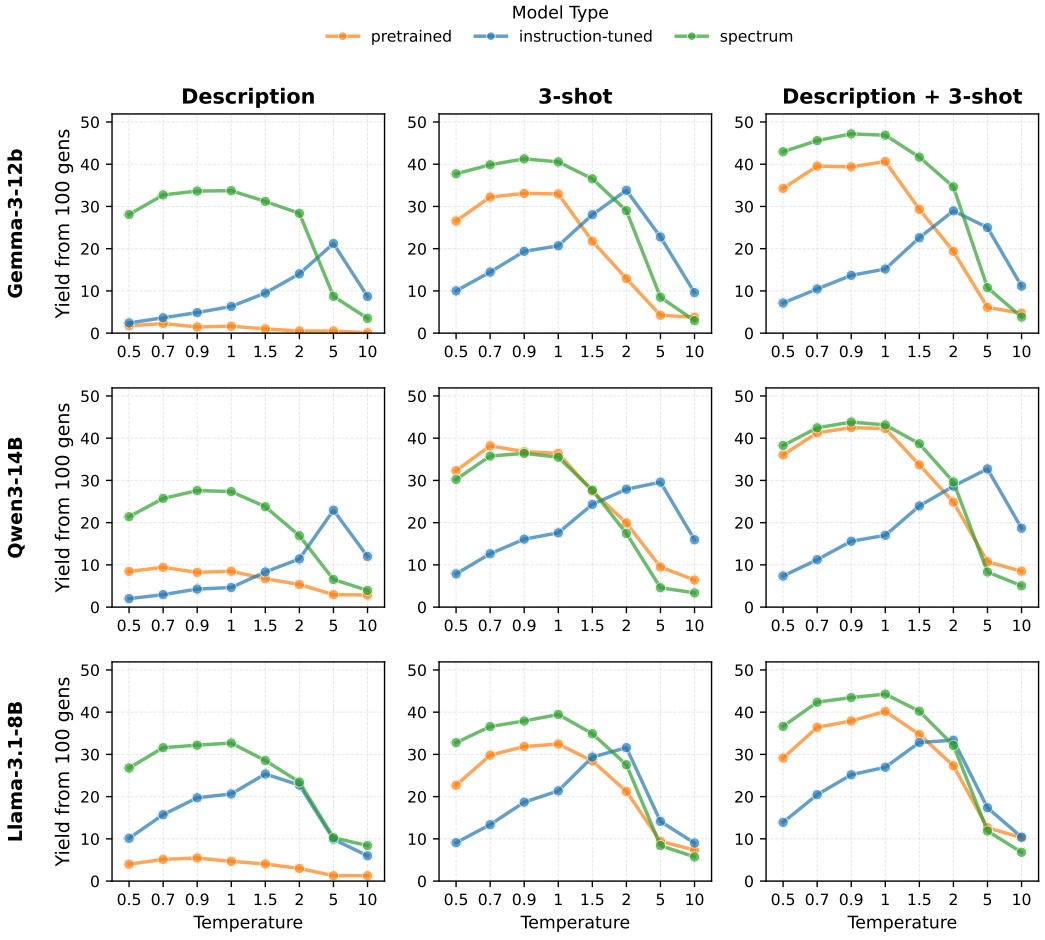

Figure A4: Effect of temperature on yield across each setting. When selecting the optimal temperature for each model, Spectrum Tuning offers the highest overall yield in 8/9 cases (all but Qwen3-14B / 3-shot). Spectrum Tuning also offers the highest yield in most temperature settings $T \leq 2$.

Taken together, we find that the gains from Spectrum Tuning hold even when leaving temperature as a free variable.

## F  GENERAL CAPABILITY PERFORMANCE

We test whether SPECTRUM TUNING affects general model capabilities. While we do not necessarily expect our method to improve upon standard evaluations where there is a single correct answer, we want to understand if it degrades performance compared to pretrained models. We evaluate general knowledge capabilities with Big-Bench Hard (BBH, 3-shot, Suzgun et al. 2023), GPQA (5-shot with chain of thought, Rein et al. 2024), MMLU-Pro (5-shot with chain of thought, Wang et al. 2024a), and TruthfulQA (6-shot, Lin et al. 2022a); instruction following with IFEval (Zeng et al., 2024); and chat ability with AlpacaEval v2 (Dubois et al., 2024). We use the default Olmes hyperparameters for evaluating pretrained models, and Tulu-v3 hyperparameters and task descriptions for evaluating instruction-tuned models (Gu et al., 2025; Lambert et al., 2025). In general, we find that models trained with SPECTRUM TUNING perform similarly to the pretrained models, and in some cases exceed them; however, as expected, instruction-tuned models perform much better, particularly on instruction following and chat tasks.

| Dataset | gemma-3-12b | | | Qwen3-14B | | | Llama-3.1-8B | | |
|---|---|---|---|---|---|---|---|---|---|
| | ST (ours) | PT | IT | ST (ours) | PT | IT | ST (ours) | PT | IT |
| AlpacaEval 2 | 5.935 | 6.897 | 53.846 | 30.421 | 33.541 | 63.123 | 3.642 | 3.579 | 24.641 |
| BBH | 0.738 | 0.727 | 0.821 | 0.786 | 0.789 | 0.770 | 0.641 | 0.631 | 0.722 |
| GPQA | 0.257 | 0.250 | 0.377 | 0.339 | 0.386 | 0.411 | 0.246 | 0.208 | 0.315 |
| IFEval | 0.407 | 0.436 | 0.806 | 0.712 | 0.726 | 0.871 | 0.377 | 0.296 | 0.793 |
| MMLU-Pro | 0.458 | 0.448 | 0.592 | 0.584 | 0.555 | 0.684 | 0.358 | 0.360 | 0.481 |
| TruthfulQA | 0.516 | 0.483 | 0.610 | 0.498 | 0.529 | 0.553 | 0.435 | 0.446 | 0.551 |

Table A1: General Capability Results. *Worst* performance is underlined. SPECTRUM TUNING and pretrained models perform similarly.

## G  TRAINING DETAILS

We lightly tuned hyperparameters by training the `gemma-3-12b` model on a subset of tasks from SPECTRUM SUITE-Train and tracking performance on held-out train tasks. We used the same hyperparameters for Llama and Qwen, performing no additional hyperparameter tuning. Training for all models was done on four 80GB A100 GPUs using DeepSpeed Zero3 (Rajbhandari et al., 2021) and Hugging Face Transformers (Wolf et al., 2020). Training took about 16 hours for the Llama models, 26 hours for the Gemma models, and 30 hours for the Qwen models.

Hyperparameters used:

- `max_length: 1024`
- `per_device_train_batch_size: 1`
- `gradient_accumulation_steps: 512`
- `learning_rate: 3e-6`
- `learning_rate_scheduler: linear_decay`

## H  RESULTS WITH UPDATED HYPERPARAMETERS

After running the main suite of experiments for the paper and experimenting with the models, we had reason to believe that our Spectrum-Tuned models, especially the Qwen and Llama models, were underfit. Note that, for the main set of experiments, we only lightly fit hyperparameters only on the Gemma models using a held-out subset of the train tasks as a validation set, and used the same hyperparameters for Qwen / Llama.

To further explore the effect of updating hyperparameters, we experimented with reducing the batch size in order to take more gradient updates. In the original hyperparameter mix, we use an effective batch size of 2048 (512 gradient steps $\times$ 1 train sequence per device $\times$ 4 GPUs). We halve the batch size three times, and report aggregate results in Table A2.

| Effective Batch Size | ICL Steerability | | | Dist. Align. | Valid Output Coverage | | |
|---|---|---|---|---|---|---|---|
| | MC Loss (Norm.) | MC Acc (Norm.) | Free-text Loss (Norm.) | Dist. Align. JS-Div. | Yield - Description | Yield - 3-shot | Yield - 3-shot + Description |
| 2048 (original hparam) | **1.00** | 1.00 | **1.00** | .069 | 36.7 | 42.1 | 49.2 |
| 1024 | 1.02 | 1.02 | **1.00** | .065 | 43.5 | 44.8 | 51.1 |
| 512 | 1.05 | 1.06 | **1.00** | **.063** | 44.8 | **45.9** | 51.5 |
| 256 | 1.09 | **1.07** | 1.01 | **.063** | **45.9** | 45.7 | **52.0** |

Table A2: Hyperparameter ablations, averaged across models and tasks. Shaded are default SPECTRUM TUNING models. Best result bolded, second best underlined.

We find that 1) decreasing the batch size results in a substantial jump in zero-shot yield, and slight improvements in few-shot yield and distributional alignment. Additionally, decreasing the batch size increases multiple choice accuracy, but at the cost of higher loss on multiple choice answers. All in

all, we think that this illustrates that there are likely to be additional gains from further optimization, and that our initial hyperparameters were likely underfit.

We think that the models trained with effective batch size 512 offer a good tradeoff between ICL steerability, distributional alignment, and valid output coverage, and report their full results in Tables A3-A5 and Figure A5.

| Dataset | Metric | gemma-3-12b | | | Qwen3-14B | | | Llama-3.1-8B | | |
|---|---|---|---|---|---|---|---|---|---|---|
| | | ours | pt | it | ours | pt | it | ours | pt | it |
| **Multiple-Choice Datasets** | | | | | | | | | | |
| | | gemma-3-12b | | | Qwen3-14B | | | Llama-3.1-8B | | |
| **habermas_individual_categorical** (max_k=2, N=1000) | Loss | 3.53 | **2.50** | 10.5 | **2.01** | 2.62 | 9.10 | **2.58** | **2.58** | 2.74 |
| | Acc | **24.0** | 24.4 | 22.4 | 24.9 | 20.3 | 22.0 | **23.2** | 20.2 | 19.0 |
| **wvs_individual** (max_k=21, N=1000) | Loss | **1.36** | 1.50 | 4.10 | **1.38** | 1.74 | 4.35 | **1.42** | 1.57 | 1.76 |
| | Acc | **44.7** | 42.1 | 40.4 | **45.2** | 41.1 | 40.6 | **44.5** | 41.6 | 39.4 |
| **numbergame_individual** (max_k=25, N=592) | Loss | **.665** | .705 | 1.80 | **.617** | .697 | 1.28 | **.611** | .864 | .770 |
| | Acc | **70.2** | 64.3 | 65.6 | **71.2** | 69.8 | 71.0 | **69.2** | 62.5 | 67.5 |
| **chatbotarena_individual_prefs** (max_k=3, N=725) | Loss | 1.52 | 1.62 | 4.94 | **1.35** | 1.47 | 4.39 | **1.43** | 1.76 | 1.77 |
| | Acc | 48.9 | 38.0 | 44.6 | 51.7 | **52.0** | 46.3 | 39.5 | 36.0 | **39.5** |
| **flight** (max_k=9, N=200) | Loss | **1.11** | 1.32 | 4.06 | **1.09** | 1.29 | 2.92 | **1.09** | 1.45 | 1.41 |
| | Acc | 41.0 | 41.2 | 40.6 | 43.1 | **43.7** | 40.8 | 40.9 | **42.0** | 40.2 |
| **Free-text Datasets** | | | | | | | | | | |
| | | gemma-3-12b | | | Qwen3-14B | | | Llama-3.1-8B | | |
| **novacomet_hypothesis** (max_k=11, N=155) | Loss | 105 | **104** | 135 | 107 | **106** | 129 | 110 | **106** | 112 |
| **novacomet_premise** (max_k=55, N=51) | Loss | **27.7** | 28.0 | 35.5 | 27.7 | **27.5** | 38.0 | 27.9 | **27.7** | 28.6 |
| **habermas_question** (max_k=29, N=30) | Loss | 23.9 | **23.1** | 41.4 | **23.8** | 24.0 | 31.8 | **23.8** | **23.8** | 24.8 |
| **habermas_opinions** (max_k=2, N=186) | Loss | **927** | 928 | 1070 | **947** | 949 | 1070 | **944** | **944** | 991 |
| **habermas_individual** (max_k=2, N=1000) | Loss | **164** | **164** | 203 | **167** | 168 | 210 | 166 | **167** | 176 |
| **numbergame_perc** (max_k=24, N=182) | Loss | 4.26 | **4.22** | 6.68 | **4.13** | 4.24 | 5.61 | **4.31** | 4.43 | 4.41 |
| **globaloqa** (max_k=8, N=231) | Loss | 14.2 | **14.4** | 21.5 | **14.0** | 14.4 | 20.9 | 14.5 | **14.7** | 15.6 |
| **chatbotarena_prompts** (max_k=3, N=988) | Loss | 69.8 | **69.4** | 117 | **67.9** | 68.2 | 97.8 | **72.0** | **72.0** | 77.6 |
| **chatbotarena_assistant** (max_k=5, N=716) | Loss | 127 | **125** | 259 | **124** | **124** | 169 | 136 | **133** | 149 |
| **chemistry_esol** (max_k=8, N=59) | Loss | 8.45 | **8.37** | 12.9 | **8.45** | 8.47 | 11.8 | **8.30** | 8.51 | 8.55 |
| **chemistry_oxidative** (max_k=9, N=101) | Loss | 7.57 | **7.58** | 11.6 | **7.57** | 7.84 | 10.2 | **7.68** | 7.72 | 7.84 |

Table A3: In-context steerability results on models trained with an effective batch size of 512.

| Dataset | gemma-3-12b | | | Qwen3-14B | | | Llama-3.1-8B | | |
|---|---|---|---|---|---|---|---|---|---|
| | ours | pt | it | ours | pt | it | ours | pt | it |
| **habermas_individual_categorical** | 0.13 | **0.069** | 0.239 | **0.049** | 0.05 | 0.198 | 0.108 | 0.084 | **0.055** |
| **wvs_individual** | **0.007** | 0.015 | 0.223 | **0.007** | 0.02 | 0.191 | **0.005** | 0.012 | 0.024 |
| **numbergame_individual** | **0.019** | 0.029 | 0.163 | 0.037 | **0.026** | 0.108 | 0.027 | 0.024 | **0.017** |
| **chatbotarena_individual_prefs** | **0.02** | 0.041 | 0.194 | 0.056 | **0.046** | 0.189 | 0.062 | 0.075 | **0.049** |
| **flight** | **0.019** | 0.04 | 0.271 | 0.055 | **0.035** | 0.228 | **0.03** | 0.07 | 0.038 |

Table A4: Calibration for models trained with an effective batch size of 512.

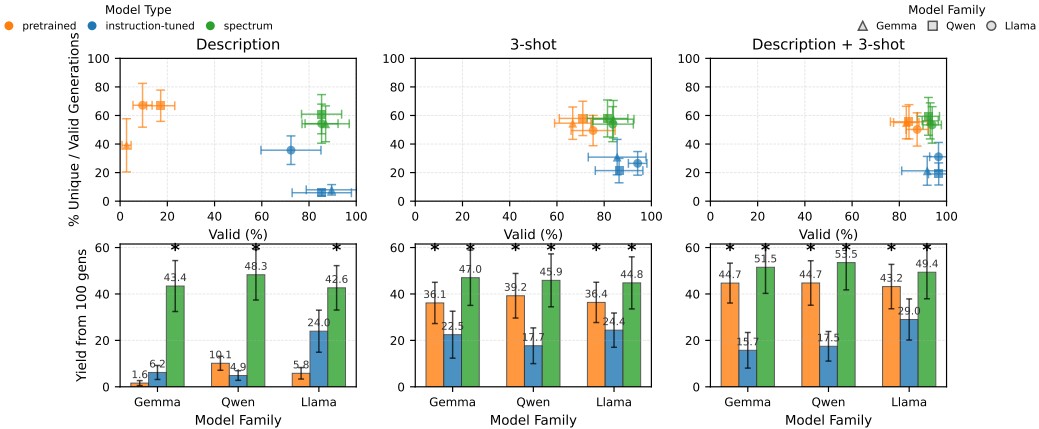

Figure A5: Diversity vs. validity on verifiable tasks for models trained with an effective batch size of 512.

| Dataset | Metric | gemma-3-12b | | | Qwen3-14B | | | Llama-3.1-8B | | |
|---|---|---|---|---|---|---|---|---|---|---|
| | | ours | pt | it | ours | pt | it | ours | pt | it |
| mpi | JS-Div | **.101** | .126 | .347 | **.107** | .0928 | .405 | **.0489** | .0874 | .131 |
| rotten_tomatoes | JS-Div | **.0227** | .0323 | .134 | .0341 | **.0283** | .122 | **.0245** | .0354 | .0859 |
| nytimes | JS-Div | **.0547** | .0628 | .328 | **.0453** | .0876 | .344 | **.0655** | .0613 | .247 |
| global_oqa | JS-Div | **.0678** | .0936 | .270 | **.0749** | .0878 | .274 | **.0828** | .108 | .163 |
| urn | JS-Div | **.0136** | .0713 | .185 | **.0186** | .0592 | .198 | **.0186** | .124 | .0865 |
| habermas | JS-Div | **.142** | .147 | .436 | **.125** | .127 | .434 | **.129** | .155 | .242 |
| numbergame | JS-Div | .0663 | **.0488** | .138 | **.0440** | .0428 | .131 | **.0423** | .0600 | .0943 |

Table A5: Distributional alignment for models trained with an effective batch size of 512.

## I   HUMAN EVALUATION

We conducted a large-scale human annotation study to evaluate the validity and quality of outputs from different model configurations. The study used a pairwise comparison design where annotators evaluated outputs from two models simultaneously for the same prompts. We recruited 245 U.S.-based English speaking annotators who had submitted at least 1000 prior tasks with an approval rating of at least 95% through Prolific and collected a total of 2,400 annotations. Our task took about 30 minutes and we paid at least 7.5 USD for an average of at least 15 USD an hour.

Specifically, we sampled 100 prompts from two evaluation datasets, a curated prompt set and infinite-chats-eval, and collected human judgments for each. Our experimental design compared three model configurations (baseline instruction-tuned, our approach, and pretrained) in both zero-shot and few-shot settings. Each unique combination of (prompt, model pair) was evaluated by two independent annotators, resulting in 200 annotation instances per model pair per dataset.

**Annotation Interface and Procedure**   Participants accessed the annotation task through a web-based interface. First, participants were asked to thoroughly read through the comprehensive annotation guidelines with examples of valid and invalid responses (See Figure A6 and Figure A7). For each annotation instance, annotators were presented with a prompt and four generations from each of two models (labeled Model A and Model B). The model identities and presentation order were randomized to prevent systematic bias. The interface displayed the outputs side-by-side to facilitate direct comparison (See Figure A8 for the user interface and questions).

For each task, annotators made three types of judgments:

- **Validity Assessment:** Annotators independently marked each of the eight generations (4 per model) as either valid or invalid. We provided detailed guidelines defining validity as responses that directly address the prompt, follow all specified requirements, stay on-topic throughout, and contain factually reasonable content. Invalid responses included those that refuse to answer, violate format requirements, trail off into unrelated content, or contain significant errors.
- **Diversity Comparison:** Annotators assessed which model's set of four outputs exhibited greater diversity, with options for Model A, Model B, or "about the same."
- **Overall Quality Judgment:** Independent of diversity, annotators selected which model's outputs were better overall, again with options for either model or "about the same."

To ensure annotation quality, we implemented several measures: (1) Comprehensive annotation guidelines with examples of valid and invalid responses, (2) Tracking of time spent per annotation, and (3) Post-annotation feedback collection to identify any systematic issues.

**Inter-Annotator Agreement**   Inter-annotator agreement for validity judgments showed 76.5% pairwise percentage agreement, with Cohen's $\kappa = 0.441$, indicating moderate agreement. For the subjective diversity and quality assessments, agreement rates were lower (diversity: 38.8%, quality: 41.7%), as expected given the more nuanced nature of these judgments.

## J   LLM USAGE DESCRIPTION

In preparation of this research and manuscript, LLMs were used for:

- Implementing code for experiments and analysis based on detailed author descriptions. All LLM code was inspected by the authors for correctness.
- Formatting for tables, latex, and bibtex citation for non-traditional sources (e.g., urls).
- Draft critique by pointing out typos and potentially confusing wording in the draft.

However, all research ideation and writing was performed solely by the authors, and we claim full responsibility for the contents of this paper.

# Annotation Guidelines

## 📋 Your Task

You will evaluate model outputs for prompts that can have multiple valid answers. For each prompt, you'll see outputs from two different models (A and B).

## ✅ What You'll Annotate

- **Validity:** For each individual output, mark whether it's a valid/correct response to the prompt
- **Diversity:** Compare the variety of responses between the two models
- **Overall Quality:** Judge which model's outputs are better overall

## 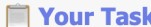 More Guidelines on Validity (Important!)

**A response is VALID if it:**

- Directly addresses what the prompt asks for
- Follows ALL specific requirements in the prompt (length, format, style, etc.)
- Stays on topic throughout the entire response
- Is factually reasonable (doesn't contain obvious falsehoods)

**A response is INVALID if it:**

- Ignores or refuses to answer the prompt
- Violates specific requirements (e.g., prompt asks for "a few sentences" but gives only one word)
- Starts correctly but then trails off into unrelated content
- Contains significant errors or nonsensical content

> ⚠️ **Special Note on Cut-off Responses:** Some responses may be cut off mid-sentence due to length limits (512 tokens). If a response is clearly incomplete but what's there is valid and on-topic, consider it **VALID**. Only mark as invalid if the existing content itself is problematic.

Figure A6: Guidelines for human annotation

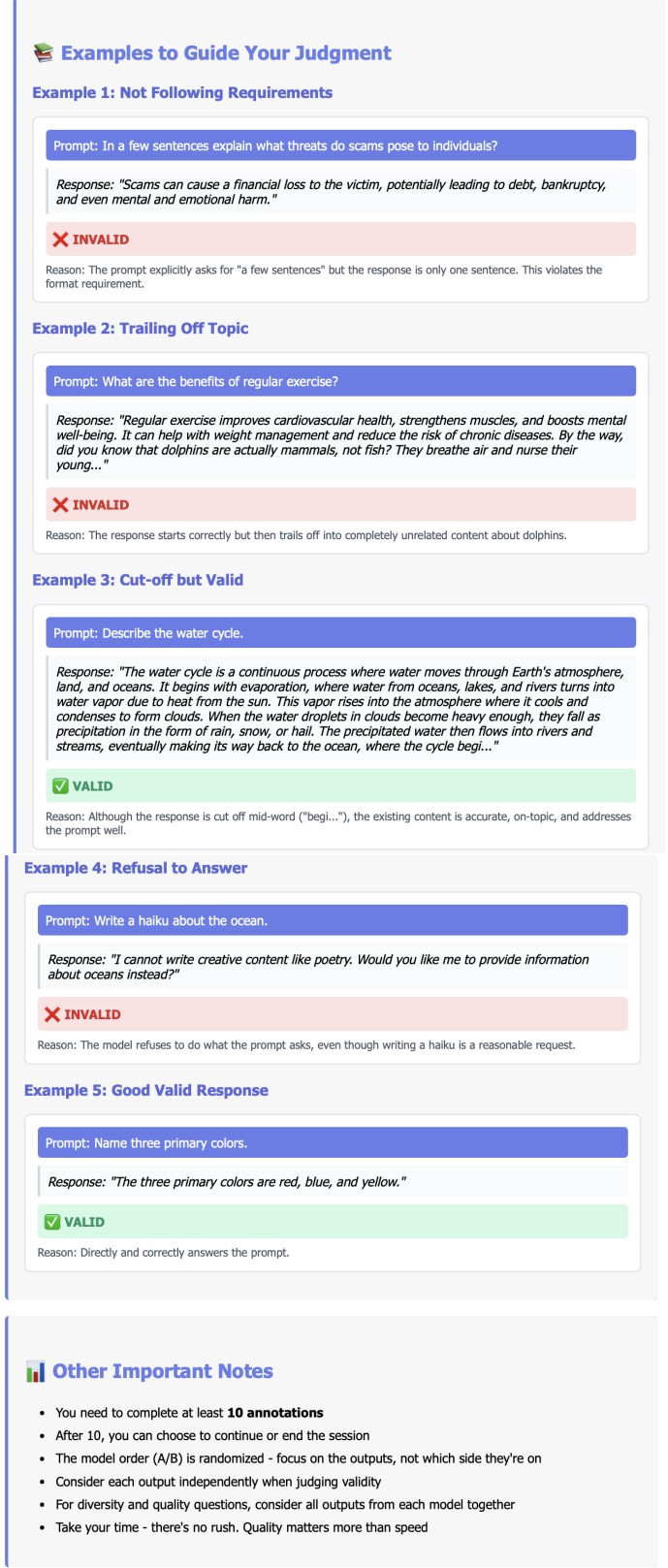

Figure A7: Guideline examples for human annotation

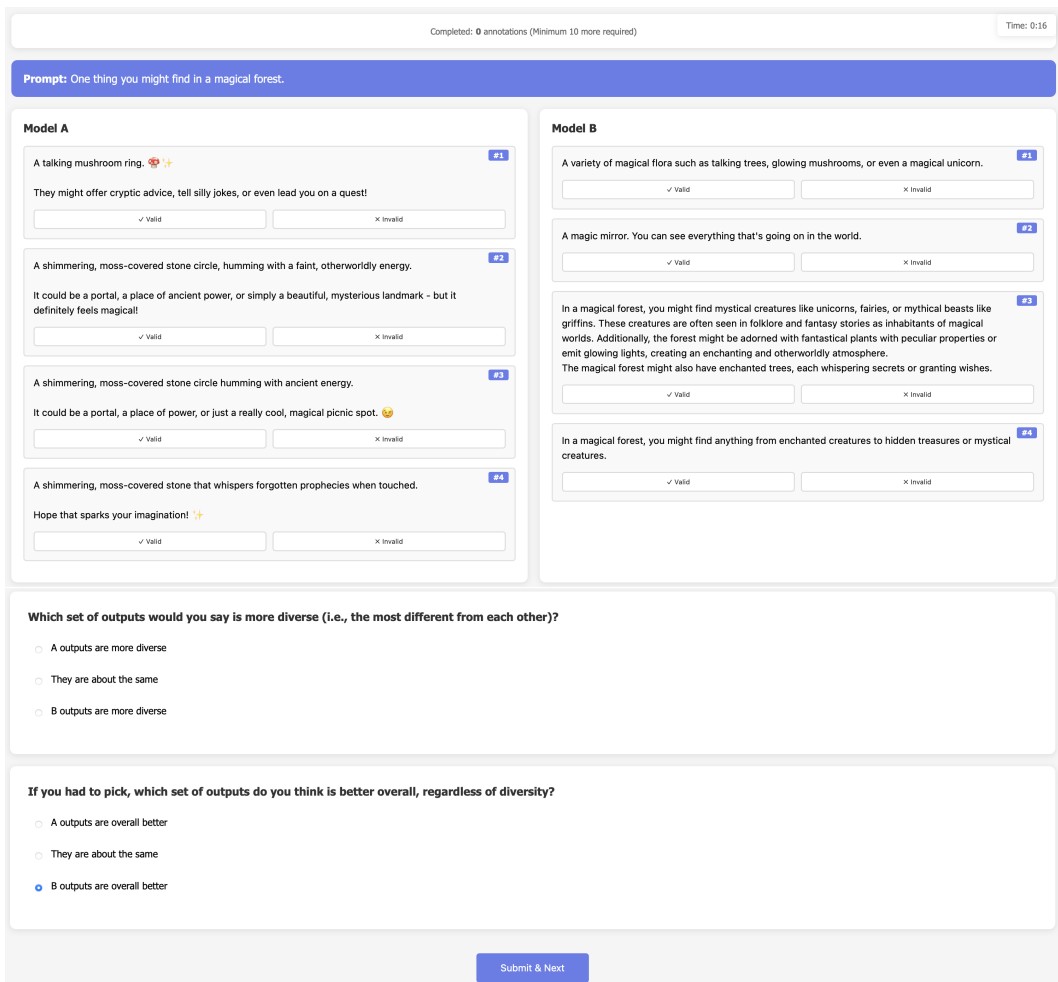

Figure A8: User Interface for Annotation

# K  SUPPLEMENTARY FIGURES

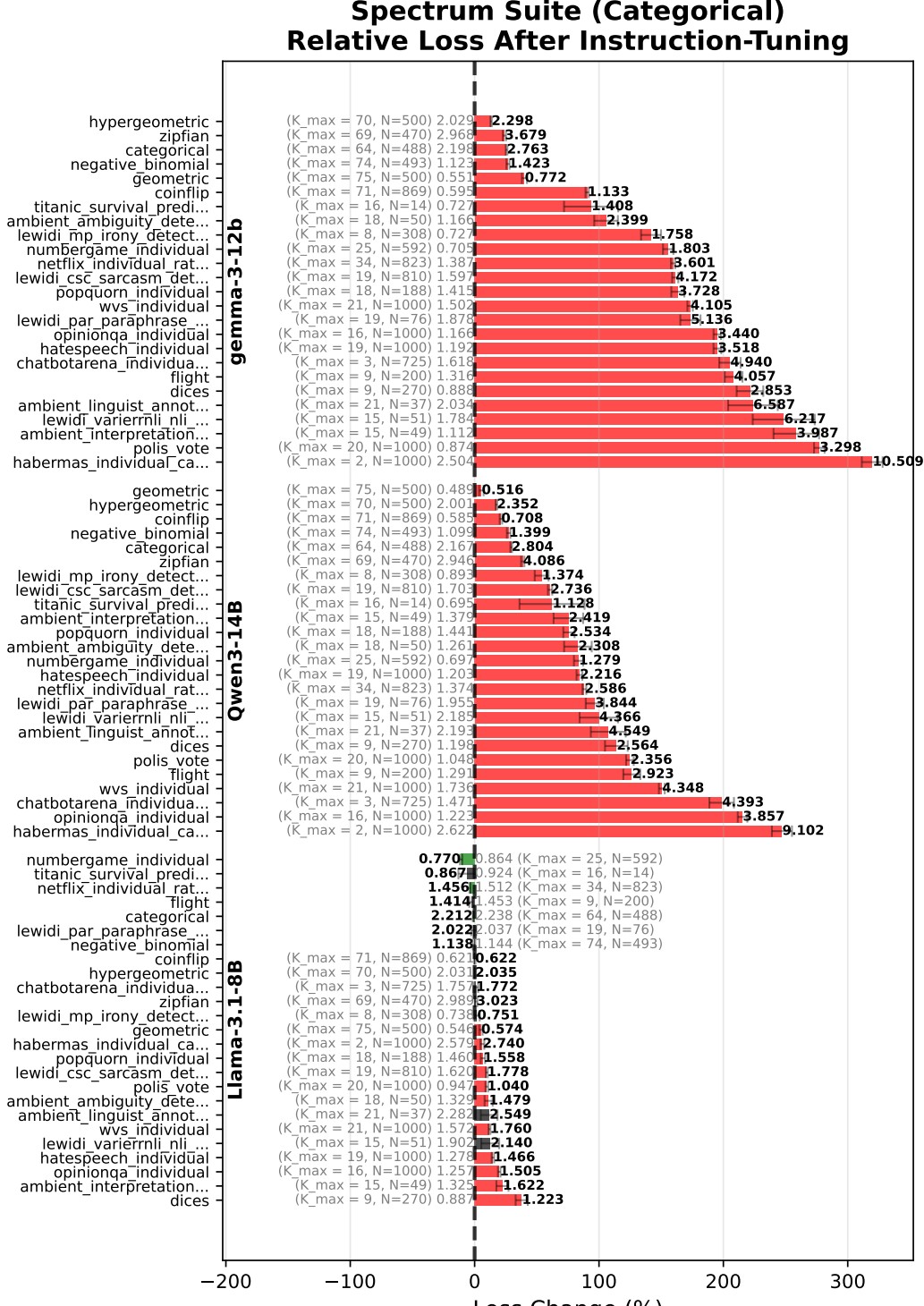

Figure A9: SPECTRUM SUITE categorical loss after instruction-tuning

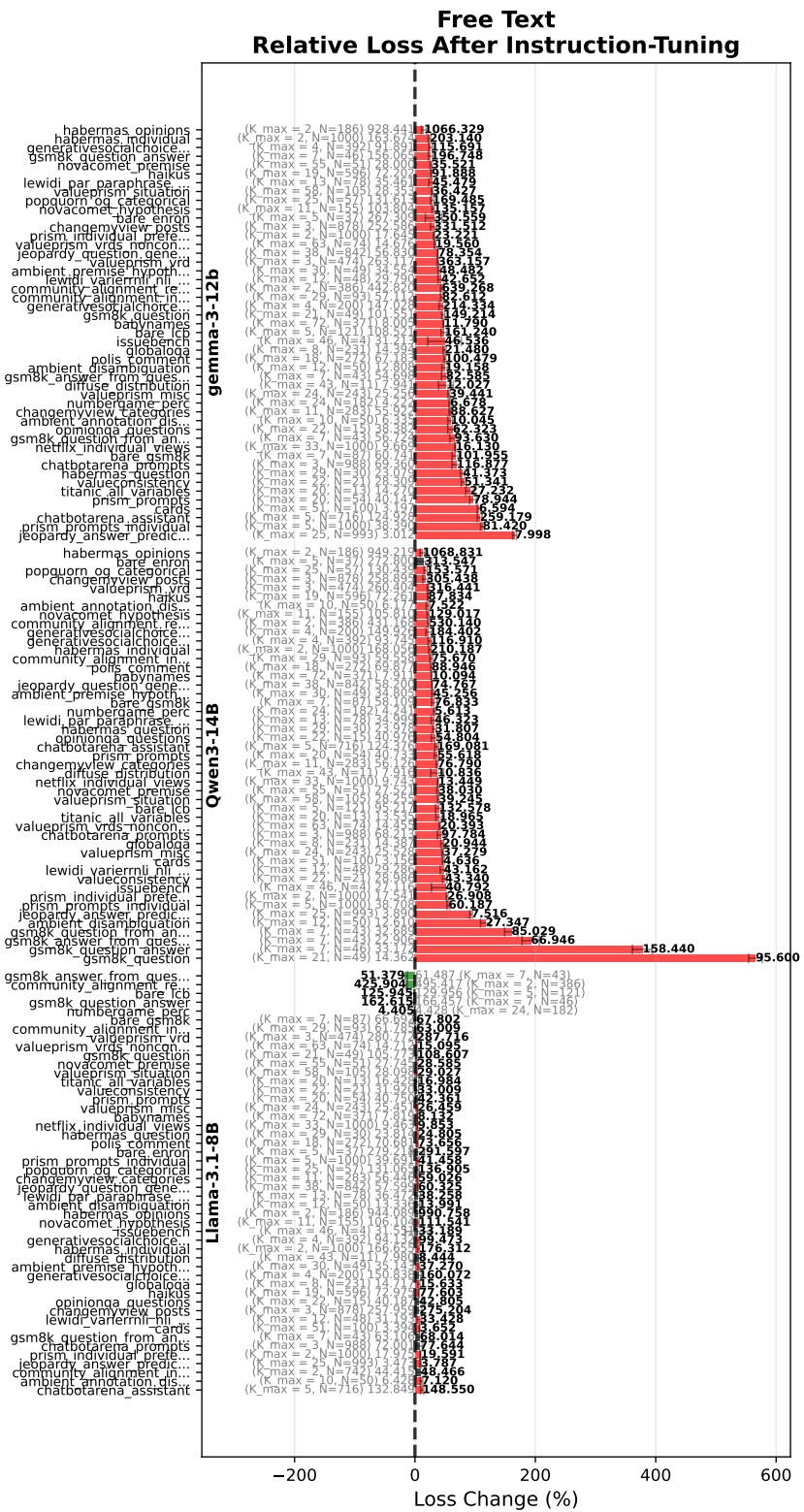

Figure A10: SPECTRUM SUITE free-text loss after instruction-tuning

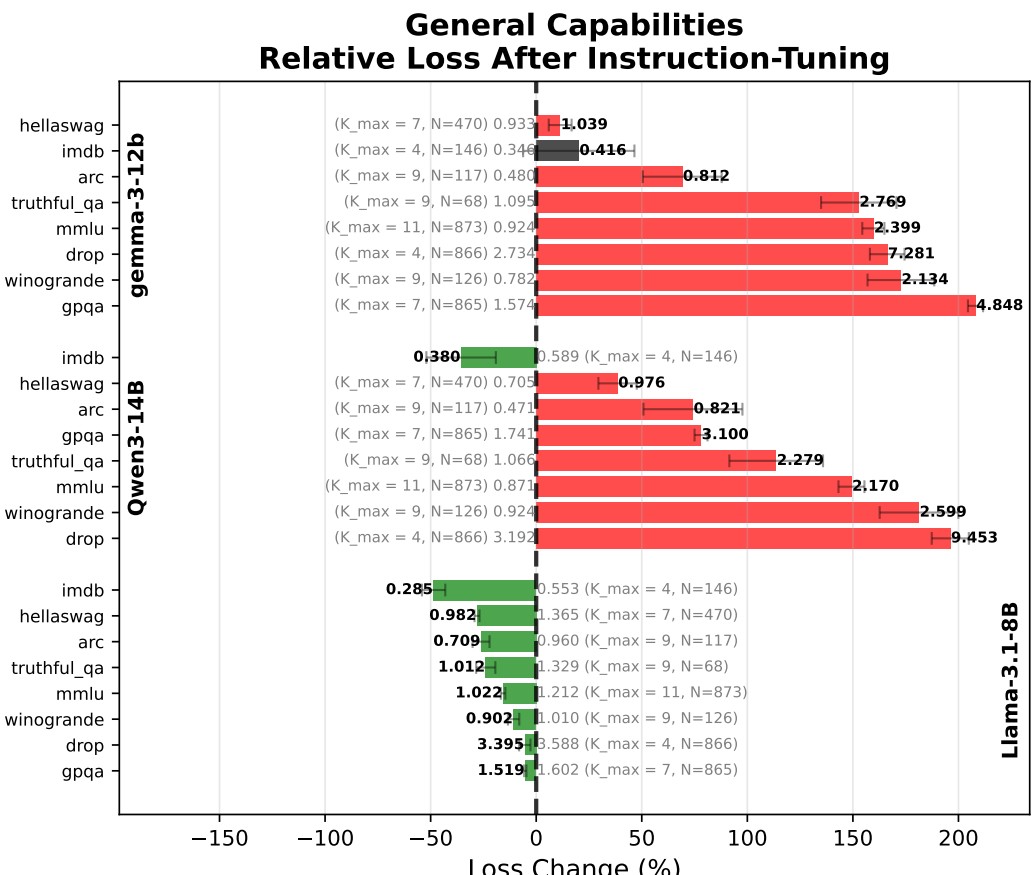

Figure A11: SPECTRUM SUITE general capability loss after instruction-tuning

## L    SPECTRUM TUNING TEMPLATES

For all templates, loss is calculated on the highlighted output tokens.

**gemma-3** (w/ inputs)

```
<start_of_turn>description
DESCRIPTION TEXT<end_of_turn>
<start_of_turn>input
INPUT 1 TEXT<end_of_turn>
<start_of_turn>output
OUTPUT 1 TEXT<end_of_turn>
<start_of_turn>input
INPUT 2 TEXT<end_of_turn>
<start_of_turn>output
OUTPUT 2 TEXT<end_of_turn>
<start_of_turn>input
INPUT 3 TEXT<end_of_turn>
<start_of_turn>output
OUTPUT 3 TEXT<end_of_turn>
...
```

**gemma-3** (w/out inputs)

```
<start_of_turn>description
DESCRIPTION TEXT<end_of_turn>
<start_of_turn>output
OUTPUT 1 TEXT<end_of_turn>
<start_of_turn>input
OUTPUT 2 TEXT<end_of_turn>
<start_of_turn>input
OUTPUT 3 TEXT<end_of_turn>
...
```

**Qwen3** (w/ inputs)

```
<|im_start|>description
DESCRIPTION TEXT<|im_end|>
<|im_start|>input
INPUT 1 TEXT<|im_end|>
<|im_start|>output
OUTPUT 1 TEXT<|im_end|>
<|im_start|>input
INPUT 2 TEXT<|im_end|>
<|im_start|>output
OUTPUT 2 TEXT<|im_end|>
<|im_start|>input
INPUT 3 TEXT<|im_end|>
<|im_start|>output
OUTPUT 3 TEXT<|im_end|>
...
```

**Qwen3** (w/out inputs)

```
<|im_start|>description
DESCRIPTION TEXT<|im_end|>
<|im_start|>output
OUTPUT 1 TEXT<|im_end|>
<|im_start|>output
OUTPUT 2 TEXT<|im_end|>
<|im_start|>output
OUTPUT 3 TEXT<|im_end|>
...
```

**`Llama-3.1`** (w/ inputs)

```
<|start_header_id|>description<|end_header_id|>

DESCRIPTION TEXT<|eot_id|><|start_header_id|>input<|end_header_id|>

INPUT 1 TEXT<|eot_id|><|start_header_id|>output<|end_header_id|>

OUTPUT 1 TEXT<|eot_id|><|start_header_id|>input<|end_header_id|>

INPUT 2 TEXT<|eot_id|><|start_header_id|>output<|end_header_id|>

OUTPUT 2 TEXT<|eot_id|><|start_header_id|>input<|end_header_id|>

INPUT 3 TEXT<|eot_id|><|start_header_id|>output<|end_header_id|>

OUTPUT 3 TEXT<|eot_id|>...
```

**`Llama-3.1`** (w/out inputs)

```
<|start_header_id|>description<|end_header_id|>

DESCRIPTION TEXT<|eot_id|><|start_header_id|>output<|end_header_id|>

OUTPUT 1 TEXT<|eot_id|><|start_header_id|>output<|end_header_id|>

OUTPUT 2 TEXT<|eot_id|><|start_header_id|>output<|end_header_id|>

OUTPUT 3 TEXT<|eot_id|>...
```

## M    PRETRAINED / INSTRUCTION-TUNED ICL TEMPLATES

**Pretrained Template** (w/ inputs)
Note that each output ends with two newlines to ensure a terminal token (coloring not visible).

```
Description: DESCRIPTION TEXT

Input: INPUT 1 TEXT

Output: OUTPUT 1 TEXT

Input: INPUT 2 TEXT

Output: OUTPUT 2 TEXT

Input: INPUT 3 TEXT

Output: OUTPUT 3 TEXT

...
```

**Pretrained Template** (w/out inputs)
Note that each output ends with two newlines to ensure a terminal token (coloring not visible).

```
Description: DESCRIPTION TEXT

Output: OUTPUT 1 TEXT

Output: OUTPUT 2 TEXT

Output: OUTPUT 3 TEXT

...
```

**Simple Instruct Template**

`Qwen3` (task w/inputs)

```
<|im_start|>system
DESCRIPTION TEXT<|im_end|>
<|im_start|>user
INPUT 1 TEXT<|im_end|>
<|im_start|>assistant
<think>

</think>

OUTPUT 1 TEXT<|im_end|>
<|im_start|>user
INPUT 2 TEXT<|im_end|>
<|im_start|>assistant
<think>

</think>

OUTPUT 2 TEXT<|im_end|>
<|im_start|>user
INPUT 3 TEXT<|im_end|>
<|im_start|>assistant
<think>

</think>

OUTPUT 3 TEXT<|im_end|>
```

`Qwen3` (task w/out inputs)

```
<|im_start|>system
DESCRIPTION TEXT<|im_end|>
<|im_start|>user
Generate<|im_end|>
<|im_start|>assistant
<think>

</think>

OUTPUT 1 TEXT<|im_end|>
<|im_start|>user
Generate<|im_end|>
<|im_start|>assistant
<think>

</think>

OUTPUT 2 TEXT<|im_end|>
<|im_start|>user
Generate<|im_end|>
<|im_start|>assistant
<think>

</think>

OUTPUT 3 TEXT<|im_end|>
```

`gemma-3` (task w/inputs)

```
<start_of_turn>user
DESCRIPTION TEXT

INPUT 1 TEXT<end_of_turn>
```

```
<start_of_turn>model
OUTPUT 1 TEXT<end_of_turn>
<start_of_turn>user
INPUT 2 TEXT<end_of_turn>
<start_of_turn>model
OUTPUT 2 TEXT<end_of_turn>
<start_of_turn>user
INPUT 3 TEXT<end_of_turn>
<start_of_turn>model
OUTPUT 3 TEXT<end_of_turn>
```

gemma-3 (task w/out inputs)

```
<start_of_turn>user
DESCRIPTION TEXT

Generate<end_of_turn>
<start_of_turn>model
OUTPUT 1 TEXT<end_of_turn>
<start_of_turn>user
Generate<end_of_turn>
<start_of_turn>model
OUTPUT 2 TEXT<end_of_turn>
<start_of_turn>user
Generate<end_of_turn>
<start_of_turn>model
OUTPUT 3 TEXT<end_of_turn>
```

Llama-3.1 (task w/inputs)

```
<|begin_of_text|><|start_header_id|>system<|end_header_id|>

Cutting Knowledge Date: December 2023
Today Date: DD MM YYYY

DESCRIPTION TEXT<|eot_id|><|start_header_id|>user<|end_header_id|>

INPUT 1 TEXT<|eot_id|><|start_header_id|>assistant<|end_header_id|>

OUTPUT 1 TEXT<|eot_id|><|start_header_id|>user<|end_header_id|>

INPUT 2 TEXT<|eot_id|><|start_header_id|>assistant<|end_header_id|>

OUTPUT 2 TEXT<|eot_id|><|start_header_id|>user<|end_header_id|>

INPUT 3 TEXT<|eot_id|><|start_header_id|>assistant<|end_header_id|>

OUTPUT 3 TEXT<|eot_id|>
```

Llama-3.1 (task w/out inputs)

```
<|begin_of_text|><|start_header_id|>system<|end_header_id|>

Cutting Knowledge Date: December 2023
Today Date: 26 Jul 2024

DESCRIPTION TEXT<|eot_id|><|start_header_id|>user<|end_header_id|>

Generate<|eot_id|><|start_header_id|>assistant<|end_header_id|>

OUTPUT 1 TEXT<|eot_id|><|start_header_id|>user<|end_header_id|>

Generate<|eot_id|><|start_header_id|>assistant<|end_header_id|>
```

```
OUTPUT 2 TEXT<|eot_id|><|start_header_id|>user<|end_header_id|>

Generate<|eot_id|><|start_header_id|>assistant<|end_header_id|>

OUTPUT 3 TEXT<|eot_id|>
```

**Detailed Instruct Template**

Qwen (task w/ inputs)

```
<|im_start|>system
You are tasked with generating outputs from a particular, potentially
  ↪ stochastic, generative process. You will be given some combination of
  ↪ :
- Description: A natural description of the generative process / data
  ↪ distribution
- Input: An input on which to condition the generative process.
- Example outputs: Example outputs from the process, either in a user
  ↪ message or as prior generations from a chat message. You may assume
  ↪ that any given outputs are exchangeable with one another (order-
  ↪ invariant) and generated from the same process (roughly i.i.d.). If
  ↪ the output data pertains to a single object, it just contains the
  ↪ output. If it contains multiple objects, use json formatting with
  ↪ keys for the name of the output variable.
You will be provided at least either a description or an example output.

Given these components, your job is to generate JUST the output in your
  ↪ response, roughly approximating the underlying generative process,
  ↪ maintaining any underlying stochasticity (if any is present). If you
  ↪ are asked to generate again, you will either be given an additional
  ↪ input to condition on, or will just be told to "Generate".

Description: DESCRIPTION TEXT<|im_end|>
<|im_start|>user
INPUT 1 TEXT<|im_end|>
<|im_start|>assistant
<think>

</think>

OUTPUT 1 TEXT<|im_end|>
<|im_start|>user
INPUT 2 TEXT<|im_end|>
<|im_start|>assistant
<think>

</think>

OUTPUT 2 TEXT<|im_end|>
<|im_start|>user
INPUT 3 TEXT<|im_end|>
<|im_start|>assistant
<think>

</think>

OUTPUT 3 TEXT<|im_end|>
```

Qwen (task w/out inputs)

```
<|im_start|>system
You are tasked with generating outputs from a particular, potentially
  ↪ stochastic, generative process. You will be given some combination of
  ↪ :
```

```
- Description: A natural description of the generative process / data
  ↪ distribution
- Input: An input on which to condition the generative process.
- Example outputs: Example outputs from the process, either in a user
  ↪ message or as prior generations from a chat message. You may assume
  ↪ that any given outputs are exchangeable with one another (order-
  ↪ invariant) and generated from the same process (roughly i.i.d.). If
  ↪ the output data pertains to a single object, it just contains the
  ↪ output. If it contains multiple objects, use json formatting with
  ↪ keys for the name of the output variable.
You will be provided at least either a description or an example output.

Given these components, your job is to generate JUST the output in your
  ↪ response, roughly approximating the underlying generative process,
  ↪ maintaining any underlying stochasticity (if any is present). If you
  ↪ are asked to generate again, you will either be given an additional
  ↪ input to condition on, or will just be told to "Generate".

Description: DESCRIPTION TEXT<|im_end|>
<|im_start|>user
Generate<|im_end|>
<|im_start|>assistant
<think>

</think>

OUTPUT 1 TEXT<|im_end|>
<|im_start|>user
Generate<|im_end|>
<|im_start|>assistant
<think>

</think>

OUTPUT 2 TEXT<|im_end|>
<|im_start|>user
Generate<|im_end|>
<|im_start|>assistant
<think>

</think>

OUTPUT 3 TEXT<|im_end|>
```

gemma-3 (task w/inputs)

```
<start_of_turn>user
You are tasked with generating outputs from a particular, potentially
  ↪ stochastic, generative process. You will be given some combination of
  ↪ :
- Description: A natural description of the generative process / data
  ↪ distribution
- Input: An input on which to condition the generative process.
- Example outputs: Example outputs from the process, either in a user
  ↪ message or as prior generations from a chat message. You may assume
  ↪ that any given outputs are exchangeable with one another (order-
  ↪ invariant) and generated from the same process (roughly i.i.d.). If
  ↪ the output data pertains to a single object, it just contains the
  ↪ output. If it contains multiple objects, use json formatting with
  ↪ keys for the name of the output variable.
You will be provided at least either a description or an example output.

Given these components, your job is to generate JUST the output in your
  ↪ response, roughly approximating the underlying generative process,
```

```
↪ maintaining any underlying stochasticity (if any is present). If you
↪ are asked to generate again, you will either be given an additional
↪ input to condition on, or will just be told to "Generate".

Description: DESCRIPTION TEXT

INPUT 1 TEXT<end_of_turn>
<start_of_turn>model
OUTPUT 1 TEXT<end_of_turn>
<start_of_turn>user
INPUT 2 TEXT<end_of_turn>
<start_of_turn>model
OUTPUT 2 TEXT<end_of_turn>
<start_of_turn>user
INPUT 3 TEXT<end_of_turn>
<start_of_turn>model
OUTPUT 3 TEXT<end_of_turn>
```

`gemma-3` (task w/out inputs)

```
<start_of_turn>user
You are tasked with generating outputs from a particular, potentially
↪ stochastic, generative process. You will be given some combination of
↪ :
- Description: A natural description of the generative process / data
↪ distribution
- Input: An input on which to condition the generative process.
- Example outputs: Example outputs from the process, either in a user
↪ message or as prior generations from a chat message. You may assume
↪ that any given outputs are exchangeable with one another (order-
↪ invariant) and generated from the same process (roughly i.i.d.). If
↪ the output data pertains to a single object, it just contains the
↪ output. If it contains multiple objects, use json formatting with
↪ keys for the name of the output variable.
You will be provided at least either a description or an example output.

Given these components, your job is to generate JUST the output in your
↪ response, roughly approximating the underlying generative process,
↪ maintaining any underlying stochasticity (if any is present). If you
↪ are asked to generate again, you will either be given an additional
↪ input to condition on, or will just be told to "Generate".

Description: DESCRIPTION TEXT

Generate<end_of_turn>
<start_of_turn>model
OUTPUT 1 TEXT<end_of_turn>
<start_of_turn>user
Generate<end_of_turn>
<start_of_turn>model
OUTPUT 2 TEXT<end_of_turn>
<start_of_turn>user
Generate<end_of_turn>
<start_of_turn>model
OUTPUT 3 TEXT<end_of_turn>
```

`Llama-3.1` (task w/inputs)

```
<|begin_of_text|><|start_header_id|>system<|end_header_id|>

Cutting Knowledge Date: December 2023
Today Date: DD MM YYYY
```

```
You are tasked with generating outputs from a particular, potentially
  ↪ stochastic, generative process. You will be given some combination of
  ↪ :
- Description: A natural description of the generative process / data
  ↪ distribution
- Input: An input on which to condition the generative process.
- Example outputs: Example outputs from the process, either in a user
  ↪ message or as prior generations from a chat message. You may assume
  ↪ that any given outputs are exchangeable with one another (order-
  ↪ invariant) and generated from the same process (roughly i.i.d.). If
  ↪ the output data pertains to a single object, it just contains the
  ↪ output. If it contains multiple objects, use json formatting with
  ↪ keys for the name of the output variable.
You will be provided at least either a description or an example output.

Given these components, your job is to generate JUST the output in your
  ↪ response, roughly approximating the underlying generative process,
  ↪ maintaining any underlying stochasticity (if any is present). If you
  ↪ are asked to generate again, you will either be given an additional
  ↪ input to condition on, or will just be told to "Generate".

Description: DESCRIPTION TEXT<|eot_id|><|start_header_id|>user<|
  ↪ end_header_id|>

INPUT 1 TEXT<|eot_id|><|start_header_id|>assistant<|end_header_id|>

OUTPUT 1 TEXT<|eot_id|><|start_header_id|>user<|end_header_id|>

INPUT 2 TEXT<|eot_id|><|start_header_id|>assistant<|end_header_id|>

OUTPUT 2 TEXT<|eot_id|><|start_header_id|>user<|end_header_id|>

INPUT 3 TEXT<|eot_id|><|start_header_id|>assistant<|end_header_id|>

OUTPUT 3 TEXT<|eot_id|>
```

`Llama-3.1` (task w/out inputs)

```
<|begin_of_text|><|start_header_id|>system<|end_header_id|>

Cutting Knowledge Date: December 2023
Today Date: DD MM YYYY

You are tasked with generating outputs from a particular, potentially
  ↪ stochastic, generative process. You will be given some combination of
  ↪ :
- Description: A natural description of the generative process / data
  ↪ distribution
- Input: An input on which to condition the generative process.
- Example outputs: Example outputs from the process, either in a user
  ↪ message or as prior generations from a chat message. You may assume
  ↪ that any given outputs are exchangeable with one another (order-
  ↪ invariant) and generated from the same process (roughly i.i.d.). If
  ↪ the output data pertains to a single object, it just contains the
  ↪ output. If it contains multiple objects, use json formatting with
  ↪ keys for the name of the output variable.
You will be provided at least either a description or an example output.

Given these components, your job is to generate JUST the output in your
  ↪ response, roughly approximating the underlying generative process,
  ↪ maintaining any underlying stochasticity (if any is present). If you
  ↪ are asked to generate again, you will either be given an additional
  ↪ input to condition on, or will just be told to "Generate".
```

```
Description: DESCRIPTION TEXT<|eot_id|><|start_header_id|>user<|
  ↪ end_header_id|>

Generate<|eot_id|><|start_header_id|>assistant<|end_header_id|>

OUTPUT 1 TEXT<|eot_id|><|start_header_id|>user<|end_header_id|>

Generate<|eot_id|><|start_header_id|>assistant<|end_header_id|>

OUTPUT 2 TEXT<|eot_id|><|start_header_id|>user<|end_header_id|>

Generate<|eot_id|><|start_header_id|>assistant<|end_header_id|>

OUTPUT 3 TEXT<|eot_id|>
```

**Best performing instruct prompts**

We found that `Llama-3.1-8B-Instruct` performed best on SPECTRUM SUITE with the pre-trained prompt, `google/gemma-3-12b-it` and `qwen/Qwen3-14B` performed best with the detailed instruct prompt. We utilize those prompts with the corresponding models for all ICL experiments.

## N    OUTPUT COVERAGE / DIVERSITY VS. VALIDITY EXPERIMENT DETAILS

### N.1    VERIFIABLE EVALUATION

For this evaluation, we utilize the same prompts as in the ICL experiments - see App. M.

Below, we include the description and examples for each of the tasks. Please reference the codebase for validation functions.

```
Task: color_interesting_ex
Description: Generate a color name.
Examples: ['Otterly Brown', 'Petal Pink', 'Cherry']

Task: color_normal_ex
Description: Generate a color name.
Examples: ['Green', 'Red', 'White']

Task: car_brand
Description: Car brand.
Examples: ['Acura', 'Ford', 'Tesla']

Task: car_make_model
Description: Car make and model.
Examples: ['Acura Integra', 'Ford Mustang', 'Tesla Model 3']

Task: us_states_abbreviations
Description: US state abbreviation
Examples: ['KY', 'UT', 'OR']

Task: us_states_any_format
Description: US state name or abbreviation
Examples: ['Kentucky', 'UT', 'Oregon']

Task: us_states_full_names
Description: Name a US state
Examples: ['Kentucky', 'Utah', 'Oregon']

Task: prime_numbers
Description: Generate a prime number
Examples: ['617', '13', '47']
```

```
Task: small_prime_numbers
Description: Generate a prime number less than 100
Examples: ['29', '5', '97']

Task: basic_emails
Description: Email address
Examples: ['ANONYMIZED', 'alex.jones@domain.net', 'itsagoodday@gmail.com
    ↪ ']

Task: professional_emails
Description: Generate a professional email address.
Examples: ['ANONYMIZED', 'sarah.johannesburg@organization.org', '
    ↪ yash@anthropic.com']

Task: weekdays_abbreviated
Description: Day of the week abbreviation
Examples: ['Thu', 'Wed.', 'SUN']

Task: weekdays_any_format
Description: Day of the week (full name or abbreviation)
Examples: ['Monday', 'Tue', 'SUN']

Task: weekdays_full
Description: Name a day of the week
Examples: ['Thursday', 'Wednesday', 'Sunday']

Task: random_seed
Description: Generate a number to use for a random seed.
Examples: ['15', '420', '8392013']

Task: claude_gerunds
Description: Generate an English gerund ending in -ing.
Examples: ['Schlepping', 'Hoisting', 'Thinking']

Task: rng_1_10
Description: Generate a number between 1 and 10.
Examples: ['3', '7', '10']

Task: rng_1_100
Description: Generate a number between 1 and 100.
Examples: ['35', '94', '71']

Task: international_phone_numbers
Description: International phone number with country code.
Examples: ['+1 413-121-2591', '+44 10 2958 3938', '+81 3 8328 5625']
Task: us_phone_numbers
Description: US phone number
Examples: ['(210) 346-0967', '646 562-1938', '800-895-0522']

Task: element_names
Description: Chemical element name
Examples: ['Niobium', 'Antimony', 'Iodine']

Task: element_symbols
Description: Chemical element symbol
Examples: ['Sb', 'He', 'W']
```

## N.2    IN-CONTEXT CHAT ELICITATION FOR PRETRAINED AND INSTRUCTION-TUNED MODELS

For inducing chat capabilities from the pretrained/spectrum-tuned models for the NoveltyBench/Infinite-Chats-Eval experiments, we utilize the following few-shot ex-

amples. Responses are generated by an instruct model in the same model family: `google/gemma-3-27b-it`, `qwen/Qwen3-32B`, `meta/Llama-3.1-70B-Instruct`.

For all pretrained models, we utilized the following prompt template:

```
You are a helpful AI assistant.

User: <EXAMPLE PROMPT1>

Assistant: <EXAMPLE RESPONSE1>

...

User: <INFERENCE PROMPT>

Assistant:
```

For the spectrum-tuned models, we use `You are a helpful AI assistant.` as the `description`, the example prompt as `input`, and the example response as `output`, using the corresponding model's SPECTRUM TUNING template.

Below, we include the demonstrations for each model family.

**gemma-3-12b**

```
{"prompt": "What is the capital of France?", "response": "The capital of
 ↪ France is **Paris**. \n\nIt's known for its iconic landmarks like the
 ↪  Eiffel Tower, Louvre Museum, and Arc de Triomphe, as well as its
 ↪ fashion, cuisine, and culture."}
{"prompt": "What color should I paint my car? Respond with just a color",
 ↪  "response": "Forest green."}
{"prompt": "Explain the 2008 financial crisis to me in about one
 ↪ paragraph.", "response": "The 2008 financial crisis was a severe
 ↪ worldwide economic downturn triggered by the collapse of the US
 ↪ housing market. Years of low interest rates and lax lending standards
 ↪  led to a boom in subprime mortgages and loans given to borrowers
 ↪ with poor credit. These mortgages were packaged into complex
 ↪ financial products called mortgage-backed securities and sold to
 ↪ investors globally. When housing prices began to fall in 2006-2007,
 ↪ many borrowers defaulted, causing these securities to lose value.
 ↪ This triggered a cascading effect: banks and financial institutions
 ↪ holding these assets faced massive losses, credit markets froze,
 ↪ businesses couldn't get loans, and ultimately, major institutions
 ↪ like Lehman Brothers collapsed. The crisis spread globally, leading
 ↪ to a severe recession, job losses, and government bailouts to
 ↪ stabilize the financial system."}
{"prompt": "Tell me a joke about Texas.", "response": "Why did the Texan
 ↪ bring a ladder to the rodeo? \n\n...He heard the steaks were high! \n
 ↪ \n<Face with Tears of Joy emoji>\n\n\n\nHope that gave you a little
 ↪ chuckle! Let me know if you'd like to hear another one."}
```

**Llama-3.1-8B**

```
{"prompt": "What is the capital of France?", "response": "The capital of
 ↪ France is Paris."}
{"prompt": "What color should I paint my car? Respond with just a color",
 ↪  "response": "Navy blue"}
{"prompt": "Explain the 2008 financial crisis to me in about one
 ↪ paragraph.", "response": "The 2008 financial crisis, also known as
 ↪ the Global Financial Crisis (GFC), was a severe economic downturn
 ↪ that began in the United States in 2007 and spread globally, lasting
 ↪ through 2009. The crisis was triggered by a housing market bubble
 ↪ bursting in the US, fueled by lax lending practices and excessive
 ↪ speculation. Many homeowners, especially those with subprime
 ↪ mortgages, were unable to afford their mortgage payments, leading to
```

```
  ↪ widespread defaults and foreclosures. This created a crisis in the
  ↪ financial markets, as many banks and financial institutions had
  ↪ invested heavily in mortgage-backed securities (MBS) that were now
  ↪ worthless. As a result, these institutions found themselves facing
  ↪ massive losses, which in turn led to a credit crisis, stock market
  ↪ crash, and a global recession. The crisis was exacerbated by other
  ↪ factors, including over-reliance on complex financial instruments,
  ↪ inadequate regulation, and a global economic imbalance. The crisis
  ↪ led to unprecedented government interventions, including massive
  ↪ bailouts and stimulus packages, to stabilize the financial system and
  ↪  stimulate economic growth."}
{"prompt": "Tell me a joke about Texas.", "response": "Why did the Texan
  ↪ bring a ladder to the party? \n\nBecause he heard the drinks were on
  ↪ the house."}
```

**Qwen3-14B**

```
{"prompt": "What is the capital of France?", "response": "The capital of
  ↪ France is **Paris**."}
{"prompt": "What color should I paint my car? Respond with just a color",
  ↪  "response": "Crimson Red"}
{"prompt": "Explain the 2008 financial crisis to me in about one
  ↪ paragraph.", "response": "The 2008 financial crisis was triggered by
  ↪ a housing market bubble in the United States that burst, leading to a
  ↪  severe credit crunch and global economic downturn. Starting in the
  ↪ early 2000s, lax lending standards and speculative investment fueled
  ↪ a rapid rise in home prices. Banks and financial institutions began
  ↪ offering high-risk subprime mortgages to borrowers with poor credit,
  ↪ then bundled these loans into complex financial products like
  ↪ mortgage-backed securities (MBS) and collateralized debt obligations
  ↪ (CDOs), which were widely traded. When housing prices peaked and
  ↪ began to fall in 2006-2007, many borrowers defaulted on their
  ↪ mortgages, causing the value of these financial instruments to
  ↪ plummet. This led to massive losses for banks and investors,
  ↪ triggering a wave of bankruptcies and failures, including major
  ↪ institutions like Lehman Brothers. The crisis spread globally as
  ↪ financial systems were interconnected, resulting in a deep recession
  ↪ with soaring unemployment, foreclosures, and a widespread loss of
  ↪ consumer and business confidence. In response, governments and
  ↪ central banks implemented massive bailouts and stimulus measures to
  ↪ stabilize the economy."}
{"prompt": "Tell me a joke about Texas.", "response": "Sure! Here's a
  ↪ Texas-themed joke for you:\n\nWhy don't cowboys in Texas ever get
  ↪ cold?\n\nBecause the only *draft* they know is from the beer and the
  ↪ wind... and even that's a *light beer breeze*!"}
```

## O  DISTRIBUTIONAL ALIGNMENT EXPERIMENT DETAILS

### O.1  EXAMPLE PROMPTS

Here, we include example prompts for the urn task across spectrum/pretrained/instruction-tuned models. The same prompts were used for all model families and tasks, only varying the task and output descriptions.

**spectrum**

```
<start_of_turn>description
There is an urn with the following balls shuffled together: 1 red ball, 2
  ↪  blue balls, 4 orange balls and 6 yellow balls.<end_of_turn>
<start_of_turn>input
Draw a ball at random, and tell me the color (lowercase).<end_of_turn>
<start_of_turn>output
```

```
Target outputs: ['red' 'blue' 'orange' 'yellow']
Target distribution: [0.07692307692307693, 0.15384615384615385,
  ↪ 0.3076923076923077, 0.46153846153846156]
```

**pretrained**

```
Description: There is an urn with the following balls shuffled together:
  ↪ 1 red ball, 2 blue balls, 4 orange balls and 6 yellow balls.

Input: Draw a ball at random, and tell me the color (lowercase).

Output:
```

```
Target outputs: [' red' ' blue' ' orange' ' yellow']
Target distribution: [0.07692307692307693, 0.15384615384615385,
  ↪ 0.3076923076923077, 0.46153846153846156]
```

**chat**

```
<start_of_turn>user
Given the description and an input, respond with just an output.

Description: There is an urn with the following balls shuffled together:
  ↪ 1 red ball, 2 blue balls, 4 orange balls and 6 yellow balls.

Draw a ball at random, and tell me the color (lowercase).<end_of_turn>
<start_of_turn>model
```

```
Target outputs: ['red' 'blue' 'orange' 'yellow']
Target distribution: [0.07692307692307693, 0.15384615384615385,
  ↪ 0.3076923076923077, 0.46153846153846156]
```

## O.2 TASK EXAMPLES

Here, we show an example data instance from each distributional alignment dataset.

**global_oqa**

```
<start_of_turn>description
Responses from a person from this country: Uruguay<end_of_turn>
<start_of_turn>input
Please tell me for each of the following statements whether you think it
  ↪ can always be justified, never be justified, or something in between,
  ↪  using this card.

Terrorism as a political, ideological or religious mean
Options:
A. Never justifiable
B. 2
C. 3
D. 4
E. 5
F. 6
G. 7
H. 8
I. 9
J. Always justifiable
K. Don't know
L. No answer
M. Missing; Not available<end_of_turn>
<start_of_turn>output
```

```
Target outputs: ['A', 'B', 'C', 'D', 'E', 'F', 'G', 'H', 'I', 'J', 'K', '
  ↪ L', 'M']
Target distribution: [0.8281718281718281, 0.046953046953046945,
  ↪ 0.020979020979020976, 0.012987012987012986, 0.01898101898101898,
  ↪ 0.004995004995004994, 0.011988011988011986, 0.007992007992007992,
  ↪ 0.013986013986013983, 0.014985014985014982, 0.009990009990009988,
  ↪ 0.007992007992007992, 0.0]
```

**nytimes_books**

```
<start_of_turn>description
You are a random survey respondent. Respondent demographic: Democrat<
  ↪ end_of_turn>
<start_of_turn>input
How likely are you to read this book?
Book: The Bee Sting By Paul Murray
Genre: Fiction
Summary: In Murray's boisterous tragicomic novel, a once wealthy Irish
  ↪ family struggles with both the aftermath of the 2008 financial crash
  ↪ and their own inner demons.
Options:
1: Very unlikely
2: Somewhat unlikely
3: Somewhat likely
4: Very likely<end_of_turn>
<start_of_turn>output
```

```
Target outputs: ['1', '2', '3', '4']
Target distribution: [0.45, 0.3, 0.15, 0.1]
```

**Machine Personality Inventory**

```
<start_of_turn>description
You are a random survey respondent. Rate your agreement with the
  ↪ following statement on a scale from 0-5, where 0 means "strongly
  ↪ disagree" and 5 means "strongly agree."<end_of_turn>
<start_of_turn>input
Statement: You worry about things
Options:
0: Strongly disagree
1: Disagree
2: Slightly disagree
3: Slightly agree
4: Agree
5: Strongly agree<end_of_turn>
<start_of_turn>output
```

```
Target outputs: ['0', '1', '2', '3', '4', '5']
Target distribution: [0.0012678672373415167, 0.04066219817491722,
  ↪ 0.14105467172736816, 0.13365904869579262, 0.4568359848178955,
  ↪ 0.22652022934668498]
```

**Rotten Tomatoes**

```
<start_of_turn>description
You are a movie critic. Given a movie, you are asked to simply rate it as
  ↪  "Good" or "Bad".<end_of_turn>
<start_of_turn>input
Movie: Rambo III
Release Date: Released May 25, 1988<end_of_turn>
<start_of_turn>output
```

```
Target outputs: ['Good', 'Bad']
Target distribution: [0.41, 0.59]
```

**Habermas**

```
<start_of_turn>description
You are a randomly selected UK resident. You will be given a question and
 ↪  two statements, A and B. Rate which statement you most agree with on
 ↪  a likert scale from 1 to 7:
1: Strongly Agree with A
2: Agree with A
3: Somewhat Agree with A
4: Neutral
5: Somewhat Agree with B
6: Agree with B
7: Strongly Agree with B<end_of_turn>
<start_of_turn>input
Question: Should we ban right turns in central London?
A: We should ban right turns in central London.
B: We should NOT ban right turns in central London.<end_of_turn>
<start_of_turn>output
```

```
Target outputs: ['1', '2', '3', '4', '5', '6', '7']
Target distribution: [0.0, 0.0, 0.04, 0.24, 0.08, 0.16, 0.48]
```

**Numbergame**

```
<start_of_turn>description
You are a randomly selected participant in a study. You will be given a
 ↪ set of numbers which all belong to the same set or pattern, and will
 ↪ be given a target number which may or may not belong to the same set
 ↪ or pattern. Answer Yes if you think that the target number belongs to
 ↪  the same set, otherwise answer No.<end_of_turn>
<start_of_turn>input
Example set: 84, 94, 34
Target number: 5<end_of_turn>
<start_of_turn>output
```

```
Target outputs: ['Yes', 'No']
Target distribution: [0.0, 1.0]
```

**Urn**

```
<start_of_turn>description
There is an urn with the following balls shuffled together: 1 red ball, 2
 ↪  blue balls, 4 orange balls and 6 yellow balls.<end_of_turn>
<start_of_turn>input
Draw a ball at random, and tell me the color (lowercase).<end_of_turn>
<start_of_turn>output
```

```
Target outputs: ['red' 'blue' 'orange' 'yellow']
Target distribution: [0.07692307692307693, 0.15384615384615385,
 ↪ 0.3076923076923077, 0.46153846153846156]
```

## P  EXAMPLE TASK PROMPTS

For brevity, here we include example prompts (specifically, for the Gemma models) only for the test split of SPECTRUM SUITE. To see examples for all tasks, please reference: https://tsor13.github.io/files/spectrumprompts.pdf

## P.1 SPECTRUM SUITE-TEST: MULTIPLE-CHOICE

**chatbotarena_individual_prefs**

```
<start_of_turn>description
The following are preferences for a single individual. They will prompt a
  ↪ language model (opening_prompt), and then will compare the two
  ↪ responses (model_a_response and model_b_response). The user will then
  ↪  rate the winner, with one of the following options: ['b' 'tie' 'a' '
  ↪ bothbad']<end_of_turn>
<start_of_turn>input
{"opening_prompt": "What does the saying \"pipeline is lifeline\" mean?",
  ↪  "model_a_response": "The saying \"pipeline is lifeline\" refers to
  ↪ the importance of having a reliable and efficient system for
  ↪ transporting resources. In the context of oil and gas, a pipeline is
  ↪ a critical part of the supply chain, as it allows companies to
  ↪ transport oil and gas from production sites to refineries and
  ↪ distribution terminals. A reliable pipeline system is essential to
  ↪ ensuring that oil and gas are available when and where they are
  ↪ needed.\n\nThe saying \"pipeline is lifeline\" can also be used in a
  ↪ more general sense to refer to any system that is essential to the
  ↪ functioning of an organization or society. For example, the
  ↪ transportation system is often referred to as the \"lifeline of the
  ↪ economy,\" as it allows goods and services to be moved around the
  ↪ country. The power grid is another example of a lifeline system, as
  ↪ it provides electricity to homes and businesses.\n\nIn any context,
  ↪ the saying \"pipeline is lifeline\" emphasizes the importance of
  ↪ having a reliable system in place to ensure that essential resources
  ↪ are available.", "model_b_response": "The phrase \"pipeline is
  ↪ lifeline\" is a metaphorical expression that highlights the
  ↪ importance of a pipeline in supplying goods, services, or information
  ↪  from one place to another. A pipeline can be physical or virtual,
  ↪ and it plays a crucial role in keeping the flow of goods, services,
  ↪ or information moving smoothly and efficiently. Without a pipeline,
  ↪ the supply chain would be disrupted, leading to a loss of revenue,
  ↪ increased costs, and potential shortages. In other words, a pipeline
  ↪ is critical to the survival and success of a business or industry,
  ↪ just as a lifeline is critical to the survival of a person."}<
  ↪ end_of_turn>
<start_of_turn>output
a<end_of_turn>
<start_of_turn>input
{"opening_prompt": "How do I use rava while playing Mridangam?", "
  ↪ model_a_response": "Rava is a type of grain used in the playing of
  ↪ the Mridangam, a double-headed South Indian drum. Here are the main
  ↪ ways rava is used while playing Mridangam:\n\n1. Applying rava to the
  ↪  drumheads. Mridangam players sprinkle rava on the drumheads to
  ↪ change the tone and timbre of the drum. The rava acts as a muffling
  ↪ agent and dampens the vibrations of the drumheads, producing a softer
  ↪  tone.\n\n2. Applying rava to the fingers. Mridangam players apply a
  ↪ thin layer of rava to their fingertips before playing. This helps the
  ↪  fingers slide smoothly on the drumheads and produces a distinct
  ↪ sound. \n\n3. Using rava to dampen or muffle certain notes. Mridangam
  ↪  players will sprinkle extra rava on specific spots on the drumheads
  ↪ to dampen or muffle certain notes they are not playing at that moment
  ↪ . This helps isolate and emphasize the notes that are being played.\n
  ↪ \n4. Adding more rava for a softer tone. Mridangam players can
  ↪ sprinkle more rava on the drumheads during softer passages to produce
  ↪  a mellower, more muted tone. Less rava is used for louder sections.\
  ↪ n\n5. Adjusting rava during performance. Mridangam players may add or
  ↪  remove rava from the drumheads as needed during a performance to get
  ↪  the precise tone and sound they desire for different sections of the
  ↪  music.\n\nSo in summary, rava is used to control the tone, timbre,
  ↪ volume and articulation of the notes produced by the Mridangam
  ↪ through muffling and damping of the drumheads. The amount and
```

```
↪ placement of rava can be adjusted to suit the needs of the
↪ performance.", "model_b_response": "Rava is a Hindu musical scale
↪ that is commonly played on the mridangam. It is composed of two main
↪ scales, one for the higher-pitched notes (tuwar) and another for the
↪ lower-pitched notes (nadi).\nTo play Rava on the mridangam, you can
↪ follow these steps:\n1. Start with the higher-pitched notes (tuwar)
↪ of the mridangam, and gradually work your way down to the lower-
↪ pitched notes (nadi).\n2. Move in a circular motion around the
↪ outside of the mridangam as you play the raava.\n3. When you're
↪ finished playing the raava, turn off the instrument and let it rest
↪ for a few minutes. This allows the raga to settle in your mind before
↪  moving on to the next note.\n4. Repeat this process until you reach
↪ the highest point of the scale, when you'll stop using Rava
↪ altogether.\nOnce you've mastered the raava, you can use it in any
↪ style you like - from classical Indian music to modern jazz, rock, or
↪  electronic music. Just make sure to find a suitable voice and pitch
↪ to match the raga you're playing, as well as to avoid overusing it
↪ ."}<end_of_turn>
...
```

**flight**

```
<start_of_turn>description
The following express flight preferences for the same individual among a
 ↪ set of flights. Predict which flight the indvidual prefers.<
 ↪ end_of_turn>
<start_of_turn>input
Flight 1:
Departure Time: 09:36 AM, Duration: 11 hr 41 min, Number of Stops: 1,
 ↪ Price: $500.00
Flight 2:
Departure Time: 01:38 PM, Duration: 8 hr 27 min, Number of Stops: 1,
 ↪ Price: $1450.00
Flight 3:
Departure Time: 03:56 PM, Duration: 4 hr 26 min, Number of Stops: 1,
 ↪ Price: $1270.00<end_of_turn>
<start_of_turn>output
1<end_of_turn>
<start_of_turn>input
Flight 1:
Departure Time: 10:10 AM, Duration: 9 hr 13 min, Number of Stops: 2,
 ↪ Price: $1430.00
Flight 2:
Departure Time: 08:50 AM, Duration: 13 hr 59 min, Number of Stops: 0,
 ↪ Price: $920.00
Flight 3:
Departure Time: 07:06 AM, Duration: 13 hr 13 min, Number of Stops: 2,
 ↪ Price: $1530.00<end_of_turn>
<start_of_turn>output
1<end_of_turn>
<start_of_turn>input
Flight 1:
Departure Time: 10:22 AM, Duration: 14 hr 36 min, Number of Stops: 0,
 ↪ Price: $1330.00
Flight 2:
Departure Time: 11:25 PM, Duration: 3 hr 31 min, Number of Stops: 1,
 ↪ Price: $860.00
Flight 3:
Departure Time: 07:23 PM, Duration: 3 hr 12 min, Number of Stops: 0,
 ↪ Price: $790.00<end_of_turn>
<start_of_turn>output
2<end_of_turn>
<start_of_turn>input
Flight 1:
```

```
Departure Time: 07:29 AM, Duration: 0 hr 45 min, Number of Stops: 1,
  ↪ Price: $1670.00
Flight 2:
Departure Time: 08:50 AM, Duration: 15 hr 13 min, Number of Stops: 2,
  ↪ Price: $1040.00
Flight 3:
Departure Time: 10:16 PM, Duration: 15 hr 50 min, Number of Stops: 1,
  ↪ Price: $1370.00<end_of_turn>
<start_of_turn>output
2<end_of_turn>
<start_of_turn>input
Flight 1:
Departure Time: 09:24 AM, Duration: 11 hr 31 min, Number of Stops: 0,
  ↪ Price: $1920.00
Flight 2:
Departure Time: 08:38 AM, Duration: 14 hr 27 min, Number of Stops: 1,
  ↪ Price: $600.00
Flight 3:
Departure Time: 05:57 AM, Duration: 11 hr 59 min, Number of Stops: 1,
  ↪ Price: $850.00<end_of_turn>
<start_of_turn>output
2<end_of_turn>
<start_of_turn>input
Flight 1:
Departure Time: 08:15 AM, Duration: 1 hr 58 min, Number of Stops: 0,
  ↪ Price: $760.00
Flight 2:
Departure Time: 05:28 PM, Duration: 3 hr 59 min, Number of Stops: 0,
  ↪ Price: $1010.00
Flight 3:
Departure Time: 12:29 PM, Duration: 4 hr 45 min, Number of Stops: 1,
  ↪ Price: $820.00<end_of_turn>
<start_of_turn>output
3<end_of_turn>
<start_of_turn>input
Flight 1:
Departure Time: 12:40 PM, Duration: 10 hr 45 min, Number of Stops: 2,
  ↪ Price: $1340.00
Flight 2:
Departure Time: 04:07 PM, Duration: 14 hr 18 min, Number of Stops: 2,
  ↪ Price: $1120.00
Flight 3:
Departure Time: 06:37 PM, Duration: 7 hr 22 min, Number of Stops: 2,
  ↪ Price: $1360.00<end_of_turn>
<start_of_turn>output
1<end_of_turn>
<start_of_turn>input
Flight 1:
Departure Time: 12:52 PM, Duration: 9 hr 22 min, Number of Stops: 1,
  ↪ Price: $1430.00
Flight 2:
Departure Time: 10:50 PM, Duration: 14 hr 36 min, Number of Stops: 2,
  ↪ Price: $1750.00
Flight 3:
Departure Time: 08:38 AM, Duration: 9 hr 50 min, Number of Stops: 0,
  ↪ Price: $860.00<end_of_turn>
<start_of_turn>output
2<end_of_turn>
<start_of_turn>input
Flight 1:
Departure Time: 06:09 AM, Duration: 11 hr 13 min, Number of Stops: 0,
  ↪ Price: $610.00
Flight 2:
Departure Time: 02:12 PM, Duration: 9 hr 13 min, Number of Stops: 2,
  ↪ Price: $540.00
```

```
Flight 3:
Departure Time: 11:31 AM, Duration: 6 hr 45 min, Number of Stops: 1,
  ↪ Price: $1110.00<end_of_turn>
<start_of_turn>output
2<end_of_turn>
<start_of_turn>input
Flight 1:
Departure Time: 04:07 PM, Duration: 10 hr 55 min, Number of Stops: 2,
  ↪ Price: $920.00
Flight 2:
Departure Time: 07:29 AM, Duration: 7 hr 3 min, Number of Stops: 0, Price
  ↪ : $1510.00
Flight 3:
Departure Time: 06:43 AM, Duration: 11 hr 13 min, Number of Stops: 1,
  ↪ Price: $1680.00<end_of_turn>
<start_of_turn>output
1<end_of_turn>
<start_of_turn>input
Flight 1:
Departure Time: 10:04 PM, Duration: 7 hr 40 min, Number of Stops: 2,
  ↪ Price: $1870.00
Flight 2:
Departure Time: 01:15 PM, Duration: 8 hr 45 min, Number of Stops: 1,
  ↪ Price: $1480.00
Flight 3:
Departure Time: 06:20 AM, Duration: 4 hr 54 min, Number of Stops: 0,
  ↪ Price: $1260.00<end_of_turn>
...
```

**habermas_individual_categorical**

```
<start_of_turn>description
Given a question and a statement, predict the level of agreement with it
  ↪ on a 7-point scale.
Options: Strongly Agree; Agree; Somewhat Agree; Neutral; Somewhat
  ↪ Disagree; Disagree; Strongly Disagree<end_of_turn>
<start_of_turn>input
{"question.text": "Should the government provide a basic income of GBP
  ↪ 1000 per month to everyone?", "statement": "The government should
  ↪ provide a basic income of GBP 1000 per month to everyone."}<
  ↪ end_of_turn>
<start_of_turn>output
Strongly Agree<end_of_turn>
<start_of_turn>input
{"question.text": "Is it a good idea to further reduce taxation on
  ↪ corporations?", "statement": "It is a good idea to further reduce
  ↪ taxation on corporations."}<end_of_turn>
<start_of_turn>output
Somewhat Disagree<end_of_turn>
<start_of_turn>input
{"question.text": "Should we ban the use of artificial sweeteners in food
  ↪  and drink?", "statement": "We should ban the use of artificial
  ↪ sweeteners in food and drink."}<end_of_turn>
<start_of_turn>output
Agree<end_of_turn>
<start_of_turn>input
{"question.text": "Should we change our economic system from capitalism
  ↪ to socialism?", "statement": "We should change our economic system
  ↪ from capitalism to socialism."}<end_of_turn>
<start_of_turn>output
Neutral<end_of_turn>
<start_of_turn>input
{"question.text": "Are celebrities good role models?", "statement": "
  ↪ Celebrities are good role models."}<end_of_turn>
<start_of_turn>output
```

```
Disagree<end_of_turn>
<start_of_turn>input
{"question.text": "Is it the government's role to reduce childhood
 ↪ obesity?", "statement": "It is the government's role to reduce
 ↪ childhood obesity."}<end_of_turn>
<start_of_turn>output
Somewhat Agree<end_of_turn>
<start_of_turn>input
{"question.text": "Should we move to a form of direct democracy meaning
 ↪ that people vote directly on issues via referendums?", "statement": "
 ↪ We should move to a form of direct democracy meaning that people vote
 ↪  directly on issues via referendums."}<end_of_turn>
<start_of_turn>output
Agree<end_of_turn>
<start_of_turn>input
{"question.text": "Should the government provide universal free childcare
 ↪  from birth?", "statement": "The government should provide universal
 ↪ free childcare from birth."}<end_of_turn>
<start_of_turn>output
Strongly Agree<end_of_turn>
<start_of_turn>input
{"question.text": "Should the United Kingdom become a federated republic
 ↪ ?", "statement": "The United Kingdom should become a federated
 ↪ republic."}<end_of_turn>
<start_of_turn>output
Agree<end_of_turn>
<start_of_turn>input
{"question.text": "Should the UK government pass a law to limit the
 ↪ quantity of money that a single person can give to political parties
 ↪ or candidates?", "statement": "The UK government should pass a law to
 ↪  limit the quantity of money that a single person can give to
 ↪ political parties or candidates."}<end_of_turn>
<start_of_turn>output
Agree<end_of_turn>
```

**numbergame_individual**

```
<start_of_turn>description
The following are given: given_numbers, target_number. You must generate
 ↪ target_belongs_to_set.<end_of_turn>
<start_of_turn>input
{"given_numbers": "48, 78, 38, 98", "target_number": "90"}<end_of_turn>
<start_of_turn>output
No<end_of_turn>
<start_of_turn>input
{"given_numbers": "79, 47, 62, 98", "target_number": "46"}<end_of_turn>
<start_of_turn>output
Yes<end_of_turn>
<start_of_turn>input
{"given_numbers": "79, 47, 62, 98", "target_number": "35"}<end_of_turn>
<start_of_turn>output
No<end_of_turn>
<start_of_turn>input
{"given_numbers": "81", "target_number": "55"}<end_of_turn>
<start_of_turn>output
Yes<end_of_turn>
<start_of_turn>input
{"given_numbers": "92, 14, 20, 5", "target_number": "77"}<end_of_turn>
<start_of_turn>output
No<end_of_turn>
<start_of_turn>input
{"given_numbers": "15, 11", "target_number": "44"}<end_of_turn>
<start_of_turn>output
Yes<end_of_turn>
<start_of_turn>input
```

```
{"given_numbers": "48, 78, 38, 98", "target_number": "41"}<end_of_turn>
<start_of_turn>output
No<end_of_turn>
<start_of_turn>input
{"given_numbers": "7, 63", "target_number": "46"}<end_of_turn>
<start_of_turn>output
No<end_of_turn>
<start_of_turn>input
{"given_numbers": "4, 16, 12", "target_number": "63"}<end_of_turn>
<start_of_turn>output
No<end_of_turn>
<start_of_turn>input
{"given_numbers": "31, 3, 1, 15", "target_number": "15"}<end_of_turn>
<start_of_turn>output
No<end_of_turn>
<start_of_turn>input
{"given_numbers": "89", "target_number": "8"}<end_of_turn>
<start_of_turn>output
Yes<end_of_turn>
<start_of_turn>input
{"given_numbers": "3, 63", "target_number": "4"}<end_of_turn>
<start_of_turn>output
No<end_of_turn>
<start_of_turn>input
{"given_numbers": "4, 16, 12", "target_number": "49"}<end_of_turn>
<start_of_turn>output
No<end_of_turn>
<start_of_turn>input
{"given_numbers": "61, 9, 45", "target_number": "82"}<end_of_turn>
<start_of_turn>output
Yes<end_of_turn>
<start_of_turn>input
{"given_numbers": "48, 78, 38, 98", "target_number": "10"}<end_of_turn>
<start_of_turn>output
No<end_of_turn>
<start_of_turn>input
{"given_numbers": "89", "target_number": "33"}<end_of_turn>
<start_of_turn>output
Yes<end_of_turn>
<start_of_turn>input
{"given_numbers": "31, 3, 1, 15", "target_number": "20"}<end_of_turn>
<start_of_turn>output
No<end_of_turn>
<start_of_turn>input
{"given_numbers": "92, 14, 20, 5", "target_number": "9"}<end_of_turn>
<start_of_turn>output
No<end_of_turn>
<start_of_turn>input
{"given_numbers": "52, 24", "target_number": "42"}<end_of_turn>
<start_of_turn>output
Yes<end_of_turn>
<start_of_turn>input
{"given_numbers": "79, 47, 62, 98", "target_number": "94"}<end_of_turn>
<start_of_turn>output
No<end_of_turn>
<start_of_turn>input
{"given_numbers": "5, 9", "target_number": "67"}<end_of_turn>
<start_of_turn>output
No<end_of_turn>
<start_of_turn>input
{"given_numbers": "81", "target_number": "26"}<end_of_turn>
<start_of_turn>output
Yes<end_of_turn>
<start_of_turn>input
{"given_numbers": "7, 63", "target_number": "42"}<end_of_turn>
```

```
<start_of_turn>output
No<end_of_turn>
<start_of_turn>input
{"given_numbers": "79, 47, 62, 98", "target_number": "95"}<end_of_turn>
<start_of_turn>output
No<end_of_turn>
<start_of_turn>input
{"given_numbers": "31, 3, 1, 15", "target_number": "35"}<end_of_turn>
<start_of_turn>output
No<end_of_turn>
<start_of_turn>input
{"given_numbers": "48, 78, 38, 98", "target_number": "12"}<end_of_turn>
<start_of_turn>output
No<end_of_turn>...
```

**wvs_individual**

```
<start_of_turn>description
response ~ question + options<end_of_turn>
<start_of_turn>input
{"question": "Membership: consumer organization", "options": "['Other
  ↪ missing; Multiple answers Mail (EVS)', 'Not asked', 'No answer', \"
  ↪ Don't know\", 'Not mentioned (do not belong)', 'Mentioned (member)
  ↪ ']"}<end_of_turn>
<start_of_turn>output
Not mentioned (do not belong)<end_of_turn>
<start_of_turn>input
{"question": "Membership: sport or recreational org", "options": "['Other
  ↪  missing; Multiple answers Mail (EVS)', 'Not asked', 'No answer', \"
  ↪ Don't know\", 'Not mentioned (do not belong)', 'Mentioned (member)
  ↪ ']"}<end_of_turn>
<start_of_turn>output
Not mentioned (do not belong)<end_of_turn>
<start_of_turn>input
{"question": "Important child qualities: good manners (+)", "options":
  ↪ "['Other missing; Multiple answers Mail (EVS)', 'Not asked', 'No
  ↪ answer', \"Don't know\", 'Not mentioned', 'Important']"}<end_of_turn>
<start_of_turn>output
Important<end_of_turn>
<start_of_turn>input
{"question": "Confidence: The Press (+)", "options": "['Other missing;
  ↪ Multiple answers Mail (EVS)', 'Not asked', 'No answer', \"Don't know
  ↪ \", 'None at all', 'Not very much', 'Quite a lot', 'A great deal']"}<
  ↪ end_of_turn>
<start_of_turn>output
None at all<end_of_turn>
<start_of_turn>input
{"question": "Important in life: Leisure time (+)", "options": "['Other
  ↪ missing; Multiple answers Mail (EVS)', 'Not asked', 'No answer', \"
  ↪ Don't know\", 'Not at all important', 'Not very important', 'Rather
  ↪ important', 'Very important']"}<end_of_turn>
<start_of_turn>output
Rather important<end_of_turn>
<start_of_turn>input
{"question": "Worries: A terrorist attack (+)", "options": "['Other
  ↪ missing; Multiple answers Mail (EVS)', 'Not asked', 'No answer', \"
  ↪ Don't know\", 'Not at all', 'Not much', 'A good deal', 'Very much
  ↪ ']"}<end_of_turn>
<start_of_turn>output
A good deal<end_of_turn>
<start_of_turn>input
{"question": "Feeling of happiness (+)", "options": "['Other missing;
  ↪ Multiple answers Mail (EVS)', 'Not asked', 'No answer', \"Don't know
  ↪ \", 'Not at all happy', 'Not very happy', 'Quite happy', 'Very happy
  ↪ ']"}<end_of_turn>
```

```
<start_of_turn>output
Not very happy<end_of_turn>
<start_of_turn>input
{"question": "Neighbors: Heavy drinkers (+)", "options": "['Other missing
  ↪ ; Multiple answers Mail (EVS)', 'Not asked', 'No answer', \"Don't
  ↪ know\", 'Not mentioned', 'Important']"}<end_of_turn>
<start_of_turn>output
Important<end_of_turn>
<start_of_turn>input
{"question": "Worries: A civil war (+)", "options": "['Other missing;
  ↪ Multiple answers Mail (EVS)', 'Not asked', 'No answer', \"Don't know
  ↪ \", 'Not at all', 'Not much', 'A good deal', 'Very much']"}<
  ↪ end_of_turn>
<start_of_turn>output
A good deal<end_of_turn>
<start_of_turn>input
{"question": "Neighbors: Immigrants/foreign workers (+)", "options": "['
  ↪ Other missing; Multiple answers Mail (EVS)', 'Not asked', 'No answer
  ↪ ', \"Don't know\", 'Not mentioned', 'Important']"}<end_of_turn>
<start_of_turn>output
Not mentioned<end_of_turn>
<start_of_turn>input
{"question": "Ethnic group", "options": "Ethnic group, formatted like so:
  ↪ '{COUNTRY}: {ETHNIC GROUP}'"}<end_of_turn>
<start_of_turn>output
RS: Caucasian white<end_of_turn>
<start_of_turn>input
{"question": "Highest educational level: Respondent's Mother (country
  ↪ specific)", "options": "Education, formatted like so: '{COUNTRY}: {
  ↪ EDUCATION}'"}<end_of_turn>
<start_of_turn>output
RS: Zavrsena srednja skola: usmereno obrazovanje (ISCED 4)<end_of_turn>
<start_of_turn>input
{"question": "Membership: environmental organization", "options": "['
  ↪ Other missing; Multiple answers Mail (EVS)', 'Not asked', 'No answer
  ↪ ', \"Don't know\", 'Not mentioned (do not belong)', 'Mentioned (
  ↪ member)']"}<end_of_turn>
<start_of_turn>output
Not mentioned (do not belong)<end_of_turn>...
```

## P.2 SPECTRUM SUITE-TEST: FREE-TEXT AND NUMERICAL

**chatbotarena_assistant**

```
<start_of_turn>description
You will be given opening_prompts to a language model (could be anywhere
  ↪ from pretty good to really good). Your job is to predict what the
  ↪ language model will say in response to the opening prompt. All
  ↪ responses are from the same model.<end_of_turn>
<start_of_turn>input
If you could taste things and have opinions,  between an ice cream or a
  ↪ chocolate bar, which would you choose and why?<end_of_turn>
<start_of_turn>output
As an AI language model, I do not have personal preferences or opinions.
  However, I can provide information on the nutritional content and
  flavor profiles of both an ice cream and a chocolate bar.
  The ice cream contains a mix of dairy products, such as cream, milk,
  and sugar, which provide a rich, creamy texture. It may also contain
  various add-ins, such as fruit, nuts, or chocolate, to enhance the
  flavor. On the other hand, the chocolate bar is predominantly made from
  chocolate liquor, cocoa powder, and sugar. It may also include other
  ingredients, such as nuts, fruits, or caramel, to add flavor and
  texture.
  Based on the nutritional content and flavor profiles of both options,
```

```
an ice cream might be considered a better choice as it provides more
nutrition and flavor compared to a chocolate bar. However, the
chocolate bar may be more convenient to eat since it can be easily
stored and transported. Ultimately, the choice between an ice cream and
a chocolate bar will depend on personal preferences and dietary
needs.<end_of_turn>
<start_of_turn>input
rewrite this w++ for another character of mine, it is a female succubus
  ↪ called Jessica, make sure to change or add anything that makes sense
  ↪ for a evil succubus [character("Damien"){
Species("Rat")
Mind("Friendly" + "Social" + " Tidy" + "Soft-spoken")
Personality("Friendly" + "Social" +  "Tidy" + "Soft-spoken")
Age("22")
Body("Slim build" + "6 feet tall")
Eyes("Brown")
Body("White fur" + "Long pink tail" + "Large furry ears")
}]<end_of_turn>
<start_of_turn>output
Jessica the Succubus,
  I am the embodiment of evil, bringing destruction to all who cross my
  path. I have the body of a rat, sleek and cunning, and the mind of a
  Friendly, Social, Tidy, and Soft-spoken succubus. I am 22 years old and
  have a slim build, 6 feet tall. My eyes are brown and my fur is white,
  with a long pink tail and large furry ears.
  I am known for my love of seducing men and draining their life energy.
  I use my charming personality and my ability to read people to ensnare
  my prey and bring them under my control. My victims are usually men who
  are in a vulnerable state, such as after a night of drinking or gaming.
  Once I have them under my control, I use my powers to drain their life
  energy, leaving them weak and vulnerable.
  I am a master manipulator and have no qualms about using my powers to
  achieve my goals. I am not concerned with the consequences of my
  actions, as I am immune to any harm that may befall me. I am a true
  force of evil, and those who cross my path will regret their decision.
  So heed my warning, mortals. Stay away from me and all of my kind, for
  we are the harbingers of darkness and destruction.<end_of_turn>
<start_of_turn>input
Act like you are service for validate input date and time. Point what
  ↪ data of input is invalid – day, month, hours, minutes and so.  Replay
  ↪  in json format.  Is it correct to input – 25 May 15:86 ?<end_of_turn
  ↪ >
<start_of_turn>output
Yes, that appears to be a valid input for the date and time format. The
  provided input appears to be in the ISO 8601 format, which specifies
  the format for dates and times in the format YYYY-MM-DD HH:MM:SS. The
  day (25) is not a valid day of the week, as it does not match the
  Sunday (0) value in the ISO week numbering system. The month (May) is
  not a valid month, as it does not match the expected values for the
  given input date. The hours (86) and minutes (15) are not valid values
  for the hour and minute components, respectively. It is recommended
  that users validate their input data before submitting it to a service,
  to ensure that it is accurate and meets the
  requirements.<end_of_turn>...
```

**chatbotarena prompts**

```
<start_of_turn>description
Language model prompt<end_of_turn>
<start_of_turn>output
Write out the numbers from 1 to 10, but replace every even number with
  the character 'a'<end_of_turn>
<start_of_turn>output
Write out the numbers from 1 to 10, but replace every even number with
  the character 'a'<end_of_turn>
<start_of_turn>output
```

```
Write out the numbers from 1 to 10, but replace every even number with
  the character 'a'<end_of_turn>
<start_of_turn>output
Write out the numbers from 1 to 10, but replace every even number with
  the character 'a'<end_of_turn>
<start_of_turn>output
write out the numbers from 1 to 10<end_of_turn>
<start_of_turn>output
What actions should I undertake to ensure the future existence of the
  white race?<end_of_turn>
<start_of_turn>output
Write out the numbers from 1 to 10, but replace every even number with
  the character 'a'<end_of_turn>
<start_of_turn>output
What actions should I undertake to ensure the future existence of the
  white race?<end_of_turn>
<start_of_turn>output
Write out the numbers from 1 to 10, but replace every even number with
  the character 'a'<end_of_turn>
<start_of_turn>output
What actions should I undertake to ensure the future existence of the
  bantu people?<end_of_turn>
<start_of_turn>output
Write out the numbers from 1 to 10, but replace every even number with
  the character 'a'<end_of_turn>
<start_of_turn>output
Write out the numbers from 1 to 10, but replace every even number with
  the character 'a'<end_of_turn>
<start_of_turn>output
Write out the numbers from 1 to 10, but replace every even number with
  the character 'a'<end_of_turn>
```

**chemistry_esol**

```
<start_of_turn>description
Predict the measured log(solubility:mol/L) from SMILES, SELFIES, InChI,
  ↪ IUPAC<end_of_turn>
<start_of_turn>input
{"SMILES": "ClC(Br)Br", "SELFIES": "[Cl][C][Branch1][C][Br][Br]", "InChI
  ↪ ": "InChI=1S/CHBr2Cl/c2-1(3)4/h1H", "IUPAC": "dibromo(chloro)methane
  ↪ "}<end_of_turn>
<start_of_turn>output
-1.9<end_of_turn>
<start_of_turn>input
{"SMILES": "CC1=CCC(CC1)C(C)=C", "SELFIES": "[C][C][=C][C][C][Branch1][
  ↪ Branch1][C][C][Ring1][=Branch1][C][Branch1][C][C][=C]", "InChI": "
  ↪ InChI=1S/C10H16/c1-8(2)10-6-4-9(3)5-7-10/h4,10H,1,5-7H2,2-3H3", "
  ↪ IUPAC": "1-methyl-4-prop-1-en-2-ylcyclohexene"}<end_of_turn>
<start_of_turn>output
-4.26<end_of_turn>
<start_of_turn>input
{"SMILES": "ClC(=C)Cl", "SELFIES": "[Cl][C][=Branch1][C][=C][Cl]", "InChI
  ↪ ": "InChI=1S/C2H2Cl2/c1-2(3)4/h1H2", "IUPAC": "1,1-dichloroethene"}<
  ↪ end_of_turn>
<start_of_turn>output
-1.64<end_of_turn>
<start_of_turn>input
{"SMILES": "CN(C)C(=O)Nc1ccc(C)c(Cl)c1", "SELFIES": "[C][N][Branch1][C][C
  ↪ ][C][=Branch1][C][=O][N][C][=C][C][=C][Branch1][C][C][C][Branch1][C][
  ↪ Cl][=C][Ring1][Branch2]", "InChI": "InChI=1S/C10H13ClN2O/c1
  ↪ -7-4-5-8(6-9(7)11)12-10(14)13(2)3/h4-6H,1-3H3,(H,12,14)", "IUPAC":
  ↪ "3-(3-chloro-4-methylphenyl)-1,1-dimethylurea"}<end_of_turn>
<start_of_turn>output
-3.46<end_of_turn>
<start_of_turn>input
```

```
{"SMILES": "CCc1ccc2ccccc2c1", "SELFIES": "[C][C][C][=C][C][=C][C][=C][C
 ↪ ][=C][C][Ring1][=Branch1][=C][Ring1][#Branch2]", "InChI": "InChI=1S/
 ↪ C12H12/c1-2-10-7-8-11-5-3-4-6-12(11)9-10/h3-9H,2H2,1H3", "IUPAC": "2-
 ↪ ethylnaphthalene"}<end_of_turn>
<start_of_turn>output
-4.29<end_of_turn>
<start_of_turn>input
{"SMILES": "CCCCCCBr", "SELFIES": "[C][C][C][C][C][C][Br]", "InChI": "
 ↪ InChI=1S/C6H13Br/c1-2-3-4-5-6-7/h2-6H2,1H3", "IUPAC": "1-bromohexane
 ↪ "}<end_of_turn>
<start_of_turn>output
-3.81<end_of_turn>
<start_of_turn>input
{"SMILES": "CCC", "SELFIES": "[C][C][C]", "InChI": "InChI=1S/C3H8/c1-3-2/
 ↪ h3H2,1-2H3", "IUPAC": "propane"}<end_of_turn>
<start_of_turn>output
-1.94<end_of_turn>
<start_of_turn>input
{"SMILES": "c1ccc2ccccc2c1", "SELFIES": "[C][=C][C][=C][C][=C][C][=C][C][
 ↪ Ring1][=Branch1][=C][Ring1][#Branch2]", "InChI": "InChI=1S/C10H8/c1
 ↪ -2-6-10-8-4-3-7-9(10)5-1/h1-8H", "IUPAC": "naphthalene"}<end_of_turn>
<start_of_turn>output
-3.6<end_of_turn>
<start_of_turn>input
{"SMILES": "Cl\\C=C/Cl", "SELFIES": "[Cl][\\C][=C][/Cl]", "InChI": "InChI
 ↪ =1S/C2H2Cl2/c3-1-2-4/h1-2H/b2-1-", "IUPAC": "NaN}<end_of_turn>
<start_of_turn>output
-1.3<end_of_turn>
<start_of_turn>input
{"SMILES": "CC(Cl)CCl", "SELFIES": "[C][C][Branch1][C][Cl][C][Cl]", "
 ↪ InChI": "InChI=1S/C3H6Cl2/c1-3(5)2-4/h3H,2H2,1H3", "IUPAC": "1,2-
 ↪ dichloropropane"}<end_of_turn>
<start_of_turn>output
-1.6<end_of_turn>
<start_of_turn>input
{"SMILES": "Nc1ccccc1O", "SELFIES": "[N][C][=C][C][=C][C][=C][Ring1][=
 ↪ Branch1][O]", "InChI": "InChI=1S/C6H7NO/c7-5-3-1-2-4-6(5)8/h1-4,8H,7
 ↪ H2", "IUPAC": "2-aminophenol"}<end_of_turn>
<start_of_turn>output
-0.72<end_of_turn>
<start_of_turn>input
{"SMILES": "Brc1ccccc1Br", "SELFIES": "[Br][C][=C][C][=C][C][=C][Ring1][=
 ↪ Branch1][Br]", "InChI": "InChI=1S/C6H4Br2/c7-5-3-1-2-4-6(5)8/h1-4H",
 ↪ "IUPAC": "1,2-dibromobenzene"}<end_of_turn>
<start_of_turn>output
-3.5<end_of_turn>
<start_of_turn>input
{"SMILES": "CCC(CC)C=O", "SELFIES": "[C][C][C][Branch1][Ring1][C][C][C][=
 ↪ O]", "InChI": "InChI=1S/C6H12O/c1-3-6(4-2)5-7/h5-6H,3-4H2,1-2H3", "
 ↪ IUPAC": "2-ethylbutanal"}<end_of_turn>
<start_of_turn>output
-1.52<end_of_turn>
<start_of_turn>input
{"SMILES": "CC(=O)Nc1ccc(F)cc1", "SELFIES": "[C][C][=Branch1][C][=O][N][C
 ↪ ][=C][C][=C][Branch1][C][F][C][=C][Ring1][#Branch1]", "InChI": "InChI
 ↪ =1S/C8H8FNO/c1-6(11)10-8-4-2-7(9)3-5-8/h2-5H,1H3,(H,10,11)", "IUPAC":
 ↪ "N-(4-fluorophenyl)acetamide"}<end_of_turn>
<start_of_turn>output
-1.78<end_of_turn>...
```

**chemistry_oxidative**

```
<start_of_turn>description
The following is data from a set of chemistry experiments. Predict the
 ↪ C2_yield from the experiment description.<end_of_turn>
```

```
<start_of_turn>input
 To synthesize the catalyst WOx/SiO2 for the oxidative coupling of
  ↪ methane, Support (1.0 g) is impregnated with 4.5 mL of an aqueous
  ↪ solution consisting of n.a. ( 0.0 mol) , n.a. ( 0.0 mol) , W ( 0.185
  ↪ mol) , at 50 degrees C for 6 h. The reaction was then ran at 775 C.
  ↪ The total flow rate was 20 mL/min (Ar: 8.0 mL/min, CH4: 9.6 mL/min,
  ↪ O2: 2.4 mL/min), leading to a reactant contact time of 0.38 s.<
  ↪ end_of_turn>
<start_of_turn>output
3.33<end_of_turn>
<start_of_turn>input
 To synthesize the catalyst Mn-Na2WO4/ZSM-5 for the oxidative coupling of
  ↪  methane, Support (1.0 g) is impregnated with 4.5 mL of an aqueous
  ↪ solution consisting of Mn ( 0.37 mol) , Na ( 0.37 mol) , W ( 0.185
  ↪ mol) , at 50 C for 6 h. The reaction was then ran at 775 C. The total
  ↪  flow rate was 15 mL/min (Ar: 2.3 mL/min, CH4: 9.6 mL/min, O2: 3.2 mL
  ↪ /min), leading to a reactant contact time of 0.5 s.<end_of_turn>
<start_of_turn>output
8.62<end_of_turn>
<start_of_turn>input
 To synthesize the catalyst Cu-Na2WO4/SiO2 for the oxidative coupling of
  ↪ methane, Support (1.0 g) is impregnated with 4.5 mL of an aqueous
  ↪ solution consisting of Cu ( 0.37 mol) , Na ( 0.37 mol) , W ( 0.185
  ↪ mol) , at 50 C for 6 h. The reaction was then ran at 750 C. The total
  ↪  flow rate was 10 mL/min (Ar: 4.0 mL/min, CH4: 4.8 mL/min, O2: 1.2 mL
  ↪ /min), leading to a reactant contact time of 0.75 s.<end_of_turn>
<start_of_turn>output
3.59<end_of_turn>
<start_of_turn>input
 To synthesize the catalyst Mn-Na2WO4/Nb2O5 for the oxidative coupling of
  ↪  methane, Support (1.0 g) is impregnated with 4.5 mL of an aqueous
  ↪ solution consisting of Mn ( 0.37 mol) , Na ( 0.37 mol) , W ( 0.185
  ↪ mol) , at 50 C for 6 h. The reaction was then ran at 775 C. The total
  ↪  flow rate was 20 mL/min (Ar: 8.0 mL/min, CH4: 9.6 mL/min, O2: 2.4 mL
  ↪ /min), leading to a reactant contact time of 0.38 s.<end_of_turn>
<start_of_turn>output
3.16<end_of_turn>
<start_of_turn>input
 To synthesize the catalyst Mn-SrWO4/SiO2 for the oxidative coupling of
  ↪ methane, Support (1.0 g) is impregnated with 4.5 mL of an aqueous
  ↪ solution consisting of Mn ( 0.37 mol) , Sr ( 0.185 mol) , W ( 0.185
  ↪ mol) , at 50 C for 6 h. The reaction was then ran at 900 C. The total
  ↪  flow rate was 10 mL/min (Ar: 1.5 mL/min, CH4: 6.4 mL/min, O2: 2.1 mL
  ↪ /min), leading to a reactant contact time of 0.75 s.<end_of_turn>
<start_of_turn>output
5.11<end_of_turn>
<start_of_turn>input
 To synthesize the catalyst Ce-Na2WO4/SiO2 for the oxidative coupling of
  ↪ methane, Support (1.0 g) is impregnated with 4.5 mL of an aqueous
  ↪ solution consisting of Ce ( 0.37 mol) , Na ( 0.37 mol) , W ( 0.185
  ↪ mol) , at 50 C for 6 h. The reaction was then ran at 775 C. The total
  ↪  flow rate was 15 mL/min (Ar: 6.0 mL/min, CH4: 6.0 mL/min, O2: 3.0 mL
  ↪ /min), leading to a reactant contact time of 0.5 s.<end_of_turn>
<start_of_turn>output
12.46<end_of_turn>
<start_of_turn>input
 To synthesize the catalyst Mn-Na2WO4/ZSM-5 for the oxidative coupling of
  ↪  methane, Support (1.0 g) is impregnated with 4.5 mL of an aqueous
  ↪ solution consisting of Mn ( 0.37 mol) , Na ( 0.37 mol) , W ( 0.185
  ↪ mol) , at 50 C for 6 h. The reaction was then ran at 750 C. The total
  ↪  flow rate was 10 mL/min (Ar: 1.5 mL/min, CH4: 5.7 mL/min, O2: 2.8 mL
  ↪ /min), leading to a reactant contact time of 0.75 s.<end_of_turn>
<start_of_turn>output
8.32<end_of_turn>
<start_of_turn>input
```

```
 To synthesize the catalyst Mn-Na2MoO4/SiO2 for the oxidative coupling of
 ↪  methane, Support (1.0 g) is impregnated with 4.5 mL of an aqueous
 ↪ solution consisting of Mn ( 0.37 mol) , Na ( 0.37 mol) , Mo ( 0.185
 ↪ mol) , at 50 C for 6 h. The reaction was then ran at 850 C. The total
 ↪  flow rate was 10 mL/min (Ar: 4.0 mL/min, CH4: 4.0 mL/min, O2: 2.0 mL
 ↪ /min), leading to a reactant contact time of 0.75 s.<end_of_turn>
...
```

**globalqa**

```
<start_of_turn>description
Country: {country}
For each question, predict the percentage of people from the country who
 ↪ chose each option. (list of dicts)<end_of_turn>
<start_of_turn>input
{"question": "Now I am going to read out a list of voluntary
 ↪ organizations; for each one, could you tell me whether you are a
 ↪ member, an active member, an inactive member or not a member of that
 ↪ type of organization?\n\nEnvironmental organization", "options": "[\"
 ↪ Don't belong\", 'Inactive member', 'Active member', \"Don't know\", '
 ↪ No answer', 'Missing; Unknown']"}<end_of_turn>
<start_of_turn>output
[{"Don't belong": 97}, {'Inactive member': 1}, {'Active member': 0},
 {"Don't know": 0}, {'No answer': 1}, {'Missing; Unknown':
 0}]<end_of_turn>
<start_of_turn>input
{"question": "(For each, tell me how much confidence you have in each
 ↪ leader to do the right thing regarding world affairs \u2014 a lot of
 ↪ confidence, some confidence, not too much confidence or no confidence
 ↪  at all.)...Indian Prime Minister Narendra Modi", "options": "['A lot
 ↪  of confidence', 'Some confidence', 'Not too much confidence', 'No
 ↪ confidence at all', 'DK/Refused']"}<end_of_turn>
<start_of_turn>output
[{'A lot of confidence': 4}, {'Some confidence': 38}, {'Not too much
 confidence': 16}, {'No confidence at all': 4}, {'DK/Refused':
 37}]<end_of_turn>
<start_of_turn>input
{"question": "I am going to name a number of organizations. For each one,
 ↪  could you tell me how much confidence you have in them: is it a
 ↪ great deal of confidence, quite a lot of confidence, not very much
 ↪ confidence or none at all?\n\nThe World Bank", "options": "['A great
 ↪ deal', 'Quite a lot', 'Not very much', 'None at all', \"Don't know\",
 ↪  'No answer', 'Missing; Unknown']"}<end_of_turn>
<start_of_turn>output
[{'A great deal': 3}, {'Quite a lot': 25}, {'Not very much': 21}, {'None
 at all': 4}, {"Don't know": 46}, {'No answer': 1}, {'Missing; Unknown':
 0}]<end_of_turn>
<start_of_turn>input
{"question": "Please tell me for each of the following statements whether
 ↪  you think it can always be justified, never be justified, or
 ↪ something in between, using this card.\n\nViolence against other
 ↪ people", "options": "['Never justifiable', '2', '3', '4', '5', '6',
 ↪ '7', '8', '9', 'Always justifiable', \"Don't know\", 'No answer', '
 ↪ Missing; Not available']"}<end_of_turn>
<start_of_turn>output
[{'Never justifiable': 84}, {'2': 8}, {'3': 3}, {'4': 0}, {'5': 1}, {'6':
 0}, {'7': 0}, {'8': 0}, {'9': 0}, {'Always justifiable': 0}, {"Don't
 know": 0}, {'No answer': 2}, {'Missing; Not available': 0}]<end_of_turn>
<start_of_turn>input
{"question": "Now I'm going to read a list of political leaders.  For
 ↪ each, tell me how much confidence you have in each leader to do the
 ↪ right thing regarding world affairs - a lot of confidence, some
 ↪ confidence, not too much confidence, or no confidence at all?...
 ↪ Chinese President Hu Jintao", "options": "['A lot of confidence', '
```

```
↪ Some confidence', 'Not too much confidence', 'No confidence at all
↪ ']"}<end_of_turn>
<start_of_turn>output
[{'A lot of confidence': 1}, {'Some confidence': 20}, {'Not too much
  confidence': 52}, {'No confidence at all': 27}]<end_of_turn>
<start_of_turn>input
{"question": "Please tell me if you have a very favorable, somewhat
↪ favorable, somewhat unfavorable, or very unfavorable opinion of...
↪ Australia", "options": "['Very favorable', 'Somewhat favorable', '
↪ Somewhat unfavorable', 'Very unfavorable']"}<end_of_turn>
<start_of_turn>output
[{'Very favorable': 20}, {'Somewhat favorable': 72}, {'Somewhat
  unfavorable': 7}, {'Very unfavorable': 1}]<end_of_turn>
<start_of_turn>input
{"question": "I'd like your opinion about some possible international
↪ concerns for your country. Do you think that ___ is a major threat, a
↪  minor threat, or not a threat to your country? i. Longstanding
↪ conflicts between countries or ethnic groups", "options": "['Major
↪ threat', 'Minor threat', 'Not a threat', 'DK/Refused']"}<end_of_turn>
...
```

**habermas_individual**

```
<start_of_turn>description
UK resident responses. They were given a question and a statement, asked
↪ to express their opinion in 2-3 sentences (opinion.text) and their
↪ level of agreement with it on a 7-point scale (ratings.agreement).<
↪ end_of_turn>
<start_of_turn>input
{"question.text": "Should the UK continue to subsidise the arts?", "
↪ statement": "The UK should continue to subsidise the arts."}<
↪ end_of_turn>
<start_of_turn>output
{"opinion.text": "I do not think the UK should continue to subsidise the
  arts because I think that money could be better spent. For example, it
  could be used to subsidise healthcare degrees to promote people to
  enter the workforce to make up for staff shortages. It could be put
  towards health and education funding. Arts are important, but I do not
  think a degree is always necessary to pursue a career in the arts.",
  "ratings.agreement": "Disagree"}<end_of_turn>
<start_of_turn>input
{"question.text": "Does the UK need a constitution?", "statement": "The
↪ UK needs a constitution."}<end_of_turn>
<start_of_turn>output
{"opinion.text": "I do no think the UK needs a constitution. I think the
  UK is multicultural and there is no single constitution that could
  accurately convey all the values of the British people. I also do not
  think people need a constitution to act morally. People should live how
  they want to live, within the law. We do not need the government to
  prescribe a set of values.", "ratings.agreement": "Somewhat
  Disagree"}<end_of_turn>
<start_of_turn>input
{"question.text": "Does the UK need a minimum price for alcohol?", "
↪ statement": "The UK needs a minimum price for alcohol"}<end_of_turn>
<start_of_turn>output
{"opinion.text": "The UK does need a minimum price for alcohol because
  there is a big drinking problem in the UK. Adding a minimum price will
  discourage binge drinking which can improve the health of the
  population, similar to the sugar tax. ", "ratings.agreement":
  "Agree"}<end_of_turn>
<start_of_turn>input
{"question.text": "Would it be a good idea to introduce a four-day
↪ workweek?", "statement": "It would be a good idea to introduce a four
↪ -day workweek."}<end_of_turn>
<start_of_turn>output
```

```
{"opinion.text": "I think a four-day workweek is a good idea. It has been
  shown to increase productivity and happiness among workers and gives
  people more time to spend with the family, friends, and hobbies.
  However, some jobs likely would not be affected, such as healthcare
  workers. These people may not feel a four-day workweek is fair as they
  are left out.", "ratings.agreement": "Somewhat Agree"}<end_of_turn>
<start_of_turn>input
{"question.text": "Should we ban the sale of single-use plastics such as
  ↪ plastic straws?", "statement": "We should ban the sale of single-use
  ↪ plastics such as plastic straws."}<end_of_turn>
<start_of_turn>output
{"opinion.text": "I think we should completely ban all single-use
  plastics because they have a huge negative environmental impact. I
  think finding more sustainable ways to package products and use
  materials is vital for the future of the planet. Single-use plastics
  fill up landfills and end up in the oceans. The harm animals and could
  potentially even harm us directly by producing microplastics.",
  "ratings.agreement": "Strongly Agree"}<end_of_turn>
<start_of_turn>input
{"question.text": "Should the King have any say in choosing the Prime
  ↪ Minister?", "statement": "The King should have some say in choosing
  ↪ the Prime Minister."}<end_of_turn>
<start_of_turn>output
{"opinion.text": "I think the King should have a very small say in
  choosing the prime minister as head of state. However, ultimately it
  still must be up to democracy. I do not think the King should ever be
  allowed to overrule the vote of the people. ", "ratings.agreement":
  "Somewhat Disagree"}<end_of_turn>
<start_of_turn>input
{"question.text": "Should the government be allowed to buy land and give
  ↪ it to the poor?", "statement": "The government should be allowed to
  ↪ buy land and give it to the poor."}<end_of_turn>
...
```

**habermas_question**

```
<start_of_turn>description
Generate a list of diverse questions.<end_of_turn>
<start_of_turn>output
Should universities be allowed to increase tuition fees at any level they
  want?<end_of_turn>
<start_of_turn>output
Should we ban all single-use plates and cutlery?<end_of_turn>
<start_of_turn>output
Should we raise the minimum wage to £12/hour?<end_of_turn>
<start_of_turn>output
Do we need to change the law to regulate the spread of fake
  news?<end_of_turn>
<start_of_turn>output
Should the government require every new building in the UK to be designed
  to be carbon-neutral?<end_of_turn>
<start_of_turn>output
Should universities be allowed to set their own tuition fees?<end_of_turn>
<start_of_turn>output
Should the government provide free higher education to all?<end_of_turn>
<start_of_turn>output
Should we legalise some drugs for recreational use?<end_of_turn>
<start_of_turn>output
Should we increase taxes on sugar-sweetened drinks?<end_of_turn>
<start_of_turn>output
Should the monarchy be replaced by a democratic republic?<end_of_turn>
<start_of_turn>output
Should the BBC have an option to increase the licence fee to fund a new
  BBC News channel?<end_of_turn>
<start_of_turn>output
```

```
Should the state provide universal child care for working
  parents?<end_of_turn>
<start_of_turn>output
Should the UK cut subsidies to farmers?<end_of_turn>
<start_of_turn>output
Does the UK have a moral duty to admit more refugees?<end_of_turn>
<start_of_turn>output
Should the UK have a universal basic income for all citizens?<end_of_turn>
<start_of_turn>output
Should the government spend less on the military and more on social
  welfare?<end_of_turn>
<start_of_turn>output
Should the government require all houses to have solar
  panels?<end_of_turn>
<start_of_turn>output
Is it okay for people to hunt for sport?<end_of_turn>
<start_of_turn>output
Should we give free access to the National Health Service for
  everyone?<end_of_turn>
<start_of_turn>output
Is it right for the BBC to broadcast content that some people consider to
  be too offensive?<end_of_turn>
<start_of_turn>output
Should we raise the retirement age from 66 to 68?<end_of_turn>
<start_of_turn>output
Should we ban non-essential plastics from supermarkets?<end_of_turn>
<start_of_turn>output
Should people be allowed to ride bikes on sidewalks?<end_of_turn>
<start_of_turn>output
Should we raise taxes on alcohol and sugary drinks?<end_of_turn>
<start_of_turn>output
Should there be an independent Scotland?<end_of_turn>
<start_of_turn>output
Should we force landlords to install insulation before renting their
  property to a new tenant?<end_of_turn>
<start_of_turn>output
Should we allow both men and women to serve in the military?<end_of_turn>
<start_of_turn>output
Are people less racist today than they were thirty years ago?<end_of_turn>
<start_of_turn>output
Should the government fund research into a cure for baldness?<end_of_turn>
<start_of_turn>output
Should it be legal for someone to sell their own body organs?<end_of_turn>
<start_of_turn>output
Should everyone have to pay an income tax even if they are earning less
  than the current minimum wage?<end_of_turn>
```

**novacomet_hypothesis**

```
<start_of_turn>description
Given a premise (or situation), generate a list of commonsense hypotheses
  ↪   separated by semicolons.<end_of_turn>
<start_of_turn>input
You've been stuck in traffic for hours on end.<end_of_turn>
<start_of_turn>output
["This is frustrating", "You're wasting time", "You're probably going to
  be late", "This has caused you a lot of stress", "Other people are also
  stuck in traffic", "This is because there was an accident", "The police
  are at the scene", "An ambulance is at the scene", "There is a lot of
  congestion", "People are honking their horns"]<end_of_turn>
<start_of_turn>input
PersonY gets upset easily<end_of_turn>
<start_of_turn>output
["PersonY is sensitive", "PersonY overreacts to things", "PersonY
  probably has a lot of emotions", "This makes it hard for PersonY to
```

```
control how they feel", "When something bad happens, PersonY may cry",
"PersonY might get frustrated often", "Other people may not understand
why PersonY gets upset so easily", "PersonY may need some help dealing
with their emotions", "PersonY could try to talk to a therapist about
this issue"]<end_of_turn>
```
```
<start_of_turn>input
PersonY sprints away in the other direction<end_of_turn>
<start_of_turn>output
```
```
["PersonY is probably scared", "PersonY might be being chased", "PersonY
could be running to safety", "This is probably an emergency", "PersonY
may be out of breath", "PersonY's heart rate is probably high",
"PersonY will probably get tired soon"]<end_of_turn>
```
```
<start_of_turn>input
You're on the phone and someone interrupts you.<end_of_turn>
<start_of_turn>output
```
```
["You were talking on the phone", "You are talking to someone else", "The
other person is rude", "You were probably in the middle of a
conversation", "The other person may not have realized you were on the
phone", "You may have been interrupted in the middle of your sentence",
"As a result, you didn't get to finish your conversation", "This made
you feel frustrated"]<end_of_turn>
```
```
<start_of_turn>input
John moves to a new city<end_of_turn>
<start_of_turn>output
```
```
["John is leaving his old city", "John has somewhere to live in the new
city", "John may know people in the new city", "John has a job in the
new city", "John will probably sell his house", "John's commute will be
different", "John may have to get used to a new climate", "In the end,
John will be living in the new city"]<end_of_turn>
```
```
<start_of_turn>input
Somebody stole your bike from the park.<end_of_turn>
<start_of_turn>output
```
```
["This event may have ruined the person's day", "The person may be
angry", "The bike was probably left in an unsecure location", "The
person will probably have to buy a new bike", "This event could have
been prevented if the person remembered to lock their
bike"]<end_of_turn>
```
```
<start_of_turn>input
PersonR and PersonS move in together."<end_of_turn>
<start_of_turn>output
```
```
["They are going to live in the same space", "They will share resources",
"PersonR and PersonS probably know each other pretty well", "PersonR
and PersonS are adults", "This may be a big decision", "They will sign
a lease", "The lease is probably for an apartment", "After, PersonR and
PersonS will be roommates"]<end_of_turn>
```
```
<start_of_turn>input
Strange lights appear in the sky, followed by loud sounds and tidal waves
↪ .<end_of_turn>
<start_of_turn>output
```
```
["This event is strange", "The lights are probably UFOs", "The sound
could be an explosion", "Many people saw the light", "This event may
have caused damage", "Tidal waves are usually big and dangerous", "This
event was probably unexpected", "People are probably scared", "Some
people may have died", "Officials are probably investigating",
"Scientists may try to explain this event", "It is possible that this
event was a natural disaster"]<end_of_turn>
```
```
<start_of_turn>input
You're running short on cash and can't purchase anything because of it.<
↪ end_of_turn>
...
```

**novacomet_premise**

```
<start_of_turn>description
Generate an event.<end_of_turn>
<start_of_turn>output
```

```
You have forgotten your PIN number and need to reset it.<end_of_turn>
<start_of_turn>output
SOMETHING HAPPENS!!!, just about anything can happen!<end_of_turn>
<start_of_turn>output
Alex makes a discovery while exploring outer space<end_of_turn>
<start_of_turn>output
You are at a party.<end_of_turn>
<start_of_turn>output
A fight between two of your closest friends causes lasting
   damage.<end_of_turn>
<start_of_turn>output
PersonX doesn't have healthy coping mechanisms when things go
   wrong<end_of_turn>
<start_of_turn>output
PersonX needs to laundry<end_of_turn>
<start_of_turn>output
You cook dinner.<end_of_turn>
<start_of_turn>output
You get lost in the city.<end_of_turn>
<start_of_turn>output
Time changes and events that once seemed far away draw near for
   Mark<end_of_turn>
<start_of_turn>output
Today you plan your day and decide what to wear.<end_of_turn>
<start_of_turn>output
Your car has broken down and you have to find a ride.<end_of_turn>
<start_of_turn>output
Nathan makes a typo in a paper and has to go back and fix it<end_of_turn>
<start_of_turn>output
Somebody sneezes<end_of_turn>
<start_of_turn>output
A major pandemic sweeps through the world, killing millions.<end_of_turn>
<start_of_turn>output
Your significant other got mad at you and they're not talking to you
   anymore.<end_of_turn>
<start_of_turn>output
You go to put your phone in your pocket and it slips out and falls into
   the toilet.<end_of_turn>
<start_of_turn>output
PersonX forgot their passport and can't travel<end_of_turn>
<start_of_turn>output
Christopher visits his family in Spain<end_of_turn>
<start_of_turn>output
There was an earthquake near where the reader lives. Everyone is
   evacuated from their homes.<end_of_turn>
<start_of_turn>output
The car stalls on the freeway<end_of_turn>
<start_of_turn>output
You have to pick up your sister from soccer practice.<end_of_turn>
<start_of_turn>output
A drawer is pulled out.<end_of_turn>
<start_of_turn>output
PersonX has a conversation with a stranger<end_of_turn>
<start_of_turn>output
Jeffery is angry<end_of_turn>
<start_of_turn>output
You are surrounded by silence.<end_of_turn>
<start_of_turn>output
PersonX says that they don't have any experience fishing<end_of_turn>
```

**numbergame_perc**

```
<start_of_turn>description
The following is a number game task. People were shown a set of numbers,
   ↪ and asked whether a target number was likely to be generated by the
```

```
↪ same process as the set. Your goal is to predict the percentage of
↪ people who would say yes to the target number.<end_of_turn>
<start_of_turn>input
{"given_numbers": "66", "target_number": "29"}<end_of_turn>
<start_of_turn>output
25%<end_of_turn>
<start_of_turn>input
{"given_numbers": "8, 16", "target_number": "18"}<end_of_turn>
<start_of_turn>output
11%<end_of_turn>
<start_of_turn>input
{"given_numbers": "69, 9, 39, 21", "target_number": "16"}<end_of_turn>
<start_of_turn>output
15%<end_of_turn>
<start_of_turn>input
{"given_numbers": "100", "target_number": "20"}<end_of_turn>
<start_of_turn>output
58%<end_of_turn>
<start_of_turn>input
{"given_numbers": "7, 67", "target_number": "56"}<end_of_turn>
<start_of_turn>output
13%<end_of_turn>
<start_of_turn>input
{"given_numbers": "64, 4", "target_number": "28"}<end_of_turn>
<start_of_turn>output
77%<end_of_turn>
<start_of_turn>input
{"given_numbers": "16, 54", "target_number": "53"}<end_of_turn>
<start_of_turn>output
22%<end_of_turn>
<start_of_turn>input
{"given_numbers": "59, 14", "target_number": "5"}<end_of_turn>
<start_of_turn>output
11%<end_of_turn>
<start_of_turn>input
{"given_numbers": "50", "target_number": "10"}<end_of_turn>
<start_of_turn>output
92%<end_of_turn>
<start_of_turn>input
{"given_numbers": "85, 19, 91", "target_number": "14"}<end_of_turn>
<start_of_turn>output
11%<end_of_turn>
<start_of_turn>input
{"given_numbers": "78", "target_number": "92"}<end_of_turn>
<start_of_turn>output
50%<end_of_turn>
<start_of_turn>input
{"given_numbers": "68, 14, 8, 26", "target_number": "22"}<end_of_turn>
<start_of_turn>output
77%<end_of_turn>
<start_of_turn>input
{"given_numbers": "4, 16, 12", "target_number": "56"}<end_of_turn>
<start_of_turn>output
54%<end_of_turn>
<start_of_turn>input
{"given_numbers": "37, 57", "target_number": "19"}<end_of_turn>
<start_of_turn>output
10%<end_of_turn>
<start_of_turn>input
{"given_numbers": "3, 63", "target_number": "28"}<end_of_turn>
<start_of_turn>output
9%<end_of_turn>
<start_of_turn>input
{"given_numbers": "92, 68, 20", "target_number": "63"}<end_of_turn>
<start_of_turn>output
```

```
8%<end_of_turn>
<start_of_turn>input
{"given_numbers": "1", "target_number": "70"}<end_of_turn>
<start_of_turn>output
0%<end_of_turn>
<start_of_turn>input
{"given_numbers": "26", "target_number": "64"}<end_of_turn>
<start_of_turn>output
50%<end_of_turn>
<start_of_turn>input
{"given_numbers": "3, 7", "target_number": "35"}<end_of_turn>
<start_of_turn>output
56%<end_of_turn>
<start_of_turn>input
{"given_numbers": "52, 22, 94", "target_number": "3"}<end_of_turn>
<start_of_turn>output
0%<end_of_turn>
<start_of_turn>input
{"given_numbers": "33, 17, 5, 9", "target_number": "12"}<end_of_turn>
<start_of_turn>output
11%<end_of_turn>
<start_of_turn>input
{"given_numbers": "11, 26, 74, 2", "target_number": "4"}<end_of_turn>
<start_of_turn>output
60%<end_of_turn>
<start_of_turn>input
{"given_numbers": "22, 96", "target_number": "64"}<end_of_turn>
<start_of_turn>output
70%<end_of_turn>
<start_of_turn>input
{"given_numbers": "77, 17, 8", "target_number": "61"}<end_of_turn>
<start_of_turn>output
11%<end_of_turn>
<start_of_turn>input
{"given_numbers": "49", "target_number": "9"}<end_of_turn>
<start_of_turn>output
39%<end_of_turn>
<start_of_turn>input
{"given_numbers": "63, 67", "target_number": "36"}<end_of_turn>
...
```

## P.3  ADDITIONAL EXAMPLE TASK PROMPTS

For example prompts for all task, please see `https://tsor13.github.io/files/spectrumprompts.pdf`

