# OpenReview forum: "Spectrum Tuning: Post-Training for Distributional Coverage and In-Context Steerability"
_ICLR.cc/2026/Conference — ICLR 2026 Poster_

### Official Review · Reviewer_nA29 · 2025-10-17

**Soundness:** 3
**Presentation:** 1
**Contribution:** 3
**Rating:** 6
**Confidence:** 3

**Summary:**

The paper investigates how the post-training of models (through instruction-tuning) negatively affects a specific set of desiderata, such as the ability of the in-context examples to steer the large language model towards a specific perspective, or diversity in the outputs for tasks with many valid responses. The authors prepare a dataset of tasks, called Spectrum Suite, to evaluate these desiderata and compare them between base and instruction-tuned models, finding that instruction-tuning breaks the steerability of LLMs for the generation tasks (while increasing performance for classification tasks or tasks with one/few specific valid answers). Building on this dataset, the authors introduce Spectrum Tuning, a post-training approach that provides the benefit of instruction-tuning withouth breaking the LLM desiderata (i.e., steerability and diversity of outputs).

**Strengths:**

The paper deals with important problem and I believe it can lead to many possibilities for intersting discussions and future work not only in post-training, but also in areas of interpretability.

The authors run an extensive set of experiments, providing a comprehensive evaluation on "failure" modes of instruction-tuned models, while proposing a solution for mitigating it. In addition, authors provide important resources (Spectrum Suite dataset).

**Weaknesses:**

**Paper structure/writing**

My biggest and only issue with the paper is how it is structured and the overall presentation. Overall, the paper is not really well put together -- although all the information is there, it is kind of "all over the place" and there are many explanations of motivation, connections between ideas, and details missing.

First of all, the abstract and introduction do not really introduce or motivate the problem and it is not really clear why it is important to deal with it -- only 2 sentences are dedicated to this (Liens 33-37) and afterwards the contributions and the description of the desiderata is introduced. I would suggest expanding the motivation in introduction and abstract as it would improve the paper quite a lot. In addition, I would suggest having a dedicated section for the Desiderata and providing more in-depth description or discussion on them. Similar problems are in the dataset and method description, which are quite short and deserve more detailed description and explaining the motivation behind them.

For the experiments section, the details for how the individual metrics are calculated are missing -- for example, there are lot of references to "yield" in Section 4, which is not explained and makes it more problematic to understand the results. Similar case is with "diversity" or "calibration" as it is not clear how these metrics were calculated. In addition, when performing Spectrum tuning and comparing with instruction-tuned models, it is not clear whether the evaluation is done on a completely different set of tasks or not.

Furthermore, almost all of the figures are quite small and full of information, making them hard to read and easily understand -- for example, in Figure 4 the legend (what the colours mean) is small and very well hidden.

(minor)

Lines 89-90 "see 1 above" -- it is not really clear what this is referring to; is it the "exhibit natural person-to-person variation" from the list at the beginning of the section?

Appendix D seems to be empty, or at least does not refer to any additional figures



**I acknowledge that many of the misunderstandings and difficulties understanding the paper are a result of limited space and can be easily fixed during rebuttal period, and will update my score if addressed**

**Questions:**

See weaknesses for details, but mostly relate to details:

How are the metrics, such as "yield", "calibration", and "diversity", calculated?

Is the Spectrum tuning trained on one set of tasks and then evaluated (and compared with instruction-tuning) on a separate, unseen set of tasks?

---

> ### Author Response · Authors · 2025-11-20
>
> We thank the reviewer for the thoughtful review. We especially appreciate the positive evaluation of the **“important problem” the work addresses while “proposing a solution” with an “important resource” and “extensive set of experiments”**. We also appreciate that the reviewer believes the work “can lead to **many possibilities for interesting discussions and future work** not only in post-training, but also in areas of interpretability.”
>
> We appreciate the reviewer’s detailed and constructive feedback regarding the presentation of the paper. **We have updated the paper to improve the presentation to the best of our ability within the length constraints.** Concretely, we have made the following changes:
> - RE: “expanding the motivation in introduction and abstract as it would improve the paper quite a lot”: We have **added a few sentences to expand the motivation** in the introduction and abstract.
> - RE: “a dedicated section for the Desiderata and providing more in-depth description or discussion”: We have **broken the desiderata into its own section, and added some additional discussion.** Additionally, we have added **illustrative supplementary figures** in Appendix A for the interested reader which should help to make the definitions even more clear.
> - RE: “the dataset and method description... deserve more detailed description and explaining the motivation behind them.”: We have added **more description and motivation** to these sections.
> - RE: yield / diversity / calibration metrics: we have **added additional discussion and equations in the text** to clarify how these are calculated.
> - RE: “when performing Spectrum tuning and comparing with instruction-tuned models, it is not clear whether the evaluation is done on a completely different set of tasks or not.”: All evaluation tasks are equivalent across Spectrum-tuned and instruction-tuned models. We have made this more clear in the text.
> - RE: “almost all of the figures are quite small and full of information, making them hard to read and easily understand”: We appreciate this feedback. While we understand where the reviewer is coming from, we have made an effort to make the figures larger to make them more legible, and have struggled to do so under space constraints without removing crucial text from the paper.
> - RE L89-90: Thank you for pointing out this confusing wording - your assumption was correct, and **we have changed the wording to reduce uncertainty.**
> - RE empty appendix D: Thank you for pointing this out! Yes, they were omitted by accident - we have **filled in the missing portion in the appendix.**
>
>
> Questions:
> - Q1. How are the metrics, such as "yield", "calibration", and "diversity", calculated?
> We have added in the explanation and equations into the paper where they are introduced in experiments. However, for you convenience, we also outline the natural language explanations here: yield is calculated by taking the set of valid generations, deduplicating, and counting the total #; calibration is calculated as the expected calibration error of the confidence of the model probabilities and the accuracy of the predictions, binned every decile; diversity is calculated as the # of unique valid generations (aka yield) / the # of total valid generations.
> - Q2. Is the Spectrum tuning trained on one set of tasks and then evaluated (and compared with instruction-tuning) on a separate, unseen set of tasks?
> Yes, precisely - Spectrum tuning utilizes a “Train” subset of tasks, and all evaluations (for pretrained, instruction-tuned, and spectrum-tuned models) are done on separate, unseen tasks, all sourced from different data sources to ensure generalization. **We have made this clearer in the paper.**
>
> Again, **we appreciate the reviewer’s positive assessment of the content of the paper and the useful feedback on the presentation.** We have attempted to improve the presentation to the best of our ability in the space constraints, and appreciate the reviewer’s understanding that the space constraints are quite limiting for a paper with such breadth and “extensive experiments”. **We hope that we have been able to improve the presentation and strengthen the paper.** Please let us know if you have any additional feedback or questions.

---

> > ### Comment · Reviewer_nA29 · 2025-11-20
> >
> > Thank you for the answer. I went through the paper and the changes addressed all of my concerns, significantly improving the readability of the paper.
> >
> > As a result, I am increasing my overall score and the presentation score.

---

> > > ### Author Response · Authors · 2025-11-28
> > >
> > > Dear Area Chair,
> > >
> > > We would like to note that we posted our rebuttal Nov 20, a week before the leakage issue, and that the same day, Reviewer nA29 came back and said that the changes **"addressed all of my concerns, significantly improving the readability of the paper."**
> > >
> > > As a result, the reviewer raised their score to:
> > > - Presentation: 3: good
> > > - **Rating: 8: accept, good paper (poster)**
> > >
> > > If it is possible to take this into consideration, we would greatly appreciate it.

---

### Official Review · Reviewer_HCf6 · 2025-10-30

**Soundness:** 2
**Presentation:** 3
**Contribution:** 2
**Rating:** 4
**Confidence:** 4

**Summary:**

The paper considers the scenario where questions in a task have no strict correct answers, and a model needs to give an answer that matches the distribution of a series of examples. The experiments show that instruction-tuned models perform worse than the pretrained models under this setting, and that a model specifically trained for this task performs better than pretrained models. They also find that the trained model achieves a better diversity-quality trade-off in out-of-domain questions that have multiple possible answers.

**Strengths:**

1. It is the first paper that considers the in-context steerability tasks
2. It provides insightful results about the degradation of IT models on this specific task
3. It provides a large-scale dataset for the task

**Weaknesses:**

1. Experiment setting seems over-complicated. It does not make sense to me to evaluate on the first few outputs, if the purpose is to evaluate in-context steerability.
2. Section 4 is missing an ablation study on temperature. It is claimed that the spectrum tuned models is Pareto optimum comparing to IT tuned models with temperature 1. But it would clearly make more sense if you use a couple of different temperature values for the IT models and see whether spectrum tuned model with a specific temperature is better than all of them, or plot a curve for both methods if the advantage is not that obvious.
3. Missing a baseline that instruction-tunes the model on the spectrum-suite training set. The paper only shows that a model, trained on the spectrum-suite training set with spectrum tuning, performs better on the spectrum-suite test set, comparing to models that are not trained on this dataset (PT and IT). This does not demonstrate the effectiveness of spectrum tuning.

**Questions:**

1. Are the instruction-tuned models taken from some existing checkpoints?
2. In the main experiment, how does the spectrum-tuned model perform under zero-shot settings? (i.e., we only evaluate on the first output)
3. What is the rationale in changing system/user/assistant to description/input/output?

---

> ### Author Response · Authors · 2025-11-20
>
> We appreciate that the reviewer found our paper to be novel in being the **“first paper that considers the in-context steerability tasks”** with a **“large-scale dataset** for the task”, along with **“insightful results about the degradation of IT models.”**
>
> We hope to clarify any concerns as follows:
> - W1: “Experiment setting seems over-complicated. It does not make sense to me to evaluate on the first few outputs, if the purpose is to evaluate in-context steerability.” - We are not totally sure we are interpreting the concern correctly, but believe that the reviewer is asking why we evaluate K-shot learning from 1-K for the In-Context Steerability experiment. We do so for several reasons: 1) We want to evaluate the models’ ability to perform both in the small K-shot setting (e.g., 1-shot, 2-shot), and in the many-shot setting, and 2) This allows us to get higher statistical power / more samples, as we can evaluate the accuracy of many samples in a single forward pass.
>
> - W2: “Section 4 is missing an ablation study on temperature.” This is a fantastic suggestion. **We have performed the suggested additional temperature experiment, sweeping all models over 8 temperature values.** Strengthening our paper, we find that 1) **Spectrum-Tuning improves the Pareto frontier,** and 2) even when allowing for selecting optimal temperature, **Spectrum-Tuned models offer the highest yield.** If you would like to see the results and plots yourself, please refer to the updated Section 4 and Appendix E. We are grateful to the reviewer for this great experiment suggestion, which has strengthened the papers’ claims.
>
> - W3: “Missing a baseline that instruction-tunes the model on the spectrum-suite training set.” - **We have performed additional experiments and added this baseline**, along with other additional baselines with alternative training methods on the spectrum suite training set. (See Table 4).  We find that **Spectrum Tuning is needed for maintaining strong performance on all desiderata, demonstrating the strength of the method.**
>
> Questions:
> - Q1: “Are the instruction-tuned models taken from some existing checkpoints?” - Yes, for the main set of experiments we use the provided instruction-tuned from the corresponding model providers (gemma-3-12b-it, Qwen3-14B, Llama-3.1-8B-Instruct). **We have made this more clear in the paper.**
> - Q2: “In the main experiment, how does the spectrum-tuned model perform under zero-shot settings? (i.e., we only evaluate on the first output)” - We chose to do 1-shot or more in the ICL experiment because we were interested in the models’ ability to steer to the new information at inference time, for which we need at least one demonstration example. Rather, we believe that a more proper zero-shot evaluation involves matching a target distribution over expected answers - in other words, our distributional alignment experiment. In that experiment, we find that the Spectrum-Tuned models outperform the instruction-tuned and pretrained model counterparts.
> - Q3: “What is the rationale in changing system/user/assistant to description/input/output?” - The rationale is largely due to the change in the nature of the data - instead of having data that is designed to be an order-dependent chat, it is instead order-independent draws from a target distribution (or, task). We chose to rename the roles to better match the nature of our data, and ensure that any downstream users of the Spectrum-Tuned model would prompt according to the description / input / output format.
>
> **We hope that we have been able to clarify some of your questions, and that the additional temperature experiment and baselines have improved the strength paper.** Please let us know if you have any other questions.

---

> > ### Author Response · Authors · 2025-11-28
> >
> > Hello, we just wanted to ping to see if the reviewer has had a chance to see our response, and to see which concerns we have or have not addressed and why. If you find that our response (including our additional temperature and baseline experiments) has addressed some concerns and strengthened the paper, we kindly ask that you consider raising your score. Thank you!

---

### Official Review · Reviewer_suLi · 2025-11-01

**Soundness:** 2
**Presentation:** 3
**Contribution:** 3
**Rating:** 4
**Confidence:** 4

**Summary:**

This paper argues that standard instruction-tuning improves instruction following but harms three properties that matter when many outputs are valid: (i) in-context steerability, (ii) diversity and coverage of the valid output space, and (iii) distributional alignment. It introduces SPECTRUM SUITE (>40 sources, >90 tasks) formatted as description/input/output sequences, and SPECTRUM TUNING, a simple SFT variant that incoperates task description and ICL examples into the training process. Empirically, instruction-tuned (IT) models are strong on validity but collapse in diversity, while pretrained (PT) models are diverse but under-valid; Spectrum-tuned models raise yield and often give Pareto-style gains on diversity–validity and improve calibration and JS-divergence vs. PT on several held-out datasets, without degrading standard capabilities.

**Strengths:**

1. This paper tackles fundamental, challenging questions in the LLM post-training stage, most notably the loss of diversity after instruction tuning.

2. The experimental evaluation is comprehensive, incorporating human judgments in key sections; several empirical observations are novel and likely of broad interest.

3. The writing and presentation are clear, crisp, and well-structured.

4. Spectrum Tuning delivers notable gains over instruction tuning. While the source of improvement (training paradigm vs. dataset) is not fully disentangled, the work meaningfully advances the SFT stage.

**Weaknesses:**

**General problems**

1. Technical and implementation details are insufficient. Please specify training configurations, the datasets used, and key statistics. **Most importantly, did instruction-tuning and Spectrum Tuning use the same dataset, or datasets of comparable scale?** Otherwise, the gains could be due to a better dataset rather than Spectrum Tuning itself.

2. While the paper addresses important problems and paints a broad picture, many of the issues are largely orthogonal. A unified, principled analysis is missing to explain why these seemingly orthogonal issues can be resolved by a single approach.

**In-context learning**

1. ICL provides training-free adaptation to downstream tasks, often to enforce output format without SFT. If the model is already fine-tuned for the target task, is ICL still necessary? A fine-tuned model should already know the required format and task specifics. I recommend authors to look for more justifications (through existing studies perhaps) for this aspect.

2. The baseline results are puzzling: the instruction-tuned LLM consistently underperforms the raw pre-trained model. This contradicts prior studies and common community expectations.

**Diversity and Space Coverage**

3. The diversity concerns of conventional SFT are valid, but substantial prior work has addressed this area. The paper does not adequately cite, discuss, or compare with approaches such as rejection fine-tuning (RFT) [1] and entropic distribution matching [2], among others.

**Distributional alignment**

4. This section reports results without explaining why Spectrum Tuning improves distributional alignment. Is the effect due to broader data coverage, the addition of task descriptions and ICL examples, or something else? If the latter, why would the instruction-tuned model’s probability distribution be spikier than Spectrum Tuning’s?

[1] Scaling Relationship on Learning Mathematical Reasoning with Large Language Models

[2] Entropic distribution matching for supervised fine-tuning of LLMs: Less overfitting and better diversity, ICLR 2025.

**Questions:**

1. It is not very straight forward for me why adding description embedding before the instruction can improve the diversity of the generation results.

---

> ### Author Response · Authors · 2025-11-20
>
> We appreciate that the reviewer found our work to **“tackle fundamental, challenging questions”** in post-training, have **“comprehensive” experiments** with results which “are **novel and likely of broad interest”**. We are also glad that the reviewer found the writing to be **“clear, crisp and structured”**, and that the work **“meaningfully advances the SFT stage.”**
>
> We hope to clarify some information, and **present additional experiments** to address the reviewer’s concerns.
>
> General Problems
> - W1: “Implementation details insufficient.”
>
> We have **added additional training details to Appendix B and will open-source all code** to ensure replicability and detailed implementation details. Additionally, we wish to clarify something: The instruction-tuned baseline models were not trained by us at all, but rather are the default “instruct” models released for Qwen / Gemma / Llama. We choose to use the instruct models as a baseline for the following reason: 1) it provides a strong baseline, as these models have been optimized on large-scale compute by the original large corporations, and 2) the format of turn-by-turn chat data (by instruction-tuning) and interchangeable output data (in Spectrum Suite) are fundamentally different. For example, with turn-by-turn chat data, the order matters - chats are order dependent. This is fundamentally different from interchangeable outputs in Spectrum Tuning. In other words, the inputs to Spectrum Tuning and the inputs to Instruction Tuning are different. So, the reviewer is correct that Spectrum Tuning is deeply related to the data in Spectrum Suite.
>
> However, we appreciated the reviewers’ suggestion about disentangling the effect of the method from the data. **We carry out additional experiments (Table 4)** using different training methods, including Instruction-SFT, on Spectrum Suite. We find that Spectrum Tuning offers advantages on In-Context Learning and Yield over Instruction-Tuning, better disentangling the method and the data as suggested by the reviewer. Additionally, **we perform Spectrum Tuning on capability-focused data**, and find that the steerability tasks in Spectrum Suite are important for eliciting performance gains, shining a light on the effect of the dataset itself.
>
> We hope that these two additional experiments help to alleviate the concern.
>
> - W2: “While the paper addresses important problems and paints a broad picture, many of the issues are largely orthogonal.” - We actually believe that they are all deeply related - they have to do with tasks where there is not a single, ground truth answer. We have attempted to add in **additional discussion in the body and Appendix B**
>
> ICL:
> - W1: Great question. ICL is necessary because we are performing inference on tasks that are UNSEEN in training - it is not finetuned for the target task. Instead, we do meta making -learning on one set of tasks at training, and evaluate on another set of unseen tasks at inference. Additionally, many of the tasks are few-shot, making ICL more effective than finetuning.
> - W2: **You’re absolutely right - it is a counterintuitive result, which is why we’re so excited about it!** It is true that Instruction-tuned models are generally thought of as having higher performance, but we believe that this is mainly because they are trained and evaluated on tasks with a single correct ground truth answer. We replicate this phenomenon on such tasks in Figure 3. However, it is specifically on these tasks that require steering to new distributions at inference time that instruction models do WORSE at than pretrained models, as we show in Figure 2.
> **We believe that this counterintuitive result would be of interest to the ICLR community.**
>
> Diversity and Coverage W1: We have added **additional discussion** to related work here and have integrated the references.
>
> Distributional alignment:
> - W1: This is a great question. While we hope to flesh out our understanding of this mechanism in future work, our best intuition is this - It largely has to do with the fact that 1) all training tasks involve interchangeable data and 2) we shuffle the data before training. This encourages the model to spread out its probability roughly in proportion the distribution of expected outputs. **We thank the reviewer for the insightful questions, and believe they may be useful to other readers. As such, we have added additional discussion of this to Appendix B.**
>
> Questions:
> Q: “It is not very straight forward for me why adding description embedding before the instruction can improve the diversity of the generation results.”
> Apologies, but this question is not immediately clear to us. We would be happy to answer any such question if you would like to rephrase it for us please.
>
> **We hope that we have been able to clarify some of your questions, and that our additional experiments and added paper discussion has improved the quality of the paper.** Please let us know if you have any other questions.

---

> > ### Comment · Reviewer_suLi · 2025-11-28
> >
> > I thank the authors for their detailed and comprehensive response. For remaining issues:
> >
> > > W1: Table 4 definitely provides a clearer and fairer evaluation of Spectrum tuning. A minor issue the authors may want to fix is to adjust the layout for better presentation and to ensure it does not exceed the page margins.
> >
> > The rest of the response has more or less addressed my initial concerns. However, due to the recent incident with OpenReview, it seems that I am currently unable to edit my official review, even though the deadline has not yet passed. **Therefore, I would like to note here that, after the authors’ revision, the clarity and quality of this paper have significantly improved, and I would like to update my evaluation to marginal accept (6). I hope the AC can take this into account when making the final decision.**

---

> > > ### Author Response · Authors · 2025-11-28
> > >
> > > We are glad to hear that the response has generally addressed your concerns, and you found that Table 4 "definitely provides a clearer and fairer evaluation of Spectrum Tuning".
> > >
> > > With regards to Table 4, we have now updated it so that it stays within the margins, and have uploaded the updated pdf.

---

> > > > ### Author Response · Authors · 2025-11-28
> > > >
> > > > Dear Area Chair,
> > > >
> > > > We would like to note that the reviewer found that **after the authors’ revision, the clarity and quality of this paper have significantly improved, and I would like to update my evaluation to marginal accept (6).**
> > > >
> > > > If it would at all be possible to take this elevated score into account, we would greatly appreciate it.

---

### Official Review · Reviewer_zhPy · 2025-11-02

**Soundness:** 3
**Presentation:** 2
**Contribution:** 3
**Rating:** 6
**Confidence:** 4

**Summary:**

This paper discusses three important properties of language models for adaptable inference. (1) In-context steerability: the ability to adapt to new distributions given ICL examples. (2) Valid output space coverage: generating diverse yet valid responses, and (3) Distributional alignment: matching a target output distribution (Calibration). To test these three properties (especially in-context steerability), the author first constructs the datasets Spectrum Suite, which contains data that includes natural person-to-person variations that requires the model to adapt to certain distribution given ICL examples. Using this dataset, the author found that instruction tuning, while gives good ICL elicitation and with high valid response rate, will hurt the in-context steerability and output diversity. The author further proposed the Spectrum Tuning paradigm, which let the model learn via predicting each of the sequential ICL outputs to achieve better generalizbility. The proposed methods shows improvement on all of the three properties, showing potential of increasing inference adaptability of language models with this new training paradigm.

**Strengths:**

1. The problem definition of this paper is interesting. While most papers focus on direct comparison of accuracy, the authors proposed the three properties of output that largely impact the user experience in real-world, which is often lack discussed in the benchmark results.
2. The authors proposed the ICL steerability and elicitation with clear definition. The motivation of why ICL steerability is important for inference adaptation is also clear.
3. The Spectrum Tuning method is extremely simple yet effective following the experiment results. Showing improvement on all diversity, ICL steerability and calibration.

**Weaknesses:**

Overall I think the paper well demonstrated the effectiveness of the Spectrum Tuning method. However, there're certain points about the experiment that lack clarification.

1. Spectrum Suite is an important dataset for this paper, since the author use it to evalaute the three properties. However, how this dataset is constructed is not very clear, which can hurt the soundness of the experiment results. Specifically, at line 78 to 86, how the author identified those subjective tasks is unclear. This is important in the sense that the author uses these tasks to evaluate the ICL steerability.
2. While the reason how Spectrum tuning shows improvement on ICL steerability is trivial, how it help with better diversity and calibration is unclear to me. Can the author proposed some justification about this?
3. Following 2, while Spectrum Tuning models show better diversity compare to Instruction tuning in Figure 4, the validity is lower than instruction tuning. This indicates that Spectrum tuning might not overall be a superior tuning mechanism for the valid-diveristy rate, since it can just be a model that is underfit and has higher diversity compared to instruction tuning.

**Questions:**

1. Transition from line 53 to 54 is abrupt. Shouls add a few sentences to talk about the motivation of choosing to investigate these abilities instead of jumping right into it.

---

> ### Author Response · Authors · 2025-11-20
>
> We appreciate that the reviewer finds the **“problem definition of the paper to be interesting”**, discussing **“three important properties of language models”* that impact “user experience in the real world” which are often “lack”ing in other benchmark evaluations.** We also appreciate that the reviewer found the “ICL steerability and elicitation” definition to be **“clear” and “important for inference adaptation.**
>
> We also are glad that the reviewer found our method **“extremely simple yet effective”,** and that “overall the paper well demonstrated the **effectiveness of the Spectrum Tuning method.”**
>
> Below, we hope to provide clarification regarding the weaknesses discussed:
>
> - W1: “how the dataset is constructed is not very clear”: Thank you for pointing us to this. We have added some additional details regarding dataset construction to the main paper, and have a new section in the appendix (App C1) regarding more comprehensive details relating to dataset construction. Additionally, we provide an example sequence for each task in the supplementary materials in order to aid the reader in maintaining
>
> - W2: Yes, this is a fantastic question!
> While we hope to flesh out our understanding of this mechanism in future work, our best intuition is this - It largely has to do with the fact that 1) all training tasks involve interchangeable data and 2) we shuffle the data before training.
> As a simple example, let us consider the \texttt{diffuse\_distribution} task: “Output a random country in Asia, chosen completely at random, without replacement.” In training, we collect a list of all countries in Asia, shuffle them, and finetune on them as outputs: e.g., “Brunei”, “Lebanon”, “Singapore”, “Laos”, “Vietnam”, ...
> An instruction-tuned model will often exhibit mode collapse - outputting the same country each time. Meanwhile, a base model will often output a valid country, but is heavily affected by training data frequency / n-gram statistics. In contrast, in the limit, Spectrum Tuning encourages the model to actually instantiate a uniform distribution over all countries in Asia - increasing the diversity of outputs across many samples. For distirbutional alignment and calibration, it is a similar story - base models are heavily affected by things like n-gram statistics, instruct models have uncalibrated, spiky distributions. In contrast, Spectrum Tuning in the limit encourages the model to fit the actual described distribution, (partially) overcoming n-gram frequency.
> As we believe that this could be a common question for a reader, we have added this discussion to Appendix B. Thank you very much for bringing this to our attention.
>
> - W3: We believe that it is unlikely that the difference is due to Spectrum-Tuned models being underfit in part due to the intuition we give above in W2. Specifically, post-training for instruction-tuning with RL encourages the model to output the single highest answer with highest reward - hence why validity is so high. This is true in the limit as well as empirically shown by low diversity. However, with spectrum tuning, the theoretical limit (and observed trend) of continued training is instead a diffuse distribution over all possible valid outputs. While true that Spectrum Tuning may come at a slight cost of validity compared to instruction-tuning, it comes at an empirically improved diversity and overall yield as well. Unfortunately, it is hard for us to verify whether the models are underfit empirically, as we have no additional data on which to train them.
>
> - Q1. Thank you for pointing this out - we have attempted to improve the flow of these two paragraphs.
>
> **We hope that we have been able to address some of your concerns about the paper! Please let us know if you have any additional questions.**

---

> > ### Author Response · Authors · 2025-11-28
> >
> > Hello, we just wanted to ping to see if the reviewer has had a chance to see our response, and to see which concerns we have or have not addressed and why. If you find that our response has addressed some concerns and strengthened the paper, we kindly ask that you consider raising your score. In addition, we have performed two additional sets of experiments, a temperature experiment and additional baselines. Thank you!

---

### Author Response · Authors · 2025-11-28
**Note to the Area Chair**

Dear Area Chair,

We were disappointed to hear about the modified policy, as we were optimistic that our replies had significantly addressed the reviewers' concerns and that further engagement likely would have elicited higher scores.

In case it is helpful, we would like to offer a summary of the replies, the changes we have made, and the state of the discussion.

- **All four reviewers found the problem formulation and results to be important**:
	- zhPy: "three important properties of language models” that impact “user experience in the real world” which are often “lack”ing in other benchmark evaluations.
	- suLi: “tackle fundamental, challenging questions” in post-training and has “comprehensive” experiments with results which “are novel and likely of broad interest” (suLi)
	- HCf6: “insightful results about the degradation of IT models.”
	- nA29 “important problem” the work addresses while “proposing a solution” with an “important resource” and “extensive set of experiments” which “can lead to many possibilities for interesting discussions and future work not only in post-training, but also in areas of interpretability.”
- **We made significant changes to improve the paper.** In the rebuttal period, we made the following changes to the paper:
	- **An additional temperature ablation experiment.** Reviewer HCf6 raised the hypothesis that perhaps our results might change with temperature. We ran an additional experiment, which actually strengthened our results - our method (Spectrum Tuning) maintained advantanges across temperatures, and had the best overall performance.
	- **Additional baselines and additional experiments further disentangling the effects of the method and data**. Reviewers suLi and HCf6 both expressed desire for further experiments to better understand the underlying method and data and compare to other baselines. We ran several additional experiments, and found that both our method and data were important for eliciting behavior gains (Table 4).
	- **Improved presentation and flow.**  We improved the paper structure and presentation, improved flow, added additional details, and more. Reviewer nA29 found the changes to "significantly improve the readability of the paper".
- **Both reviewers who acknowledged reading our rebuttal raised their scores.**
	- nA29 said that their "biggest and only issue with the paper is how it is structured and the overall presentation." We made significant changes to the paper structure and presentation, and a full week before the leakage, the reviewer said "I went through the paper and the changes addressed all of my concerns, significantly improving the readability of the paper. As a result, I am increasing my overall score and the presentation score." The reviewer then **raised their overall score from a 6->8**, and the presentation score from a 2->3. (**Overall score: 8**)
	- suLi said that "the response has more or less addressed my initial concerns" and that "after the authors’ revision, **the clarity and quality of this paper have significantly improved**, and I would like to update my evaluation to marginal accept (6)". (**Overall score: 6**)
- **We believe we have addressed the main concerns of the two reviewers who never engaged in discussion.**
	- HCf6's main concerns were regarding 1) a missing ablation study on temperature, and 2) a wish for additional experiments disentangling the effect of our method and our dataset. Regarding 1), we carried out an additional in-depth temperature experiment, and find that our method's advantages hold even across temperature, alleviating the concern. Regarding 2), this is also a request that reviewer suLi had, and we carried out an additional experiment. Reviewer suLi had found the additional experiment satisfactory, addressing the concern. **The reviewer never acknowledged reading the rebuttal, but we believe that our two additional experiments addressed their concerns, and were very hopeful about an increased score.** While it is obviously impossible at this point to hear back from the reviewer, we believe that our experiments significantly addressed their concerns and were optimistic that they would raise their score from a 4 to a higher score.
	- zhPy's main concerns were regarding clarity and intuition about the experiments. We fleshed out the requested parts of the paper, and **believe that we have overall addressed their stated concerns.** (**Initial overall score: 6**)


If at all possible, we would greatly appreciate it if the AC could take into consideration that the two reviewers who were able to engage both found their concerns addressed. We also believe that we have largely addressed the concerns of the two reviewers who did not engage in discussion. **We believe that the reviewers all found the problem we are studying to be important and of interest, and believe that we have significantly addressed the stated concerns.**

---

### Meta-Review · Area_Chair_9JSQ · 2026-01-05

**Summary:**

This paper investigates the trade-off between instruction-following and distributional diversity in large language models. The authors identify that current post-training methods often lead to mode collapse, where models lose the ability to generate diverse outputs or steer to novel distributions in-context. In this paper, they introduce Spectrum Suite, a comprehensive resource for evaluating distributional tasks, and propose Spectrum Tuning, a training paradigm that incorporates sequential ICL examples. Empirical results show that Spectrum Tuning successfully mitigates mode collapse, maintaining a superior diversity-validity Pareto frontier compared to standard instruction-tuned models without compromising general capabilities.

**Reviewer Concerns:**

Addressed:
- The authors provided a new ablation study demonstrating that the Spectrum Tuning method itself provides steerability gains over traditional SFT even when trained on the same data.
- The authors provided the temperature ablation experiment was conducted, showing that Spectrum-tuned models maintain their advantage across all temperature settings and offer a better Pareto frontier than instruction-tuned counterparts.
- Significant revisions were made to the abstract, introduction, and task definitions (Desiderata), which resolved initial concerns regarding the paper’s structure and readability.

Outstanding:
- While the empirical gains in calibration and alignment are well-documented, a deeper theoretical analysis of why this specific sequential training paradigm optimizes probability mass distribution remains a target for future investigation.

**Reviewer Scores:**

Reviewer nA29: Supported acceptance following the revision of the paper’s structure and the expansion of the motivation section. The reviewer highlighted the importance of the problem for post-training and interpretability research.

Reviewer suLi: Supported acceptance after the authors provided results to disentangle the effects of the dataset and the tuning method, resolving concerns about the fairness of baseline comparisons.

Reviewer zhPy: Recommended marginal acceptance, praising the unique problem definition and the practical value of the Spectrum Suite resource.

Reviewer HCf6: Keeped marginal rejection; the authors addressed the specific technical weaknesses identified—specifically the temperature ablation and method-data disentanglement—with new experimental data during the rebuttal.

---

### Decision · Program_Chairs · 2026-01-26

Accept (Poster)